# Empirical values and assumptions in the convection schemes of numerical models

Anahí Villalba-Pradas and Francisco J. Tapiador

University of Castilla-La Mancha, Earth and Space Sciences (ess) Research Group, Department of Environmental Sciences, Institute of Environmental Sciences, Avda. Carlos III s/n, Toledo 45071, Spain

*Correspondence to*: Anahí Villalba-Pradas (Anahi.Villalba@uclm.es)

**Abstract.** Convection influences climate and weather events over a wide range of spatial and temporal scales. Therefore, accurate predictions of the time and location of convection and its development into severe weather are of great importance. Convection has to be parameterized in Global Climate Models and Earth System Models as the key physical processes occur at scales much lower than the model grid size. This parameterization is also used in some Numerical Weather Prediction models (NWPs) when convection is not explicitly resolved. The convection schemes described in the literature represent the physics by simplified models that require assumptions about the processes and the use of a number of parameters based on empirical values. These empirical values and assumptions are rarely discussed in the literature. The present paper examines these choices and their impacts on model outputs and emphasizes the importance of observations to improve our current understanding of the physics of convection. The focus is mainly on the empirical values and assumptions used in the activation of convection (trigger), the transport and microphysics (commonly referred to as the cloud model) and the intensity of convection (closure). Such information can assist satellite missions focused on elucidating convective processes (e.g. the INCUS mission) and the evaluation of model output uncertainties due to spatial and temporal variability of the empirical values embedded into the parameterizations.

**Table of contents**

**Table 1.** List of acronyms.

| Acronym | Meaning | Acronym | Meaning |
|---|---|---|---|
| ADHOC | Assumed-Distribution Higher-Order Closure | HWRF | Hurricane Weather Research and Forecasting model |
| ALARO | Aire Limitée Adaptation/Application de la Recherche à l'Opérationnel (ALARO). | ICON | Icosahedral Nonhydrostatic model |
| ALE | Available Lifting Energy | IFS | Integrated Forecasting System |
| ALP | Available Lifting Power | IN | Ice Nuclei |
|  |  | INCUS | Investigation of Convective Updrafts Mission |
| AM4.0 | Atmospheric Model version 4 | IOP | Intensive Observation Period |
| AOT | Aerosol Optical Thickness | ITCZ | Intertropical Convergence Zone |
| ARM | Atmospheric Radiation Measurement | KF | Kain-Fritsch scheme |
| ARW | Advanced Research WRF | KIM | Koel isolatie maatschappij (The Netherlands Institute for Transport Policy Analysis) |
| AS | Arakawa-Schubert scheme | KWAJEX | Kwajalein Experiment |
| ATBD | Algorithm Theoretical Basis Documents | LBN | Level of Neutral Buoyancy |
| ATEX | Atlantic Trade-Wind Experiment | LCL | Lifting Condensation Level |
| BCL | Buoyant Condensation Level | LFC | Level of Free Convection |
| BMJ | Betts-Miller-Janjić | LFS | Level of Free Sinking |

| Acronym | Meaning | Acronym | Meaning |
|---|---|---|---|
| BRAMS | Brazilian developments on the Regional Atmospheric Modeling System | LMDZ | Laboratoire de Météorologie Dynamique Zoom |
| BOMEX | Barbados Oceanographic and Meteorological EXperiment | LWC | Liquid Water Content |
| CA | Cellular Automaton | MIROC | Model for Interdisciplinary Research on Climate |
| CAM | Community Atmosphere Model | MJO | Madden-Julian Oscillation |
| CAPE | Convective Available Potential Energy | MM5 | Mesoscale Model version 5 |
| CCM3 | Community Climate Model version 3 | MMF | Multiscale Model Framework |
| CCN | Cloud Condensation Nuclei | MP | Microphysics Parameterization |
| CCSM | Community Climate System Model | NAM | North American Mesoscale model |
| CDNC | Cloud Droplet Number Concentration | NAVGEM | Navy Global Environmental Model |
| CESM | Community Earth System Model | NCAR | National Center for Atmospheric Research |
| CFSv2 | Climate Forecast System version 2 | NCEP | National Centers for Environmental Prediction |
| CIN | Convective Inhibition | NWP | Numerical Weather Prediction |
| CISK | Conditional Instability of the Second Kind | PBL | Planetary Boundary Layer |
| CLUBB | Cloud Layers Unified By Binomials | PCAPE | Integral over pressure of the buoyancy of an entraining ascending parcel with density scaling |
| COARE | Coupled Ocean-Atmosphere Response Experiment | PDF | Probability Density Function |
| CP | Cumulus Parameterization | PECAN | Plains Elevated Convection at Night |
| CRCP | Cloud Resolving Convective Parameterization | PML | Potential Mixed Layer |
| CRM | Cloud Resolving Model | QE | Quasi-Equilibrium |
| CSRM | Cloud System Resolving Model | RACORO | Routine AAF (ARM Aerial Facility) CLOWD (Clouds with Low Optical Water Depths) Optical Radiative Observations |
| CWF | Cloud Work Function | RAS | Relaxed Arakawa-Schubert scheme |
| DBL | Downdraft Base Layer | RCM | Regional Climate Model |
| dCAPE | Dynamic Convective Available Potential Energy | RH | Relative Humidity |
| DDL | Downdraft Detrainment Level | RICO | Rain In Cumulus over the Ocean field campaign |
| DualM | Dual mass flux framework | SAS | Simplified Arakawa-Schubert scheme |
| DYNAMO | Dynamics of the Madden Julian Oscillation | SCAM | Single-column Community Atmosphere Model |
| ECHAM | General circulation model developed by the Max Planck Institute for Meteorology | SCM | Single Cloud Model |
| ECMWF | European Centre for Medium-Range Forecasts | SGP97 | Southern Great Plains 97 |
| EDMF | Eddy Diffusivity Mass Flux | SILHS | Subgrid Importance Latin Hypercube Sampler |
| EL | Equilibrium Level | SNU | Seoul National University |
| ENSO | El Niño-Southern Oscillation | SP | Super-Parameterization |
| EPS | Ensemble Prediciton System | | |
| ESM | Earth System Model | SPCZ | South Pacific Convergence Zone |
| EUREC4A | Elucidating the role of clouds-circulation coupling in climate | SST | Sea Surface Temperature |
| GARP | Global Atmospheric Research Program | STOMP | STOchastic framework for Modeling Population dynamics of convective clouds |
| GATE | GARP Atlantic Tropical Experiment | TC | Tropical Cyclone |
| GCM | Global Circulation/Climate Model | TKE | Turbulent Kinetic Energy |
| GEOS-5 | Goddard Earth Observing System, Version 5 model | TOGA | Tropical Ocean-Global Atmosphere |
| GFDL | Geophysical Fluid Dynamics Laboratory | TWP-ICE | Tropical Warm Pool – International Cloud Experiment |
| GFS | Global Forecast System | UIUC | University of Illinois, Urban–Champaign |
| GISS GCM | Goddard Institute for Space Studies Global Climate Model | UM | Unified Model |
| GOAmazon | Green Ocean Amazon field campaign | UNICON | Unified Convection scheme |
| HadGEM3 | Hadley Centre Global Environmental model | USL | Updraft Source Layer |
| GA2.0 | Global Atmosphere version 2 | | |
| HCF | Heated Condensation Framework | WRF | Weather Research and Forecasting model |

## 1 Introduction

Numerical Weather Prediction models, Global Climate Models, and Earth System Models (NWP, GCMs, and ESMs) generate precipitation mainly through two parameterizations: microphysics of precipitation (MP hereafter) and cumulus parameterization (CP) schemes. They produce what is known as large-scale precipitation and convective precipitation, respectively. While other schemes, such as the planetary boundary layer (PBL) parameterization used to parameterize turbulence within the PBL without accounting for moist convection also affect precipitation occurrence, the especially intricate processes by which water vapor becomes cloud droplets or ice crystals and then liquid or solid precipitation are mainly modeled by the two former modules.

The empirical values and assumptions embedded in the MP were explored in Tapiador et al. (2019a). The goal of the present paper is to provide a comprehensive account of the empirical choices and assumptions behind the representation of convective precipitation in models. There are indeed several reviews thoroughly discussing the empirical values and assumptions in convective models (e.g. De Roode et al. 2012), but they are generally focused on a particular parameter. To the best of our knowledge, there is no such extensive review of the empirical values and assumptions in the convection schemes available in the literature. Also, excellent recent reviews describing convection schemes already exist, namely Arakawa (2004) or Plant (2010), but the empiricisms in their physics have been rarely discussed. This paper aims to fill that void.

The scientific interest of our endeavor is twofold. First, it can assist dedicated satellite missions such as the Investigation of Convective Updrafts (INCUS) mission, a new Earth Venture Mission-3 (EVM-3) of three SmallSats expected to be launch in 2027 that aims to increase our knowledge of precipitation processes, and specifically on the many nuances behind convection (Stephens et al. 2020). Indeed, INCUS aims to advance our present understanding and modeling of convection on the directions identified in the 'decadal survey' (cf. Jakob, 2010; National Academies of Sciences, Engineering and Medicine, 2018, hereafter 'decadal survey'). The precise description and rationale behind the empirical parameters in the parameterization of convection can help INCUS and similar missions to focus on the key parameters, and to analyze their impacts on weather and climate models.

Another science goal of our review is to pinpoint the more relevant empirical values so systematic sensitivity studies can be readily carried out. We exemplify the latest goal showing that the spread of a perturbed ensemble of just a few parameters can be substantial. Thus, we have used the European Centre for Medium-Range Forecasts (ECMWF) Integrated Forecasting System (IFS) to perform a sensitivity experiment with seven parameters (organized entrainment, entrainment for shallow convection, turbulent detrainment, adjustment time, rain conversion, momentum transport, and shallow vs deep cloud thickness). While this is a small subset of the many parameters we have identified in this review, and the experiment is intended as an illustration of the spread in the simulations for two tropical storms, the case invites to more systematic runs in both space (global coverage) and time (decadal simulations) over the whole empirical set of parameters of any given model. The spread of the results will help to gauge the uncertainties due to the empiricisms embedded in the convection modules, and to constraint those through dedicated campaigns and targeted observations.

Precipitation is arguably the most important component of the water cycle. Extreme hydrological events in the form of floods are responsible for the loss of thousands of lives every year and great damage to property, while droughts affect water resources, livestock, and crop production. Both extremes represent important threats for human life and developing economies (e.g., Trenberth, 2011; Pham-Duc et al., 2020). Changes in the hydrological cycle also affect human activities such as the production of electricity in hydropower plants, where a better optimization of electricity production depends on water input (García-Morales and Dubus, 2007; Tapiador et al., 2011). Precipitation is also a key environmental parameter for biota. The types of vegetation and animal life that exist in a certain area are conditioned by temperature but even more by precipitation. Changes in the precipitation regime alter plant growth and survival and consequently impact the food chain (McLaughlin et al., 2002; Choat et al., 2012; Barros et al., 2014; Deguines et al., 2017). Prolonged droughts may increase the risk of wildfires, with the associated loss of local species (Holden et al., 2018). Therefore, it is not surprising that providing an accurate representation of precipitation in models is an active research topic. Specifically, in the climate realm it is already known that the effects of climate change will strongly modify the distribution and variability of precipitation around the world (Easterling et al., 2000; Dore, 2005; Giorgi and Lionello, 2008; Trenberth, 2011), posing many risks to life and human activities (Patz et al., 2005; McGranahan et al., 2007; IPCC, 2014; Woetzel et al., 2020). Thus, it is important to provide an explicit account of how models produce rain and snow in order to fully understand the outputs of the simulations.

The paper is organized as follows. A brief note on model parameterization, tuning, and the importance of convection follows (Sect. 1.1 and 1.2). Then, the main strategies to model cumulus convection are briefly presented to provide the framework to the rest of the paper (Sect. 2). The core of the review is in the following three sections, which present the assumptions and empirical values in the trigger (Sect. 3), the cloud model (Sect. 4) and the closure of the scheme (Sect. 5). The paper concludes with notes and considerations on the topic, bringing together the most important results. The acronyms used through the paper may be found in Table 1.

## 1.1 Model parameterizations

Parameterizations in numerical models address the fact that some significant physical processes in nature occur at scales much lower than the grid size used in models (Arakawa and Schubert, 1974; Stensrud, 2007; McFarlane, 2011). That is the case of convection, where spatial resolutions of at least 100 m are required to realistically solve its dynamics (Bryan et al., 2003). However, typical horizontal grid resolutions in current models range from a kilometer scale for high resolution NWP applied to a particular area, to dozens of kilometers in global NWPs, GCMs, and ESMs. With these model grids, convection is a subgrid-scale process not explicitly resolved. The physics is then represented by a simplified model that requires assumptions about the processes and the use of several parameters based on empirical values. These are used as thresholds, constraints, or mean values of a number of processes, whereas the former simplification requires a compromise between reducing complexity and a fair representation of the atmosphere.

While sometimes neglected and seldom explicit, tuning is an integral procedure of modeling (Hourdin et al., 2017; Schmidt et al., 2017; Tapiador et al., 2019a, b). It consists of estimating sensible values for the empirical parameters to reduce the

discrepancies between model outputs and observations. An example of these discrepancies is shown in Fig. 1 and Fig. 2.
Hence, tuning may have a significant influence on model results and can help identify the parts of the model that need further attention. However, blind tuning can mask fundamental problems within the parameterization, leading to non-realistic physical states of the system, compensating for errors that translate into an inappropriate budget equilibrium, or affect other metrics (Tapiador et al., 2019a). This is particularly important for climate models, since projections and simulations of future climates always include the *ceteris paribus* assumption (Smith, 2002), i.e. the tenet that in the future the multiple feedbacks between
the many processes will operate in the same way as in the present.

As stated in Couvreux et al. (2021), different approaches have been proposed to avoid tuning, including the use of convection permitting models, or machine learning approaches that replace some parameterizations by neural networks. In the former approach, the high spatial and temporal resolutions of the model allow to simulate convection directly without resorting to parameterization. Couvreux et al. (2021) proposed a new method that performs a multi-case comparison between Single Cloud
Models (SCM) and Large Eddy Simulation (LES) to calibrate parameterizations. The method uses machine learning without replacing parameterizations due to their important role in the production of reliable climate projections. Indeed, the computing power required to perform global, centennial ensemble simulations below kilometer resolution and under several anthropogenic forcings would be enormous, so improving the parameterization of convection schemes still is a thriving research field, as described below.

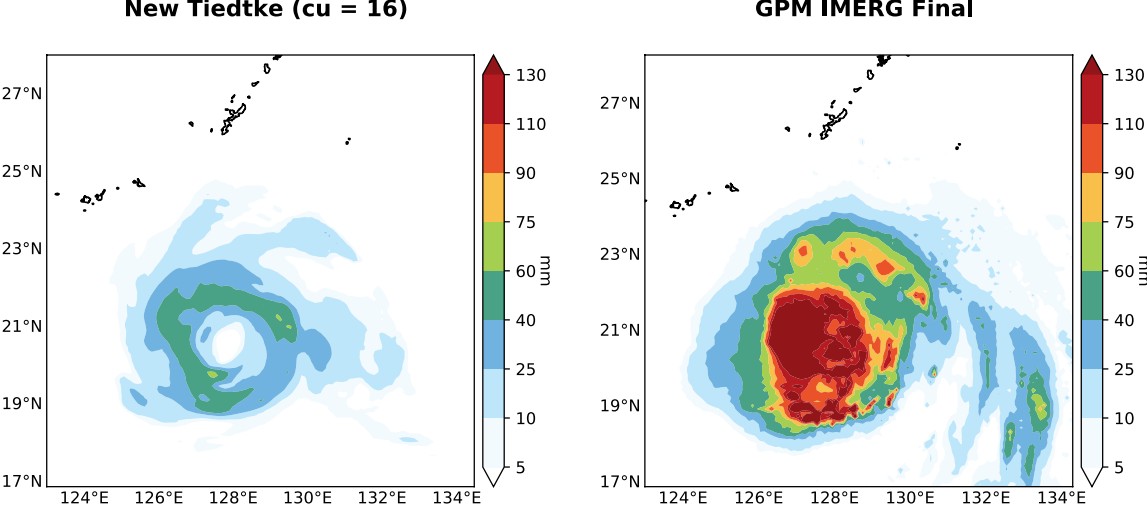

**Figure 1.** Comparison between simulated 6-hour accumulated surface liquid precipitation with the New Tiedtke convection parameterization in the WRF model using GFS initial and boundary conditions (cumulus option 16 in WRF, left) and GPM IMERG Final run (right) for Typhoon Megi on 2016/09/25 from 18.00 UTC. The accumulated precipitation includes cumulus, shallow cumulus and grid scale rain. The domain is located over the Philippine Sea with a horizontal grid size of 10 kilometers. Radiation scheme: RRTMG shortwave and longwave
schemes, boundary layer scheme: Mellor-Yamada-Janjic scheme, microphysics scheme: NSSL 2–moment scheme, land surface option: unified Noah land surface model, surface layer option: Eta similarity scheme. Spinning time: 24 hours. The typhoon was not seeded.

## 1.2 Convection: a key process in models

There is a wide range of recent research topics in convection. These topics include machine learning to parameterize moist convection (e.g., Gentine et al., 2018; O'Gorman and Dwyer, 2018; Rasp et al., 2018); stochastic parameterizations of deep convection (e.g., Buizza et al., 1999; Majda et al., 1999, 2001; Majda and Khouider, 2002; Khouider et al., 2003; Majda et al., 2003; Shutts, 2005; Plant and Craig, 2008; Dorrestijn et al., 2013; Khouider, 2014; Wang et al., 2016); the use of convective parameterization on "gray zones" (e.g., Wyngaard, 2004; Kuell et al., 2007; Mironov, 2009; Gerard et al., 2009; Yano et al., 2010; Mahoney, 2016; Honnert et al., 2020); aerosols and their influence on convection (e.g., Heever and Cotton, 2007; Storer et al., 2010; Heever et al., 2011; Morrison and Grabowski, 2013; Grell and Freitas, 2014; Kawecki et al., 2016; Peng et al., 2016; Han et al., 2017; Grabowski, 2018); microphysics impacts (e.g., Grabowski, 2015); impact of new cumulus entrainment (e.g., Chikira and Sugiyama, 2010; Lu and Ren, 2016); orographic effects on convection (e.g., Panosetti et al., 2016); new mass flux formulations (e.g., Gerard and Geleyn, 2005; Piriou et al., 2007; Guérémy, 2011; Arakawa and Wu, 2013; Park, 2014; Grell and Freitas, 2014; Yano, 2014; Gerard, 2015; Kwon and Hong, 2017; Han et al., 2017); large eddy simulations (LES) (e.g., Siebesma and Cuijpers, 1995; Brown et al., 2002; De Rooy and Siebesma, 2008; Heus and Jonker, 2008; Neggers et al., 2009; Dawe and Austin, 2013) and scale-aware cumulus parameterization (e.g., Kuell et al., 2007; Arakawa et al., 2011; Arakawa and Wu, 2013; Grell and Freitas, 2014; Zheng et al., 2016; Kwon and Hong, 2017; Wagner et al., 2018).

Such a wealth of papers illustrates the strength of this research topic in a vast number of fields. Of these, developing parameterization schemes for models is a thriving subfield, with several teams advancing the field (see Sect. 2 below). Difficulties persist, however. Convective processes have been identified in the latest decadal survey as a major source of uncertainty and dedicated efforts are needed to fill the gaps in our present knowledge of the processes involved. Owing to the influence of convection on climate and weather events over a large range of spatial and temporal scales, one of the most important objectives of the decadal survey is to improve the predictions of the timing and location of convective storms, and their evolution into severe weather. Besides the drawbacks associated with the spatial resolution, the multiscale interactions leading to the organization and evolution of convective systems are difficult to observe and represent. Improving the observed and modeled representation of natural, low-frequency modes of weather/climate variability was also identified in the decadal survey as one of the most important challenges of the coming decade. Including interactions between large-scale circulation and organization of convection such as the Madden–Julian Oscillation (MJO) or El Niño–Southern Oscillation (ENSO) aims to improve predictions by 50 % at lead times of 1 week to 2 months, which will have a high societal impact. It is therefore essential to further understand the physics and dynamics of the underlying processes, currently described with simple parameterizations in many models. Advanced observations of atmospheric convection and high-resolution models are also needed. While models will likely increase their nominal resolution in the next decade, it is also likely that global, century-long simulations from multi-ensembles under different assumptions will need to resort to parameterizing convection to reduce the computational burden.

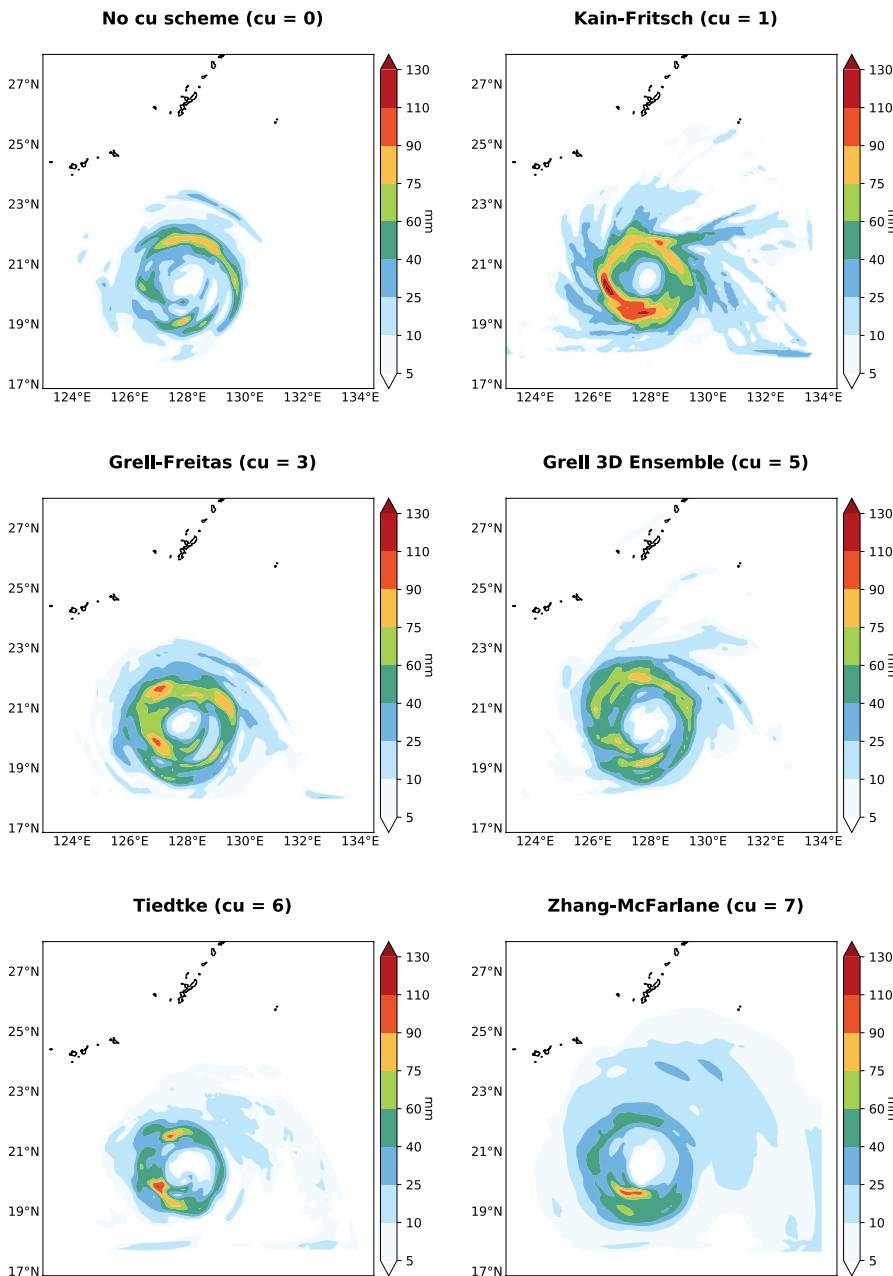

**Figure 2.** Simulated 6-hour accumulated surface liquid precipitation for Typhoon Megi without using a CP (upper left) and using five different CPs in the WRF model. The accumulated precipitation includes cumulus, shallow cumulus, and grid scale rain. The simulations start on 2016/09/25 at 18.00 UTC. The domain is located over the Philippine Sea with a horizontal grid size of 10 kilometers. Radiation scheme: RRTMG shortwave and longwave schemes, boundary layer scheme: Mellor-Yamada-Janjic scheme, microphysics scheme: NSSL 2–moment scheme, land surface option: unified Noah land surface model, surface layer option: Eta similarity scheme. Spinning time: 24 hours. GFS data were used to perform these simulations. The typhoon was not seeded.

## 2 Overview of the main schemes in cumulus convection modeling

Soon after Charney and Eliassen (1964), and Ooyama (1964) introduced the idea of cumulus parameterization, two approaches emerged: the convergence and the adjustment schemes (Arakawa, 2004). Later, a new scheme was introduced by Ooyama (1971): mass-flux parameterization. Despite all these schemes attempting to explain the interaction between cumulus clouds and the large-scale environment, the choice of empirical values for certain parameters and the simplifications in the physics yield different convective parameterizations and strategies. Indeed, as shown in Fig. 2 for the 6-hours total accumulated precipitation for Typhoo Megi, even today model outputs look different depending on the cumulus parameterization used. Many operational weather models and most climate models still use updated version of schemes described in the 1980s and 1990s. However, in recent years, new developments have emerged such as parameterizations including stochastic elements in the cumulus scheme, scale-aware approaches or the addition of processes such as cold pools, among others (Rio et al., 2019). Many of these new schemes have been developed to simulate convection across the so-called gray zones, i.e., zones where traditional convective parameterizations are no longer valid but convection cannot be yet resolved explicitly (Wyngaard, 2004). Different treatments for shallow and deep convection have been traditionally used in convection parameterizations. However, this trend has changed towards a unified treatment in recent years based on the seamless transition between shallow and deep convection observed in nature (e.g., Park, 2014).

As of 2021, the main cumulus convection schemes publicly available for NWPs are convergence schemes, adjustment schemes, mass flux schemes, cloud system resolving models (CSRM), super-parameterization (SP), PDF-based schemes, unified models, scale-aware and scale-adaptive models, and models that account for convective memory and spatial organization The purpose of this paper is not to compare the performances of the schemes but to make explicit and investigate their empirical values and assumptions, so the focus of the following section is on these. The other drive of the paper, the assumptions in convective parameterizations, concern the trigger model, the transport and microphysics, commonly referred to as the cloud model in classical convection schemes, and the closure of the scheme (Fig. 4 right). These are also described in the sections below.

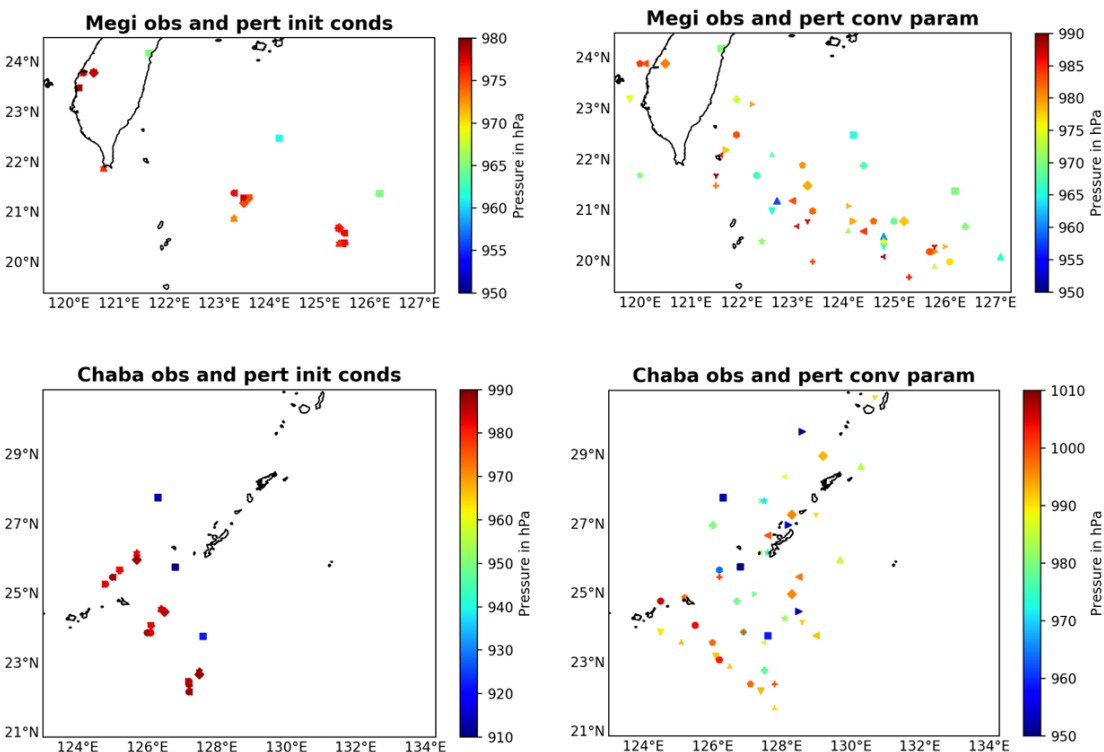

**Figure 3.** Simulated 24-hours position and pressure for Typhoon Megi (up) and Typhoon Chaba (down) using 15 ensembles in the ECMWF IFS model at 18 kilometers horizontal grid size. Each marker represents one ensemble member. Square markers indicate observations. The simulations start on 2016/09/26 at 06.00 UTC for Typhoon Megi and on 2016/10/03 at 00.00 UTC for Typhoon Chaba. Figures on the left depict observations (obs) and perturbed initial conditions (pert init conds), while figures on the right show 7 perturbed convection parameters (pert conv param) using the ECMWF Stochastically Perturbed Parameterization (SPP). The perturbed parameters are: organized entrainment, entrainment for shallow convection, turbulent detrainment, adjustment time, rain conversion, momentum transport, and shallow vs deep cloud thickness.

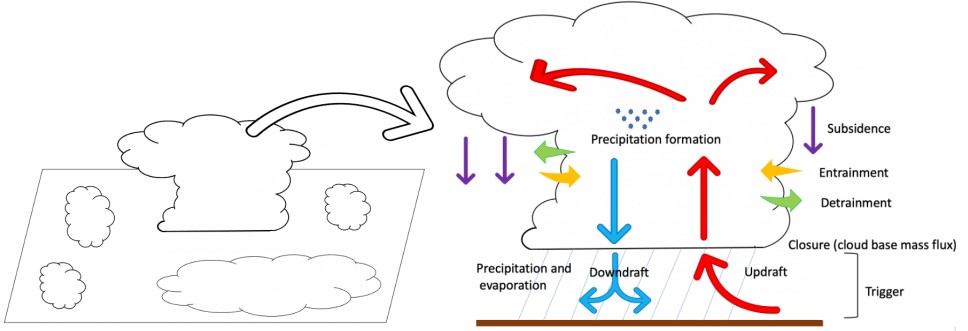

**Figure 4.** Schematic ensemble of cumulus cloud (left) and bulk convection scheme (right) showing the main components of a bulk convection scheme: trigger, updraft, downdraft, entrainment, detrainment, closure, conversion of cloud water to rainwater, precipitation and evaporation, and subsidence. Schemes based on Arakawa and Schubert (1974, left) and Bechtold (2019, right).

## 2.1 Convergence schemes: the key role of the total moisture convergence parameter

Convergence schemes consider that synoptic scale convergence destabilizes the atmosphere, while the heat released through
condensation in cumulus clouds stabilizes it. Typical examples of this approach are Charney and Eliassen (1964), Ooyama (1964) and Kuo (1974). Charney and Eliassen (1964) did not use cloud models to explain these interactions. Instead, the concept of conditional instability of the second kind (CISK) was introduced. In the Tropical Cyclone (TC) case, CISK states that cyclones provide moisture that maintains cumulus clouds, and cumulus clouds provide the heat that cyclones need. Ooyama (1964) used a similar formulation, but represented the heating released through condensation in cumulus clouds in
terms of a mass flux and considered the entrainment of ambient air. Kuo (1965, 1974) used a simple cloud model scheme to describe the interaction between a large-scale environment and cumulus clouds. One of the key assumptions in this scheme is that the total moisture convergence can be divided into a fraction $b$, which is stored in the atmosphere, and the remaining fraction $(1 - b)$, which precipitates and heats the atmosphere. This parameter was further modified by Anthes (1977), who proposed a relationship between $b$ and the mean relative humidity (RH) in the troposphere, with $b \leq 1$. In the evaluation of
rainfall rates using the Global Atmospheric Research Program Atlantic Tropical Experiment (GATE) scale phase III, Krishnamurti et al. (1980) obtained the most realistic precipitation rates for $b \approx 0$ for Kuo scheme (Kuo 1974). This value of $b$ is not realistic as it implies that no moisture is stored in the atmosphere. In a later paper, Krishnamurti et al. (1983) introduced an additional subgrid-scale moisture supply to account for the observed vertical distributions of heat and moisture that the Kuo scheme failed to reproduced, as well as to address the major limitation of $b = 0$ reported in Krishnamurti et al. (1980). The
total moisture supply was expressed as $I = (1 + \eta)I_L$, with $I_L$ the large-scale moisture supply. The authors used a multiple regression approach to find the values of $b$ and $\eta$. Another approach consists of using the wet-bulb characteristics to locally determine the partition between precipitation and moistening (Geleyn, 1985).

Due to its formulation, the Kuo scheme cannot produce a realistic moistening of the atmosphere and cannot represent shallow convection. Moreover, it assumes that convection consumes water and not energy, which violates causality (Raymond and
Emanuel, 1993; Emanuel, 1994). Despite these drawbacks, it can produce acceptable results in various applications (e.g., Kuo and Anthes, 1984; Molinari, 1985; Pezzi et al., 2008), such as in GCMs and NWP models (e.g., Rocha and Caetano, 2010; Mbienda et al., 2017). This convective parameterization scheme demands the least computational power and is thus sometimes used for large, centennial simulations.

## 2.2 Adjustment schemes: two strategies to remove instability

In adjustment schemes, the atmospheric instability is removed through an adjustment towards a reference state. Therefore, the physical properties of clouds are implicit and no cloud model has to be explicitly specified. The first proposed adjustment scheme was the moist convective adjustment by Manabe et al. (1965), also known as the hard adjustment. In this parameterization, moist convection occurs if the air is supersaturated and conditionally unstable. The instability is removed through an instantaneous adjustment of the temperature to a moist-adiabatic lapse rate, and of water vapor mixing ratio to

saturation. Moreover, all the condensed water in this process precipitates immediately. The main problems of this scheme are the production of very large precipitation rates, and its saturated final state after convection, which is rarely observed in nature (Emanuel and Raymond, 1993).

The so-called soft or relaxed adjustment schemes attempt to alleviate these problems by assuming that the hard adjustment occurs only over a fraction a of the grid area, or by specifying the final mean RH (Cotton and Anthes, 1992). For example, Miyakoda et al. (1969) defined saturation as 80 % RH, while Kurihara (1973) performed the adjustment based on the buoyancy condition of a hypothetical cloud element instead of the saturation criterion.

Further improvements to the adjustment schemes were introduced by Betts and Miller (1986), whose scheme is also known as a penetrative adjustment scheme. The authors proposed an adjustment of large-scale atmospheric temperature T and moisture q to reference profiles over a specified time scale τ (adjustment timescale).

$$(\partial T/\partial t)_{cu} = (T_{ref} - T)/\tau \qquad\qquad\qquad (1)$$

$$(\partial q/\partial t)_{cu} = (q_{ref} - q)/\tau$$

where subscript *cu* refers to cumulus convection and *ref* to the reference profile for each field. The reference profiles, different for shallow and deep convection, are quasi-equilibrium states based on observational data from GATE, Barbados Oceanographic and Meteorological Experiment (BOMEX), and Atlantic Trade-Wind EXperiment (ATEX). For the construction of the temperature reference profile, Betts (1986) used a mixing line model (Betts, 1982, 1985). Then, the moisture reference profile was calculated from the temperature profile by specifying the pressure difference between air parcel saturation level and pressure level at cloud base, freezing level, and cloud top. Therefore, the three adjustment parameters used in this scheme are the adjustment timescale $\tau$, the stability weight $W_s$, and the saturation pressure departure, $S_p$.

The sensitivity of the scheme to the adjustment parameters has been evaluated by numerous authors. For instance, Baik et al. (1990) analyzed the influence of different values of each adjustment parameter on the simulation of a tropical cyclone, while Vaidya and Singh (1997) did the same for the simulation of a monsoon depression using four sets of values, including those from Betts and Miller (1986) and Slingo et al. (1994). In all cases, the adjustment parameters had to be modified depending on the different climate regimes. While Baik et al. (1990) set $W_s = 0.95$ and $S_p = (-30, -37.5, -38)$ hPa as the optimal parameters to simulate a tropical cyclone, Vaidya and Singh (1997) obtained the best forecast for a monsoon depression with $W_s = 1.0$ and $S_p = (-60, -70, -50)$ hPa. Despite the improvements achieved through adjusting the parameters for different climate conditions, the original Betts-Miller scheme occasionally produced heavy spurious rainfall over warm water and light precipitation over oceanic regions (Janjić, 1994). To overcome this problem, Janjić (1994) proposed considering a range of reference equilibrium states, and characterizing the convective regimes by a parameter called "cloud efficiency", which is related to precipitation production and depends on cloud entropy. This parameter is the sort of empirical value that requires attention when future climates are to be simulated. The modified scheme, known as the Betts-Miller-Janjić (BMJ) scheme, is one of the most widely used adjustment schemes in NWP models (e.g., Vaidya and Singh, 2000; Evans et al., 2012; Fiori et al., 2014; Fonseca et al., 2015; García-Ortega et al., 2017), despite its large bias for light rainfall (e.g., Gallus and Segal, 2001; Jankov and Gallus, 2004;

Jankov et al., 2005). Convective adjustment schemes are computationally efficient, which makes them suitable for large-scale simulations.

## 2.3 Mass flux schemes: assuming the rates of mass detrainment and entrainment

Because of the nature of both convergence and adjustment schemes, a cloud model does not have to be explicitly specified to describe the interaction between cumulus clouds and the large-scale environment. This is not the case for the mass-flux schemes, where convective instability is removed through the vertical eddy transport of heat, moisture, and momentum. The main objective of mass flux schemes is to describe this convective vertical eddy transport in terms of convective mass flux (Plant and Yano, 2015). To do so, the total flux is defined as $\overline{\omega \psi}$, where $\omega$ is the vertical velocity and $\psi$ a physical variable, e.g., the total specific humidity $q$. Then, the total flux is expressed as the sum of a large-scale mean $\overline{\omega}\overline{\psi}$ and an unresolved eddy contribution $\overline{\omega' \psi'}$ (Reynolds averaging). Decomposing the total flux into flux contributions from cumulus cover areas and environmental regions, defining an active cloud fractional area $a$, and using again Reynolds averaging, the turbulent flux is expressed as

$$\overline{\omega' \psi'} = a\overline{\omega' \psi'}^{c} + (1-a)\overline{\omega' \psi'}^{e} + a(1-a)(\omega_c - \omega_e)(\psi_c - \psi_e) \tag{2}$$

where the overbar indexes $c$ and $e$ denote cloud (environmental) average of the fluctuations with respect to the cloud (environmental) average, and the superscripts $c$ and $e$ denote active cloud and passive environmental averages (Siebesma and Cuijpers, 1995). Commonly, the so-called "top-hat" approximation is used in convective scheme. This approximation implies neglecting the first two terms of the right-hand side in Eq. (2) in favor of the third one (the organized turbulent term due to organized updraft and compensating subsidence), which is considered dominant. Classical convective parameterizations further assumed that $a$ is small compared to the large-scale system, i.e., $a \ll 1$ (e.g., Yanai et al., 1973; Arakawa and Schubert, 1974, hereafter AS). Then, the mass flux formulation, using the definition of the convective mass flux is

$$M = -a\omega_c/g = \bar{\rho}aw_c \tag{3}$$

$$-\overline{\omega' \psi'} = gM(\psi_c - \bar{\psi}) \tag{4}$$

where $w_c$ represents the in-cloud vertical velocity. The reader is referred to Bechtold (2019) and Siebesma and Cuijpers (1995) for detailed derivation of these equations. Using a simple entraining plume model, and setting $\rho$ to unit, the continuity equations for the mass, updraft properties and vertical momentum are

$$\frac{\partial a}{\partial t} = -\frac{\partial}{\partial z}(aw_c) + E - D \tag{5.1}$$

$$\frac{\partial}{\partial t}(a\psi_c) = -\frac{\partial}{\partial z}(a\overline{w\psi}^c) + E\psi_e - D\psi_c + aS_\psi \tag{5.2}$$

$$\frac{\partial}{\partial t}(aw_c) = -\frac{\partial}{\partial z}(a\overline{w^2}^c) + Ew_e - Dw_c + a\frac{B}{1+\zeta} - \frac{\partial}{\partial z}(aP_c) \tag{5.3}$$

where E and D refer to entrainment and detrainment rates, respectively, $S_\psi$ represents sources and sinks of $\psi$, $\zeta$ is a virtual mass parameter that reduces buoyancy due to the pressure gradient force, $P_c$ includes pressure perturbations within the cloud, and the overbar denotes average values. The first formulation of this type was introduced by Ooyama (1971). The author

assumed that cumulus clouds of different sizes coexist, and that they could be represented by an ensemble of independent non-interacting buoyant elements. The definition of the so-called dispatcher function would close the parameterization. However, the author left this question open. Numerous schemes have been proposed since then mostly using the steady state assumption, i.e., $\partial/\partial t = 0$ (e.g., Yanai et al., 1973; Arakawa and Schubert, 1974; Kain and Fritsch, 1990). As mentioned in Roode et al. (2012), early mass flux schemes did not apply a vertical velocity equation for convective updrafts (Eq. 5.3) and used an *ad-hoc* assumption to specify the cloud top that depended on the vertical resolution. To alleviate this issue, recent mass flux parameterizations include a vertical velocity equation for updrafts in their formulation inspired by Simpson and Wiggert (1969):

$$\frac{1}{2}\frac{\partial w_c^2}{\partial z} = a_w B - b_w \varepsilon w_c^2 \tag{6}$$

where $\varepsilon$ is the fractional entrainment ($E = \varepsilon M$), and $a_w$ and $b_w$ are tunable parameters related to pressure perturbation and subplume contributions, respectively (see Table 2). Since then, numerous convection scheme applied equations similar to

**Table 2:** A sample values $a_w$ and $b_w$ used in Eq. (6). Based on Roode et al. (2012).

| Equation | a | b | Other constants | Reference |
|---|---|---|---|---|
| $\frac{1}{2}\frac{\partial w_c^2}{\partial z} = a_w B - 0.18\frac{w_c^2}{R}$, were $R$ is the cloud radius | 2/3 | | | Simpson and Wiggert, (1969) |
| $\frac{1}{2}\frac{\partial w_c^2}{\partial z} = a_w B - b\varepsilon w_c^2$ | 2/3 | 1 | | Bechtold et al. (2001) |
| | 1/6 | 1 | | von Salzen and McFarlane (2002) |
| | 1/3 | 2 | | Jakob and Siebesma (2003) |
| | 1 | 2 | | Bretherton et al. (2004) |
| | 1 | 1 | | Cheinet (2004); Pergaud et al. (2009) |
| | 2 | 1 | | Soares et al. (2004) |
| | 0.62 | 1 | | De Rooy and Siebesma (2010) |
| | 0.40 (core), 0.19 (updraft), 0.14 (cloud) | 1.06 (core), -0.29 (updraft), -0.02 (cloud) | | Wang and Zhang (2014) |
| $\frac{1}{2}\frac{\partial w_c^2}{\partial z} = a_w B - b_w \varepsilon w_c^2 - c_w \delta w_c^2$ | 1/6 | 1 | $c_w = 1/2$ | Gregory (2001) |
| $\frac{1}{2}(1-2\mu)\frac{\partial w_c^2}{\partial z} = a_w B - b_w \varepsilon w_c^2$ | 1 | 1/2 | $\mu = 0.15$ | Neggers et al. (2009) |
| $\frac{1}{2}\frac{\partial w_c^2}{\partial z} = a_w B - (b_w \varepsilon + c_w)w_c^2$ | 2/3 | 1 | $c_w = 0.002$ | Rio et al. (2010) |
| | 2/3 | 1.5 | $c_w = 0.002$ | Sušelj et al. (2012, 2013) |
| $\frac{1}{2}(1-\mu)\frac{\partial w_c^2}{\partial z} = B - b_w \varepsilon w_c^2$ | 1 | 0.5 | $\mu = 0.15$ | Sakradzija et al. (2016) |
| $\frac{\partial w_c^2}{\partial z} = a_w B - b_w \varepsilon w_c^2$ | 0.8 | 0.4 | | Han et al. (2017) |
| | 1 | 1.5 | | Suselj et al. (2019a, b) |

Eq. (6) for the in-cloud vertical velocity (e.g., Bechtold et al., 2001; Gregory, 2001; von Salzen and McFarlane, 2002; Jakob and Siebesma, 2003; Bretherton et al., 2004; Cheinet, 2004; Soares et al., 2004; Rio and Hourdin, 2008; Neggers et al., 2009; Pergaud et al., 2009; Rio et al., 2010; De Rooy and Siebesma, 2010; Kim and Kang, 2012; Roode et al., 2012; Sušelj et al., 2012, 2013; Wang and Zhang, 2014; Morrison, 2016a, b; Peters, 2016; Suselj et al., 2019). The reader is referred to Roode et al. (2012) for a detail derivation of Eq. (6) from Eq. (5.3) and a discussion about the values of the tunable parameters $a_w$ and $b_w$.

To overcome the gray zone issue, schemes should be scale-aware, which requires to drop the traditional assumption of $a \ll 1$ in convective parameterizations (Arakawa et al., 2011). Numerous cumulus schemes no longer use this assumption (e.g., Neggers et al., 2009; Arakawa and Wu, 2013; Grell and Freitas, 2014).

Mass flux convective parameterization schemes still are the most common convective parameterizations used in ESMs, Regional Climate Models (RCMs), and NWP models.

**2.4 Cloud System Resolving Models (CSRM)**

The performances of the previous schemes prompted the search for new strategies to model convection. Krueger (1988) put forward the CSRM idea (also known as the explicit convection, convection-permitting or cloud ensemble models) to explicitly simulate convective processes over a kilometer scale, instead of using parameterizations. Most convective parameterizations tend to produce too little heavy rain and too much light rain (e.g., Dai and Trenberth, 2004; Sun et al., 2006; Dai, 2006; Allan and Soden, 2008; Stephens et al., 2010), though these results depend on the model used for the simulations, and have problems representing diurnal precipitation cycles over land (e.g., Yang and Slingo, 2001; Guichard et al., 2004). The use of convection-permitting models can solve errors associated with other convective parameterizations (e.g., Kendon et al., 2012; Prein et al., 2013; Brisson et al., 2016), but entails higher computational costs, which limits their application in climate modeling (e.g., Wagner et al., 2018; Randall et al., 2019). They are also increasingly used in NWP though (e.g., Kain et al., 2006; Gebhardt et al., 2011). Recently, Prein et al. (2015) reviewed prospects and challenges in regional convection-permitting climate modeling.

**2.5 Super-Parameterization (SP)**

Hybrid approaches also exist. SP (also known as cloud-resolving convective parameterization (CRCP) or multiscale model framework (MMF)) is an approach between parameterized and explicit convection, which consists of replacing the convective parameterizations by 2D cloud resolving models (CRMs), or even a 3D LES model, at each grid cell of a GCM (Grabowski and Smolarkiewicz, 1999; Grabowski, 2016). Randall et al. (2003) proposed SP as "the only way to break the cloud parameterization deadlock." SP is mostly applied in GCMs (e.g., Grabowski, 2001; Khairoutdinov and Randall, 2003;

Khairoutdinov et al., 2005; Zhu et al., 2009; Jung and Arakawa, 2014; Sun and Pritchard, 2016). Several studies have compared the performance of SP with convective parameterizations, in particular using the Community Atmosphere Model (CAM). Among the most notable improvements achieved by SP in CAM are simulations of heavy rainfall events that are much more similar to observations, a better diurnal precipitation cycle over land (e.g., (Khairoutdinov et al., 2005; DeMott et al., 2007; Zhu et al., 2009; Holloway et al., 2012; Rosa and Collins, 2013), and the production of a realistic MJO (e.g., Thayer-Calder and Randall, 2009; Holloway et al., 2013). However, simulations with SP also have problems that need solving, such as the failure to simulate light rainfall rates reported by Zhu et al., (2009). The computational cost of this approach is also higher than the one for convective parameterizations (Krishnamurthy and Stan, 2015) but smaller than the computational cost for global CSRMs performing climate simulations (Randall et al., 2003).

## 2.6 PDF-based schemes

Numerous cloud and stochastic parameterizations are based on probability density functions (PDFs) of moist conserved thermodynamic variables. The so-called statistical schemes use PDFs to improve simulations of cloud clover so important in the planetary energy budget (e.g., Cahalan et al., 1994; Bony and Dufresne, 2005; Neggers and Siebesma, 2013; Bony et al., 2015). To our knowledge, the first scheme suggesting a joint PDF to compute cloud cover was that of Sommeria and Deardorff (1977) followed by Mellor (1977). These schemes used a single-Gaussian PDF. Various PDF distributions have been proposed since the formulation of the first statistical scheme, including gamma (Bougeault, 1982), Gaussian (Sommeria and Deardorff, 1977; Mellor, 1977; Bechtold et al., 1992), triangular (Smith, 1990), uniform (Le Trent and Li, 1991), lognormal (Bony and Emanuel, 2001), beta (Tompkins, 2002), and double-Gaussian (Lewellen and Yoh, 1993; Larson et al., 2002; Golaz et al., 2002a; Naumann et al., 2013). Studies such as those of Tompkins (2002) and Watanabe et al. (2009) included prognostic equations for the shape parameters of the PDF which reduced cloud cover bias when tested in ECHAM5 (Tompkins, 2002) and MIROC (Model for Interdisciplinary Research on Climate, Watanabe et al., 2009), respectively.

In the stochastic parameterization context, Craig and Cohen (2006) used statistical mechanics to describe fluctuations about a large-scale equilibrium to provide a theoretical basis for stochastic parameterizations. A PDF in the form of an exponential law provides random values of the mass flux per cloud. Plant and Craig (2008) followed this scheme and used a PDF in their formulation together with a plume model and closure assumption adapted from Kain-Fritsch scheme (Kain and Fritsch, 1990, KF hereafter), while Teixeira and Reynolds (2008) obtained a stochastic component from a normal PDF to perturb the tendencies related to the convective parameterization. Tompkins and Berner (2008) used a similar approach to perturb the initial humidity of the convective parcel and/or the humidity of the air entrained during ascent. More recently, Sakradzija et al. (2015) extended the deep convective formulation in Plant and Craig (2008) to shallow convection.

PDFs are also used to unify the representation of moist convection and boundary layer turbulence into one single scheme (see section 2.7). Randall et al. (1992) and Lappen and Randall (2001) used double-delta PDF to model the subgrid-scale variability of vertical velocity, temperature, and moisture. The scheme is called Assumed-Distribution Higher-Order Closure (ADHOC) and it is a combination of assumed distributions of higher-order closure and mass-flux closure. Bechtold et al. (1995) used a

positively skewed distribution function to account for shallow clouds. Later, Chaboureau and Bechtold (2002, 2005) extended this approach to include all types of clouds. Based on results from Larson et al. (2002) and the binormal model of Lewellen and Yoh (1993), Golaz et al. (2002a, b) proposed the Cloud Layers Unified By Binomials (CLUBB) approach that uses a double-Gaussian PDF instead of a double-delta PDF. More recently, Jam et al. (2013), Hourdin et al. (2013) and Qin et al.
(2018), represented shallow cumulus clouds with the PDF variances diagnosed from the turbulent and shallow convective processes. In the context of the EDMF framework, Cheinet (2003, 2004) used a Gaussian distribution of the thermodynamic variables, Soares et al. (2004) parameterized cloudiness with a PDF, Sušelj et al. (2012) and further modifications of the scheme (Sušelj et al., 2013, 2014; Suselj et al., 2019b, a) use a PDF to describe the moist updraft characteristics. Sakradzija et al. (2016) coupled the extension of the Plant and Craig (2008) described in  Sakradzija et al. (2015) to the Eddy Diffusivity
Mass Flux (EDMF) parameterization in ICON.

A number of studies that attempt to unify the representation of shallow and deep convection also use PDFs (e.g., Park, 2014a, b, see section 2.8).

## 2.7 Unified models

Traditionally, models have used separate parameterizations for boundary layer, shallow and deep convection. Deficiencies
associated to deep convection schemes, such as the representation of the MJO, the diurnal cycle of precipitation or the double Intertropical Convergence Zone (ITCZ), have been addressed by introducing different modifications in existing models. However, Guichard et al. (2004) showed that these modifications are not sufficient to resolve deficiencies of convection parameterization, and stressed the necessity of using and ensemble of parameterizations that represents a succession of convective regimes. Numerous attempts to merge shallow and deep convection parameterizations into a single framework can
be found in the literature (e.g., Bechtold et al., 2001; Kain, 2004; Kuang and Bretherton, 2006; Hohenegger and Bretherton, 2011; Mapes and Neale, 2011; D'Andrea et al., 2014; Park, 2014a, b). Hohenegger and Bretherton (2011) proposed a unified parameterization modifying the University of Washington (UW) shallow convection scheme (Bretherton et al., 2004; Park and Bretherton, 2009) to make it more suitable for deep convection. The authors kept the assumption that mass flux at cloud base is proportional to CIN/TKE but modified the proportionality factor following Fletcher and Bretherton (2010), who set it to
0.06. Besides, the increase of the average TKE over the depth of the boundary layer due to cold pools is included in the calculations of TKE and, therefore, in the closure. Mapes and Neale (2011) also modified the UW shallow convection scheme by making entrainment dependent on a prognostic variable called *organization* (see section 2.9). Guérémy (2011) proposed a new mass flux scheme based on continuous buoyancy, and D'Andrea et al. (2014) extended the shallow convection of Gentine et al. (2013a, b) to deep convection. Park (2014a, b) described a unified convection scheme (UNICON) for both shallow and
deep convection without relying on an equilibrium closure. The scheme diagnoses the dynamics, macrophysics and microphysics of multiple plumes. Besides, it includes a prognostic cold pool parameterization and mesoscale organized flow within the PBL, thus accounting for convective memory. Later, Park et al. (2017) modified UNICON to diagnose additional detrainment following Tiedtke (1993) and Teixeira and Kim (2008). More recently, Shin and Park (2020) developed a

stochastic UNICON model where the correlated multivariate Gaussian distribution for updraft vertical velocity and thermodynamic scalars is used to randomly sample convective updraft plumes.

In general, models split the turbulence parameterization among the PBL and moist convection (usually based on different conceptual models) simplifying the treatment of turbulence but requiring the addition of an artificial closure to match both schemes (Sušelj et al., 2014). Examples of PBL schemes that produce precipitation include the IFS EDMF, the EDMF developed by Neggers (2009) or the CLUBB scheme implemented in CAM (Thayer-Calder et al., 2015), among others. To our knowledge, the first scheme proposing a unified scheme in this way was that of Chatfield and Brost (1987), further evaluated by Petersen et al. (1999) and extended by Lappen and Randall (2001a, b) (see section 2.6 for further details). Golaz et al. (2002a, b) and Larson et al. (2002) proposed an approach to combine the representation of shallow convection and turbulence, the so-called Cloud Layers Unified By Binomials (CLUBB, section 2.6 for more details). Efforts to applied CLUBB to deep convection include those of Cheng and Xu (2006) and Bogenschutz and Krueger (2013) in CRMs or Davies et al. (2013) in a SCM. To improve deep convective simulations, Storer et al. (2015) and Thayer-Calder et al. (2015) used a Subgrid Importance Latin Hypercube Sampler (SILHS; Larson et al., 2005; Larson and Schanen, 2013) that draws samples from the joint PDF to drive microphysical processes. More recently, Larson (2020) described the unified configuration of CLUBB-SILHS, where no separated deep parameterization is used (the reader is referred to this papers for a detailed explanation of CLUBB-SILHS).

The EDMF approach was proposed by Siebesma and Teixeira (2000) and Siebesma et al. (2007) to overcome the commonly *ad-hoc* matching between the mass flux approach for convective transport within the clouds, and the eddy diffusivity approach to parameterize turbulent transport in the atmospheric boundary layer. Starting from Eq. (2), assuming $a << 1$ and identifying the third term in the equation with the convective mass flux,

$$\overline{\omega'\psi'} = \overline{\omega'\psi'}^e + M(\psi_c - \bar{\psi}) \tag{7}$$

Then, the first term in Eq. (7) is approximated by an eddy-diffusivity approach (Siebesma et al., 2007)

$$\overline{w'\psi'} \cong -K\frac{\partial\bar{\psi}}{\partial z} + M(\psi_u - \bar{\psi}) \tag{8}$$

Thus, the transport in the atmospheric boundary layer is determine as the sum of an eddy diffusivity component, defined as the product of a diffusivity coefficient $K$ and the local gradient of a thermodynamic state variable $\psi$, and a mass flux part, defined as the product of a mass flux and the difference between $\psi$ in the updraft and its horizontal mean value. The authors used a K-profile (Holtslag, 1998) for the eddy diffusivity coefficient, took the updraft fractional area as a constant and scaled the mass flux with the standard deviation of the vertical velocity $\sigma_w$. Despite originally used for dry convective boundary layers (Siebesma and Teixeira, 2000; Siebesma et al., 2007; Witek et al., 2011), numerous versions of the scheme extended it to moist convection (e.g., Soares et al., 2004; Angevine, 2005; Rio and Hourdin, 2008; Neggers et al., 2009; Neggers, 2009; Pergaud et al., 2009; Angevine et al., 2010; Köhler et al., 2011; Sušelj et al., 2012, 2013, 2014; Suselj et al., 2019).

Besides extending the EDMF model to moist convection, a number of versions included a multiple plume formulation. For example, Cheinet (2003) combined the EDMF model with the multiparcel model described in Neggers et al. (2002). With the goal of finding the least complex mass flux framework that can reproduce the smoothly varying coupling between the sub-cloud mixed layer and the shallow convective cloud layer, Neggers et al. (2009) and Neggers (2009) proposed a new formulation combining the EDMF concept with a dual mass flux (DualM) framework. There, two different updrafts are considered: a dry updraft and a moist updraft. Each of the updrafts are characterized by an area fraction (see Table 16) that varies in time, with a continuous area partitioning between moist and dry updraft. In order to realistically represent not only convectively driven boundary layers but also the transition between shallow and deep convection, Sušelj et al. (2013) further developed the scheme described in Sušelj et al. (2012). One of the main innovations included the use of a Monte Carlo sampling of the PDF of updraft properties at cloud base. Sušelj et al. (2014) described a simplified version of Sušelj et al. (2013) stochastic model where the eddy-diffusivity parameterization is based on Louis (1979), among other modifications. Later, Tan et al. (2018) extended the EDMF approach by using prognostic plumes and adding downdrafts, among other changes.

Neggers (2015) reformulated the EDMF approach in terms of discretized size densities with a limited number $n$ of bins. This new version, referred as to ED(MF)$^n$, was studied in a SCM. Han et al. (2016) proposed a hybrid EDMF parameterization where EDMF is used only for the strongly unstable PBL. For weakly unstable PBL, the scheme uses a nonlocal PBL scheme with an eddy-diffusivity countergradient approach (Deardorff, 1966; Troen and Mahrt, 1986; Hong and Pan, 1996; Han and Pan, 2011). Han and Bretherton (2019) replaced the ED parameterization in this scheme by a new TKE-based moist EDMF parameterization for vertical turbulence mixing, included downdrafts, and assumed a decreased of the updraft mass flux with decreasing grid size, which makes the scheme scale-aware. More recently, Wu et al. (2020) implemented a new downdraft parameterization in EDMF through a Mellor–Yamada–Nakanishi–Niino (MYNN) ED component. Kurowski et al. (2019) implemented a stochastic multi-plume EDMF scheme into CAM5 and Sakradzija et al. (2016) coupled Sakradzija et al. (2015) to EDMF in ICON. Several NWP models have included EDMF approaches, i.e., ECMWF (Köhler, 2005; Köhler et al., 2011), AROME (Pergaud et al., 2009), NCEP GFS (Han et al., 2016a), Navy Global Environmental Model (NAVGEM) (Sušelj et al., 2014), and the Laboratoire de Météorologie Dynamique Zoom (LMDZ; Hourdin et al., 2013) model. Recently, Bhattacharya et al. (2018) and Wu et al. (2020) implemented different versions of the EDMF scheme in WRF.

## 2.8 Scale-aware and scale-adaptive models

Wyngaard (2004) coined the terms *terra incognita* or "gray zone" to refer to zones where traditional convective parameterizations are no longer valid, but convection cannot be resolved explicitly yet. To palliate the gray zone parameterizations should become scale-aware and scale-adaptive. This means that the scheme is aware of the processes that need to be parameterized and parameterizes only those processes. Recently, Honnert et al. (2020) reviewed schemes that have been proposed for the convective boundary layer in the gray zone.

In the context of mass flux representations, the Quasi-Equilibrium (QE) assumption on a negligible small cloud area fraction $a$ has to be eliminated to make parameterizations scale-aware (Arakawa et al., 2011). Arakawa et al. (2011) and Arakawa and Wu (2013) described a seamless approach in their unified parameterization where the assumption about $a$ is eliminated, the vertical eddy transport is rederived and the parameterization is forced to converge to an explicit simulation as $a \rightarrow 1$. Following this approach, Grell and Freitas (2014) extended the Grell and Dévényi (2002) scheme based on Grell (1993) by specifying $a$ as a function of the convective updraft radius $R$ obtained from the traditional definition of entrainment $\varepsilon$ (Siebesma and Cuijpers, 1995; Simpson and Wiggert, 1969; Simpson, 1971), i.e., $\varepsilon = 0.2/R$. Later, Freitas et al. (2017) tested this scheme in the Brazilian developments on the Regional Atmospheric Modeling System (BRAMS) version 5.2 obtaining a smooth transition between convective and grid-scale precipitation even at gray zone scales.

Lim et al. (2014) modified the Simplified Arakawa-Schubert scheme (SAS; e.g, Grell, 1993; Pan and Wu, 1995; Hong and Pan, 1998; Han and Pan, 2011) in NCEP GFS by introducing a grid-scale dependency in the trigger. More recently, Kwon and Hong (2017) extended this grid-scale dependency to the convective inhibition, mass flux and detrainment of hydrometeors, and Han et al. (2017) updated the SAS scheme with a cloud mass flux that decreases with increasing grid resolution to include scale dependency.

Zheng et al. (2016) modified the adjustment time scale in KF scheme following Bechtold et al. (2008), and include a scale-aware entrainment equation, among other modifications.

Other approaches to overcome the gray zone issue include spreading subsidence to neighboring cells in Grell3D scheme (Grell and Freitas, 2014) or a hybrid parameterization for non-hydrostatic weather prediction models as described in Kuell et al. (2007). This scheme uses a traditional cumulus parameterization for mass and energy transport in the updraft and downdraft, and treats environmental subsidence by grid-scale equations. More recently, Freitas et al. (2018) implemented and tested a new version of the Grell and Freitas (2014) scheme in the the NASA Goddard Earth Observing System (GEOS). The new scheme uses a trimodal formulation with different entrainment rates that depend on the normalized mass flux profile, which is prescribed by a beta PDF, among other modifications. Gao et al. (2017) compared the performance of the traditional KF scheme with the Grell and Freitas (2014) scheme in the simulation of summer precipitation across gray zone resolutions. Better results were reported with the scale-aware scheme. An integrated package of subgrid and grid-scale parameterizations in the range 2-10 km, also known as the Modular Multiscale Microphysics and Transport (3MT), was proposed by Gerard (2007). Zheng et al. (2016) added scale-awareness to the KF scheme (Kain and Fritsch, 1990, 1993; Kain, 2004) by introducing scale dependency in in-cloud properties, such as entrainment or grid scale vertical velocity.

Another way to introduce scale-awareness and adaptivity consists in using multiple plumes instead of a single one. The first scheme using multiple plumes is that of Arakawa and Schubert (1974). Different schemes have been proposed based on multiple plumes for deep (e.g., Donner, 1993; Donner et al., 2001; Nober and Graf, 2005; Wagner and Graf, 2010) and shallow convection (e.g., Neggers et al., 2002; Sušelj et al., 2012; Neggers, 2015). Due to the lack of observations on cloud entrainment, Neggers et al. (2002) used LES results to formulate an expression for the lateral entrainment rate as a function of the vertical

velocity of each parcel, while Sušelj et al. (2012) described moist updraft characteristic through a PDF. Other parameterizations, such as those of Wagner and Graf (2010), Nober and Graf (2005) or Neggers and Siebesma (2013) make use of active population dynamics such as those in the Lotka-Volterra equations (Lotka, 1910, 1920; Volterra, 1926), where two species interact with a predator-prey behavior. Neggers (2015) also introduce population dynamics in a new EDMF called the ED(MF)$^n$. The author used bin-macrophysics, where plumes are described in terms of discrete size densities formed by a limited number $n$ of bins. The scale-adaptivity of this scheme was further evaluated in Brast et al. (2018). Population dynamics were also used by Park (2014) in his multi-cloud model in UNICON and by Hagos et al. (2018) in the STOchastic framework for Modeling Population dynamics of convective clouds (STOMP), among others. Khouider et al. (2010) described a stochastic multi-cloud model based on the deterministic multi-cloud model of Khouider and Majda (2006) but using a Markov chain lattice model. In this scheme, four possible convective states in each lattice are considered, namely clear sky, deep, congestus or stratiform clouds, that randomly evolve in time as a birth-death process (Gillespie, 1975, 1977). Dorrestijn et al. (2013, 2015) also used this approach but estimating transition probabilities from one state to another using LES results and observations, respectively. Further works followed, such as those of Deng et al. (2015) for representing the MJO, the coupling of Khouider et al. (2010) to simplified primitive equations of Frenkel et al. (2012), the use of observations to estimate transition probabilities in Peters et al. (2013), or the implementation of a stochastic multi-cloud scheme in ECHAM6.3 by Peters et al. (2017), among others. Later, Khouider (2014) improved Khouider et al. (2010) by using a coarse-grained Markov chain lattice model. Examples of stochastic parameterizations based on concepts from statistical mechanics include Plant and Craig (2008) for deep convection or Sakradzija et al. (2015, 2016) and Sakradzija and Klocke (2018) for shallow convection. Recently, Keane et al. (2014) evaluated the scale adaptivity of Plant and Craig (2008) in ICON model. Rochetin et al. (2014a, b) added a stochastic component to the trigger function in LMDZ5B and Sakradzija et al. (2016) introduced scale-awareness in ICON model by coupling the stochastic scheme described in Sakradzija et al. (2015) to the EDMF scheme. Other scale-aware schemes include CLUBB due to its limitation of the turbulent length scale to the horizontal grid spacing (Larson et al., 2012). Other studies have included a scale-dependent entrainment and/or convective time scale (e.g., Bechtold et al., 2008; Zheng et al., 2016; Han et al., 2017; Gao et al., 2020) based on results obtained in entrainment-mixing studies (e.g., Burnet and Brenguier, 2007; Lu et al., 2011, 2014; Kumar et al., 2018; Kooperman et al., 2018).

The best way to achieve scale-aware and scale-adaptive cumulus schemes is still unknown but the field is rapidly evolving.

## 2.9 Models accounting for convective memory and spatial organization

As pointed out in Davies et al. (2009), the QE hypothesis does not account for convective memory, which can be defined as the dependence of convection on their past states. Different strategies have been proposed to include it in convective parameterizations, such as the use of prognostic variables or cold pools, among others. The first scheme to include convective memory was that of Pan and Randall (1998). The authors chose a cumulus kinetic energy prognostic closure in their formulation. Later, Gerard and Geleyn (2005) also account for convective memory. Based on Bougeault (1985), the authors defined cloud base mass flux as the product of a prognostic vertical updraft velocity and a prognostic updraft fraction area,

obtained by a moist static energy closure. Gerard (2007) and Gerard et al. (2009) also used this approach and even applied it for downdrafts (Gerard et al. 2009). Piriou et al. (2007) used precipitation evaporation as the source of convective memory and related entrainment to the probability of undiluted updrafts. Mapes and Neale (2011) also chose precipitation evaporation as the source of convective memory and introduced a prognostic variable called *organization* that links precipitation evaporation with the entrainment rate. Other authors selected the precipitation at convective cloud base as the source of

convective memory and made entrainment a function of it (e.g., Hohenegger and Bretherton (2011) or Willett and Whitall (2017) in the UK Met Office model). Another way to introduce convective memory consists in using a master equation or Markov chains, such as the schemes of Hagos et al. (2018) or Khouider et al. (2010). In their extended EDMF, Tan et al. (2018) included convective memory using prognostic equations for updrafts and downdrafts and for the area fraction (see Table 16).

Evaporation of precipitation from deep convective clouds gives rise to cold pools that, when spread at the surface, are able to initiate further convective events, therefore adding memory to the system (e.g., Khairoutdinov and Randall, 2006; Rio et al., 2009; Böing et al., 2012; Schlemmer and Hohenegger, 2014). Based on this, recent studies include convective memory through cold pools (e.g., Grandpeix and Lafore, 2010; Park, 2014; Del Genio et al., 2015). The prognostic variables are the cold pool thermodynamic properties and fractional area (Grandpeix and Lafore, 2010) as well as the cold pool depth (Del Genio et al.,

2015) or the mesoscale organized flow (Park, 2014). More recently, Colin et al. (2019) performed numerical experiments to identify the source of convective memory using CRMs. The results showed that memory comes from low-level thermodynamic process such as rain evaporation, cold pools or hot thermals, among others.

Based on the "game of life" (Chopard, 2009), Bengtsson et al. (2011) used a cellular automaton (CA) in their subgrid scheme. The authors introduced convective memory by assigning a prescribed lifetime to each active cell. Bengtsson et al. (2013) also

included memory in their stochastic parameterization for deep convection using this approach in Aire Limitée Adaptation/Application de la Recherche à l'Opérationnel (ALARO). The definition of the area fraction in the cumulus scheme (Gerard et al., 2009) now includes the contribution from CA. Sakradzija et al. (2015) accounted for convective memory by considering that the cloud rate distribution in shallow convection comes from the superposition of two modes. These two modes consider passive and active clouds, respectively. In their work, the authors considered convective memory due to the

finite lifetime of individual clouds. Later, Sakradzija et al. (2016) used this scheme in the calculation of the moist-convective area fraction in EDMF in ICON.

Results from Davies et al. (2013) suggested that spatial organization could strongly affect convective memory more than the microphysics parameterizations. Later, Colin (2020) confirmed this hypothesis.

Understanding spatial organization of convection is not only important for developing stochastic and scale-aware

parameterizations but also due to its impact in the radiative-convective equilibrium (Neggers and Griewank, 2021) . Few studies have proposed parameterizations to represent convective organization in GCMs (e.g., Donner, 1993; Donner et al., 2001; Mapes and Neale, 2011; Donner et al., 2011; Khouider and Moncrieff, 2015; Moncrieff et al., 2017). Donner (1993), Alexander and Cotton (1998) and Donner et al. (2001) represented the effects of mesoscale circulations and downdrafts based

on the Leary and Houze (1980) water budget model. A similar model was developed by Gray (2000) who also considered momentum fluxes and related the strength of mesoscale circulation to detrainment of the convective mass flux. As mentioned before, Mapes and Neale (2011) introduced a prognostic variable called *organization* into the UW shallow convection scheme (Bretherton et al., 2004; Park and Bretherton, 2009). This variable, that represents the degree of subgrid organization, could affect plume calculations in terms of plume-base vertical velocity, convective inhibition, preferential rising of warmer air in updrafts, area fraction and closure, as well as a shift in the spectrum toward wider plumes with lower lateral mixing and a preferential growth in preconditioned local environments. All this would lead to more and deeper convection, and therefore more organization.

Other studies accounted for convective organization by including surface cold pools in their convective parameterizations (e.g., Rio et al., 2009; Grandpeix and Lafore, 2010; Rochetin et al., 2014a, b; Park, 2014a, b; Böing, 2016). Grandpeix and Lafore (2010) proposed a density current parameterization based on the first convective wake parameterization described by Qian et al. (1998). The impact of the cold pools on convection is implemented through two variables: the available lifting energy (ALE) provided by the density current, and the available lifting power (ALP, see section 5.1.1). In UNICON model, Park (2014a) parameterized subgrid mesoscale convective organization in terms of the evaporation of convective precipitation and downdrafts. Later, Böing (2016) described an object-based model of the organization of moist convection by cold pools inspired by Abelian sandpile models (Bak et al., 1987). The model is a two-way feedback between instability and convection, where convection and instability are represented as particles coupled to a lattice grid. The authors suggested that an object-based model could capture properties of convective organization. Stratton and Stirling (2012) used the height of the lifting condensation level as a variable to introduce convective organization into their parameterization, while Folkins et al. (2014) introduced a dependency on the local precipitation generated by the convective scheme over the past 2 h. Khouider and Majda (2006) developed a multicloud parameterization where three cloud types control the heating fields of organized convection in the tropics. It was later refined by Khouider and Majda (2008) and applied by Khouider and Moncrieff (2015) in their parameterization of organized convection in the ITCZ. Moncrieff et al. (2017) proposed a new method referred to as multiscale coherent structure parameterization (MCSP) to parameterize physical and dynamical effects of organized convection. This new approach consists in using a slantwise overturning model with a special focus on top-heavy heating and upgradient momentum transport. Despite all this proposals, the model of Donner et al. (2011) is the only operational GCM representing all aspects of mesoscale convective systems (Rio et al., 2019).

In Shutts (2005) the spatial and temporal correlations of the atmospheric mesoscale are represented by a CA. Bengtsson et al. (2011) extended the implemented CA in ECMWF Ensembe Prediciton System to be able to interact with the numerical model. Later, Bengtsson et al. (2013) introduced this CA approach in ALARO and analyzed it in a regional gray-zone resolution model over Europe. This approach produced a precipitation intensity and convective organization in better agreement with OPERA observations than results obtained from the reference model. In Bengtsson et al. (2019), CA is conditioned by a prescribed stochastically generated skewed distribution with the goal of introducing subgrid-scale organization.

Other attempts to represent convective organization include the use of a damped-driven oscillator (Davies et al., 2009), spatially coupled oscillators (Feingold and Koren, 2013) or a Markov chain lattice model (e.g., Khouider et al., 2010). Moncrieff and Liu (2006) proposed a hybrid approach to represent convective organization. Mesoscale organization is represented by explicit convectively driven circulations using a CSRM and transient cumulus by the BMJ convective parameterization (Betts, 1986; Betts and Miller, 1986; Janjić, 1994). PDF-based or spectral schemes based on a discretized distribution (e.g., Neggers et al., 2003; Wagner and Graf, 2010; Neggers, 2012; Park, 2014; Neggers, 2015) include size information into the system, which allows representing impacts of spatial organization (Neggers et al., 2019; Laar, 2019). More recently, Neggers and Griewank (2021) developed a binomial stochastic framework referred to as Binomial Objects on Microgrids (BiOMi) model, which probed to capture convective memory and simple forms of spatial organization, among other important convective behaviors, at a cheap computational cost.

This paper considers all the aforementioned convective parameterizations with emphasis on the mass-flux schemes.

# 3 Trigger function: assumptions and empiricisms

In a CP, the accurate simulation of convection greatly depends on the trigger function. The trigger function determines whether convectively unstable air at the boundary layer leads to the onset of convection and if so, activate the CP.

There are as many strategies to initiate convection as there are convection schemes. This section focuses on the assumptions and empirical values of the most important trigger functions, the starting levels, and the impacts of the trigger formulations on the simulation of convective processes. Table 3 lists the most common choices used in the main trigger function types.

## 3.1 Trigger function types

According to the physical variable used as the main trigger condition, the most used trigger functions in CPs may be classified into (1) moisture convergence, (2) cloud work function (CWF), (3) cloud base stability and convective available potential energy (CAPE) triggers, and (4) large-scale vertical velocity. Other triggers used are (5) stochastic and heated condensation framework (HCF) triggers. Table 3 lists the assumptions and empirical values used in the main trigger function types, which are discussed below.

### 3.1.1 Moisture convergence trigger

The main condition to activate convection, together with the existence of a deep layer of conditional instability, is exceeding a minimum threshold value of the vertically integrated moisture convergence. This is the case in the Anthes-Kuo scheme (Kuo, 1965; Anthes, 1977) and in the original Tiedtke scheme (Tiedtke, 1989). The latter has undergone several modifications since its publication. For instance, Gregory et al. (2000) substituted the condition of positive moisture convergence to activate deep convection by a minimum cloud depth threshold in the European Centre for Medium-Range Forecast (ECMWF) convective parameterization. Other authors replaced the moisture convergence trigger in the Tiedtke scheme by triggers based on positive

buoyancy (Zhang et al., 2011) or the existence of unstable parcel withing some height above the ground (Bechtold et al., 2004). Therefore, these schemes are no longer classified as moisture convergence trigger.

**Table 3:** A sample of empirical values and assumptions used in the main trigger function types.

| Empirical value or assumption | Choices in the literature | Reference |
|---|---|---|
| Large-scale moisture convergence | Yes | Kuo (1974); Anthes (1977); Tiedtke (1989) |
| CWF | Positive | Arakawa and Schubert (1974); Pan and Wu (1995); Han et al. (2019) |
| | Fixed value | Moorthi and Suarez (1992) |
| Large-scale vertical velocity ω | Controls $\delta T$ to trigger convection | Fritsch and Chappell (1980); Kain and Fritsch (1990); Bechtold et al. (2001); Kain (2004); Ma and Tan (2009); Berg et al. (2013) |
| CAPE | At least some CAPE | Betts (1986); Betts and Miller (1986); Janjić (1994) |
| | Must be positive | Zhang and McFarlane (1995); Xie and Zhang (2000); Bechtold et al. (2004); Zhang and Mu (2005a); Wu (2012) |
| | $CAPE > 70 \, \mathrm{J\,kg^{-1}}$ | Lin and Neelin (2003); Wu et al. (2007) |
| dCAPE | $dCAPE > 100 \, \mathrm{J\,kg^{-1}}$ | Xie and Zhang (2000); Zhang (2002); Song and Zhang (2009); Zhang and Song (2010) |
| | $dCAPE > 45 \, \mathrm{J\,kg^{-1}\,h^{-1}}$ | Song and Zhang (2018) |
| Stochastic | Stochastic perturbation in the large-scale vertical velocity $\omega$ in KF trigger | Bright and Mullen (2002) |
| | Markov process | Majda and Khouider (2002); Khouider et al. (2003); Stechmann and Neelin (2011) |
| | Bayesian Monte Carlo | Song et al. (2007) |
| | Adds a stochastic feature to the SAS trigger | Zhang et al. (2014) |
| | Adds a stochastic trigger to Emanuel (1991) | Rochetin et al. (2014a) |
| Dilute dCAPE | $dilute \; dCAPE > 70 \, \mathrm{J\,kg^{-1}}$ | Neale et al. (2008) |
| | $dilute \; dCAPE > 55 \, \mathrm{J\,kg^{-1}\,h^{-1}}$ | Song and Zhang (2017) |
| HCF | Yes | Tawfik and Dirmeyer (2014); Bombardi et al. (2015); Tawfik et al. (2017) |

### 3.1.2 CWF trigger

The first CWF trigger was introduced by AS, who proposed that convection activation depends on a threshold value of the CWF, which is defined as the integral buoyancy force of each entraining cloud between cloud base and cloud top. Several variations of the original CWF trigger function have been suggested. Tokioka et al. (1988) included a modification in the AS

to suppress deep convection in those areas where the depth of the PBL is not sufficiently thick. This modification is defined on a critical value of the entrainment rate below which deep convection is suppressed and moist air can accumulate in the

large-scale low level convergence zone. For example, the GFDL global atmosphere and land model (AM2–LM2; Anderson et al., 2004) includes this modification. In the relaxed Arakawa-Schubert scheme (RAS) (Moorthi and Suarez, 1992), the activation of convection depends on a critical value of the CWF, while the SAS scheme (Grell, 1993; Pan and Wu, 1995)

triggers convection if the CWF is positive, as shown in Table 3. Another condition to activate convection in SAS is based on the pressure difference between the starting level, i.e., the level of maximum moist static energy between the surface and 700-hPa level, and the level of free convection (LFC), which defines a threshold value for the convection inhibition (CIN) factor. With the aim of decreasing convection in large-scale subsidence regions and increasing it in large-scale convergent regions, Han and Pan (2011) modified the limit to reach the LFC, which is now proportional to large-scale vertical velocity $\omega$. Further

improvements to the SAS activation criteria include a grid-spacing dependency in the convective trigger function (Lim et al., 2014), considering the spatial resolution dependency, and a new definition of the CIN threshold value applying a scale-aware factor (Kwon and Hong, 2017). Different versions of the AS scheme are currently used in the Global Forecast System (GFS) of the National Centers for Environmental Prediction (NCEP), the Mesoscale Model 5 (MM5), the Goddard Earth Observing System model version 5 (GEOS-5), the Geophysical Fluid Dynamics Laboratory (GFDL) model, and in the WRF model.

To improve the representation of the diurnal cycle, Rio et al. (2009) proposed a new trigger for deep convection: the so-called available lifting energy (ALE). This trigger is defined as the kinetic energy of the parcel inside thermals and activates deep convection when it overcomes CIN. In this case, convection activation is controlled by lifting processes in the sub-cloud layer, e.g. gust fronts. The authors obtained a better representation of the diurnal cycle with their new formulation. Grandpeix and Lafore (2010) also used the ALE trigger in their coupled wake-convection scheme. Together with a closure based on the flux

of kinetic energy associated with thermals and the splitting of convective heating and drying, a more realistic representation of moist convection was possible. More recently, Hourdin et al. (2013) confirmed these results in the implementation of ALE trigger into a new version of the LMDZ atmospheric general circulation (LMDZ5B).

### 3.1.3 Cloud base stability and CAPE triggers

Many CPs have been proposed to simplify the formulation and implementation of the AS scheme. Among other assumptions,

some CPs substitute the convection trigger based on CWF by CAPE, defined in a similar way as CWF but without including dilution of ascending parcel by entrainment. For instance, BMJ developed a new parameterization based on empirical results, in which the activation of convection requires the existence of CAPE. In this scheme, cloud base is the lifting condensation level (LCL) of a lifted parcel with the largest CAPE in the lowest 130 hPa of the model. From there, the parcel is lifted moist adiabatically until the equilibrium level (EL) is reached. In general, the cloud top is at the level immediately beneath EL.

Moreover, deep convection continues if the cloud depth is greater than a certain value and covers at least two model layers (Baldwin et al., 2002). Finally, deep convection activates if the adjustment using reference profiles of temperature (based on a moist adiabat) and moisture (based on imposed sub-saturation at the cloud base) results in the column drying. The reference profiles computed in the BMJ scheme are different for shallow and deep convection.The scheme is currently used in NCEP North American Mesoscale model (NAM), MM5, and WRF models. Another important convective parameterization also using

a CAPE trigger is the Zhang-McFarlane scheme (Zhang and McFarlane, 1995, hereafter ZM). To improve climate simulations in the Canadian Climate Center GCM, the authors proposed a simplified version of the AS scheme that includes a positive CAPE trigger. However, it initiates convection too often during the day, which led Xie and Zhang (2000) to modify the scheme. They kept the positive CAPE condition and added a second condition based on the change of CAPE due to large-scale forcing (dCAPE). This new trigger improved the simulations of the ITCZ and MJO (Zhang, 2002; Song and Zhang, 2009; Zhang and

Song, 2010). Alternative formulations of convection trigger include the addition of an RH threshold of 80 % in the convection trigger (Zhang and Mu 2005a, b) to suppress convection if the boundary layer air is too dry. Another modification is the inclusion of dilution in CAPE calculation due to entrainment (dilute CAPE) by Neale et al. (2008) to reduce excessive precipitation over land in the simulations of ENSO.

    Unlike some of the trigger criteria already discussed, a more recent trigger function by Tawfik and Dirmeyer (2014), the HCF,

is not based on the lifting parcel method, but uses vertical profiles of temperature and humidity. First, it finds the buoyant condensation level (BCL) and determines several variables such as the buoyant mixing potential temperature, $\theta_{BM}$, , defined as the 2 m potential temperature needs to reach the BCL, and the potential temperature deficit, $\theta_{def}$, defined as the difference between the $\theta_{BM}$ and the 2 m potential temperature, or the sum of all the temperature increments needed to attain the BCL. In HCF, convection will activate when $\theta_{def} \leq 0$. The HCF trigger reduces the number of false positives compared to the parcel-

based trigger. When the HCF trigger is implemented in the NCEP Climate Forecast System version 2 (CFSv2), the representation of the Indian monsoon and tropical cyclone intensity improves (Bombardi et al., 2016). In the Community Earth System Model (CESM), the strategy improves the frequency of heavy precipitation events and reduces the overactivation of convection in the model (Tawfik et al., 2017).

**3.1.4 Large-scale vertical velocity trigger**

    Drawing on the observations in Fritsch and Chappell (1980) suggesting a positive impact of background vertical motion on convective development, Kain and Fritsch (1990) (KF) proposed a trigger based on large-scale vertical velocity. In this scheme, the first potential source layer for convection, also known as the updraft source layer (USL), is a layer of at least 60 hPa thickness that is constructed by mixing vertically adjacent layers, beginning at the surface. The temperature and pressure of

the parcel at its LCL is calculated, as well as a temperature perturbation $\delta T$, which is proportional to $\omega$ (see Table 4). If the sum of the parcel temperature and the temperature perturbation is higher than the environmental temperature, the parcel is released from its LCL. Above the LCL, the parcel is lifted upwards with entrainment, detrainment, water loading, and a vertical velocity determined by the Lagrangian parcel method (Bechtold et al., 2001). Convection is activated if the vertical velocity remains positive for a minimum depth of 3–4 km. Otherwise, the USL is moved up one model level and the procedure starts

again. This process continues until a suitable USL is found or the search has moved up above the lowest 300 hPa of the atmosphere, where the search is terminated. The lake-effect snow observations of Niziol et al. (1995) forced to reduce the

minimum cloud-depth threshold in Kain and Fritsch (1993) from 3–4 km to 2 km as they showed that clouds with this depth can produce significant snowfall. In Plant and Craig (2008), the temperature perturbation to find the USL is set to 0.2 as in Gregory and Rowntree (1990) . If no buoyant source layer can be found, then the process (like in KF) is repeated with a temperature perturbation of 0.1 K. The plume radii are determined with an exponential PDF.

Other authors, such as Ma and Tan (2009), included moisture advection in the temperature perturbation to improve the KF scheme for the case of weak synoptic forcing. Berg et al. (2013) defined a PDF that generates a range of virtual potential temperature and water vapor mixing ratio to substitute $\delta T$ in the trigger function. With this new trigger, the scheme more realistically accounts for subgrid variability within the convective boundary layer in a way. Both the modified version of the KF scheme, and the KF itself, are used in the WRF mode.

As for the trigger of shallow convection, Bechtold et al. (2001) proposed a deep convective scheme based on Kain and Fritsch (1990, 1993) but also included a shallow parameterization. In this regard, the triggering criterion is only based on a cloud-depth condition without using the temperature perturbation included in the deep scheme. Besides, cloud-depth condition and cloud radius take smaller values than those use for deep convection (see Table 4). Jakob and Siebesma (2003) also used a cloud-depth condition to decide whether deep or shallow convection is triggered. In this case, the maximum value of the cloud depth to activate shallow convection is set to 200 hPa. The procedure of finding cloud base is the same for both parameterizations.

In the shallow convection parameterization for mesoscale models described in Deng et al. (2003) based on Kain and Fritsch (1990, 1993), maximum cloud depth is set to 4 km and cloud radius is allowed to increase smoothly with time from a minimum value of 0.15 km to a maximum value of 1.50 km. Moreover, shallow convection trigger is a function of boundary layer TKE. In  Han and Pan (2011), the USL is set to the level of maximum moist static energy withing the PBL and the maximum cloud top for shallow convection is restricted by the ratio between the layer pressure and surface pressure that cannot be higher than 0.7. A cloud-depth criterion to activate shallow or deep convection is also used in this case. Han et al. (2017) developed a scale-aware parameterization for NCEP GFS, where the cloud-depth criterion is increased to 200 hPa compared to the 150 hPa used in Han and Pan (2011).

In Kain (2004) the conditions to trigger shallow convection are the same as for deep convection except for the cloud depth, that must be smaller than the one for deep convection (see Table 4). In this parameterization, the values of cloud radius are the same for both shallow and deep convection for computational reasons. Bretherton et al. (2004) triggers convection if the vertical velocity of the parcel is equal or higher than a critical value derived from the vertical velocity equation (Eq. (6)). This critical velocity takes the form $w_{crit,sh} = \sqrt{2a_w(CIN)}$, where $a_w$ is the virtual mass coefficient used in the updraft vertical velocity equation (Eq. (6), see Roode et al. (2012)). Park and Bretherton (2009) used the same triggering conditions as (Bretherton et al., 2004).

**Table 4:** A sample of empirical values and assumptions used in the trigger. (Note: subscript *sh* refers to shallow convection)

| Components | Empirical value or assumption | Choices in the literature | Reference |
|---|---|---|---|
| Buoyancy threshold | Includes a temperature perturbation $\delta T$ linked to the large-scale vertical velocity $\omega$ | $T_{LCL} + \delta T > T_{env}$, $\delta T = k\,\omega^{1/3}$, where $k$ is a unit number with dimensions K s$^{1/3}$ cm$^{-1/3}$ | Fritsch and Chappell (1980) |
| | | $\delta T = k[\omega_{LCL} - c(z)]^{1/3}$, with $k$ a unit number with dimensions K s$^{1/3}$ cm$^{-1/3}$ and $$c(z) = \begin{cases} \omega_0(z_{LCL}/2000), & z_{LCL} \leq 2000 \\ \omega_0 & z_{LCL} > 2000 \end{cases}, \text{where}$$ $\omega_0 = 2$ cm s$^{-1}$, and $z_{LCL}$ is the height (m) of the LCL above the ground | Kain and Fritsch (1990, 1993); Kain(2004) |
| | Includes a constant $\delta T$ | $\delta T = 0.2$ K | Gregory and Rowntree (1990); Bechtold et al. (2001); Plant and Craig (2008) if not USL found, search repeat with $\delta T = 0.1$ K |
| | | $\delta T = 0.65$ K | Emanuel and Živković-Rothman (1999) |
| | | $\delta T = 0.90$ K | Bony and Emanuel (2001) |
| | Includes $\delta T$ composed of horizontal $\delta T_h$ and vertical $\delta T_v$ components with associated normalized moisture advections ($R_h$ and $R_v$) | $\delta T = R_h\,\delta T_h + R_v\,\delta T_v$ | Ma and Tan (2009) |
| | Uses probability density function (PDF) | Substitute $\delta T$ in the trigger function by a generated range of virtual potential temperature and water vapor mixing ratio $q_v$ | Berg et al. (2013) |
| CIN | Must be smaller than a certain threshold | $CIN < 10$ J kg$^{-1}$ | Donner (1993); Donner et al. (2001) |
| | | $CIN < 100$ J kg$^{-1}$ | Wilcox and Donner (2007) |
| | Smaller than the Available Lifting Energy (ALE) | $|CIN| < ALE$ | Rio et al. (2009); Grandpeix and Lafore (2010); Hourdin et al. (2013) Rochetin et al. (2014) proposed a stochastic definition of ALE. |
| | Higher than a critical value and inversely proportional to large-scale vertical velocity $\omega$ | $CIN \geq CIN_{crit}$, where $CIN_{crit} \in (-120, 80) m^2 s^{-2}$ | Han et al. (2017), in addition to the condition on LFC |
| Cloud base | At LCL | | Betts (1986); Betts and Miller (1986); Janjić (1994) |
| | Height at which air parcel is mostly saturated and $T_{parcel} - T_{env} > -0.5$ K | | Tiedtke (1989); Baba (2019) |
| | Determined from sounding | Cloud base is lower than LNB | Emanuel (1991) |
| | Can be anywhere in the troposphere | | Grell (1993) |
| | Below PBL top | | Zhang and McFarlane (1995) |
| | Might be above PBL top | | Zhang and Mu (2005a) |
| | Lowest level where an adiabatic parcel is supersaturated | | Wu (2012) |
| Cloud depth | Should be higher than a certain threshold value | $CD > 300$ hPa | Kuo (1965); Anthes (1977) |
| | | $CD > 3 - 4$ km | Kain and Fritsch (1990) |

| Components | Empirical value or assumption | Choices in the literature | Reference |
|---|---|---|---|
| | | $CD > 150$ hPa | Hong and Pan (1998); Han and Pan (2011); Stratton and Stirling (2012) |
| | | $CD \geq 3$ km | Bechtold et al. (2001) |
| | | $CD > 200$ hPa | Gregory (2001); Jakob and Siebesma (2003; Bechtold et al. (20049; Han et al. (2017) |
| | Within a certain range | $0.5$ km $\leq CD_{sh} < 3$ km | Bechtold et al. (2001) |
| | | $200$ m $< CD_{sh} < 500$ m | Vogelmann et al. (2012); Lu et al. (2018) |
| | Minimum cloud depth is a function of the parcel temperature at LCL $T_{LCL}$ | $CD_{min} = \begin{cases} 4000, & T_{LCL} > 20\ ℃ \\ 2000, & T_{LCL} < 0\ ℃ \\ 2000 + 100\,T_{LCL}, & 0\ ℃ \leq T_{LCL} \leq 20\ ℃ \end{cases}$ | Kain (2004) |
| | Maximum value for shallow convection | $CD_{max,sh} = 200$ hPa | Gregory (2001); Jakob and Siebesma (2003); Han et al. (2017) |
| | | $CD_{max,sh} = 4$ km | Deng et al. (2003) |
| | | $CD_{max,sh} = 150$ hPa | Han and Pan (2011) |
| Cloud radius | Constant | | Arakawa and Schubert (1974) |
| | | $R = 1500$ m | Kain and Fritsch (1990); Bechtold et al. (2001) |
| | | $R_{sh} = 50$ m | Bechtold et al. (2001) |
| | Varies as a quadratic expression within a certain range | $0.15$ km $\leq R_{sh} \leq 1.5$ km | Deng et al. (2003) |
| | Depends on the large-scale vertical velocity at LCL $\omega_{LCL}$ | $R = \begin{cases} 1000, & W_{KL} < 0 \\ 2000, & W_{KL} > 10 \\ 1000 + W_{KL}/10, & 0 \leq W_{KL} \leq 10 \end{cases}$ <br> where $W_{KL} = \omega_{LCL} - c(z)$ (see buoyancy threshold for Kain (2004)) | Kain (2004) |
| | PDF of plume radii | | Plant and Craig (2008) |
| Cloud top | Determined by a temperature condition | Level where $T_{cloud} = T_{env}$ | Kuo (1974); Fritsch and Chappell (1980); Wu (2012) |
| | Level where buoyancy vanishes | | Arakawa and Schubert (1974); Tiedtke (1989); Wu (2012); Hong and Pan (1996) searches from the highest model down |
| | Immediately beneath EL | | Betts (1986); Betts and Miller (1986); Janjić (1994) |
| | No lower than level of minimum saturated moist static energy | | Zhang and McFarlane (1995) |
| | Determined by the vertical velocity of the parcel $w$ | Level where $w$ becomes negative | Bechtold et al. (2001) |
| | | $w = 0$ m s$^{-1}$ | Jakob and Siebesma (2003); Bechtold et al. (2004) |

| Components | Empirical value or assumption | Choices in the literature | Reference |
|---|---|---|---|
| | | $w < 0.2 \text{ m s}^{-1}$ | Wagner and Graf (2010) |
| | Function of ratio layer pressure P to surface pressure $P_s$ | Maximum value $P/P_s = 0.7$ for shallow convection | Han and Pan (2011) |
| Entrainment rate | Convection is suppressed if the entrainment in the updraft $\varepsilon^u$, is smaller than a certain threshold value $\varepsilon_c^u$ | $\varepsilon_c^u = c_{Tok}/D$, where D is the depth of the PBL and $c_{Tok}$ a constant | Tokioka et al. (1988); Anderson et al. (2004); Kim et al. (2011) says that $c_{Tok} = 0.025$ or $0.1$ in AM2, and $c_{Tok} = 0$ or $0.1$ in SNU |
| RH | Set to a constant value | $RH = 100\%$ | Manabe et al. (1965) |
| | Must be greater than a certain threshold value | $RH > 80\%$ | Zhang and Mu (2005a, b); Chikira and Sugiyama (2010)Zhang et al. (2011) |
| | | $RH > 75\%$ at lifting level | Wu (2012) |
| | | $RH > 40\%$ | Zhao et al. (2018) |
| Vertical velocity of the parcel | | $w > 0$ | Kain and Fritsch (1990); Jakob and Siebesma (2003); Bechtold et al. (2004); Kain (2004) |
| | | $w_{crit,sh} = \sqrt{2a_w(CIN)}$, where $a_w = 1$ | Bretherton et al. (2004); Park and Bretherton (2009) |

## 3.1.5 Stochastic trigger

The traditional convective triggers lead to deficiencies in the simulation of different atmospheric events, as stated in Sect. 2. A promising strategy to reduce these deficiencies is the use of stochastic triggering (Rochetin et al. 2014a, b). Instead of using a deterministic parameterization in which the subgrid-scale response is fixed to a certain resolved-scale state, the response is sampled from a suitable probability distribution (Dorrestijn et al., 2013b). For example, Majda and Khouider (2002), and Khouider et al. (2003) used a stochastic model based on CIN using a Markov process. Stechmann and Neelin (2011) used a two-state Markov jump process as their stochastic trigger. Bright and Mullen (2002) modified the KF trigger function by applying stochastic perturbation to *w*, while Song et al. (2007) included several random parameters in the trigger criteria using a Bayesian learning procedure. Zhang et al. (2014) added a stochastic term to the SAS trigger function in the Hurricane Weather Research and Forecasting model (HWRF), and Rochetin et al. (2014a, b) used LES to introduce a stochastic trigger in the Emanuel parameterization (Emanuel, 1991).

## 3.2 Starting levels

The LFC is located at, or near, the cloud base or at the top of the PBL. Different methods are applied for calculating the LFC in the literature. For instance, KF used the potential source layers for clouds (USL) in their procedure to find LFC, while Pan and Wu (1995) first determined the convection starting level and then imposed a critical depth to find the LFC (see Sect. 3.1). In their stochastic parameterization, Plant and Craig (2008) set to 50 hPa the depth of potential source layers, being the base of each 5 hPa higher than the potential layer previously tested. To trigger convection, both deep and shallow, Han and Pan (2011) set a threshold value for the pressure difference between LFC with and without sub-cloud layer entrainment. Differences

 **Table 5:** A sample of empirical values and assumptions used in the starting levels. (Note: subscript *sh* refers to shallow convection)

| Components | Empirical value or assumption | Choices in the literature | Reference |
|---|---|---|---|
| USL | Level of maximum moist static energy between surface and pressure level $p_{max}$ | $p_{max} = 700$ hPa | Grell (1993);Pan and Wu (1995); Zhang and McFarlane (1995); Han and Pan (2011); Wu (2012) |
| | | $p_{max} = 400$ hPa | Hong and Pan (1996, 1998) |
| | Layer with a minimum depth $D_{crit}$ and below the lowest 300 hPa | $D_{crit} = 60\ hPa$ | Kain and Fritsch (1990) |
| | Surface | | Park (2014a, b) |
| USL$_{sh}$ | Level of maximum moist static energy within PBL | | Han and Pan (2011) |
| LFC | Level of positive buoyancy | | Tiedtke (1989); Fritsch and Chappell (1980); Kain and Fritsch (1990); Donner (1993); Bechtold et al. (2001); Bechtold et al. (2004); |
| | Reached within an upper limit | In the lowest 300 hPa of the atmosphere | Kain and Fritsch (1990); Bechtold et al. (2004) |
| | Reached within a critical depth $D_{crit}$ from the convection starting level in proportion to vertical velocity at cloud base $\omega$ | $D_{crit} = 150$ hPa | Hong and Pan (1996, 1998) |
| | | $120$ hPa $< D_{crit} < 180$ hPa , with $D_{crit} = f(\omega, \omega_1, \omega_2)$, $\omega_1 = -5 \cdot 10^{-3}(-1 \cdot 10^{-3})$ and $\omega_1 = -5 \cdot 10^{-4}(-2 \cdot 10^{-5})$ over land(ocean) | Han and Pan (2011); Han et al. (2017) Lim et al. (2014) and Han et al. (2019) computed $\omega_1$ and $\omega_2$ assuming $\omega = f$(model horizontal resolution) Kwon and Hong (2017) added a scale-aware factor to $D_{crit}$ |
| | | $D_{crit} \propto RH$ | Han et al. (2020) |
| LFS | Level at which the temperature of a saturated mixture of equal amounts of updraft and environmental air becomes less than T$_{env}$ | | Fritsch and Chappell (1980); Tiedtke (1989); Nordeng (1994); Baba (2019) it has to be located below the level of minimum moist static energy h |
| | Level of minimum environmental saturated equivalent potential temperature between LCL and cloud top | | Kain and Fritsch (1990); Bechtold et al. (2001); Wu (2012) |
| | Level of minimum moist static energy $h$ | | Grell et al. (1991); Grell (1993) |
| | Level of minimum moist static energy $h$ if lower than the base of the detrainment layer. If not, it matches the detrainment level | | Zhang and McFarlane (1995) |
| | Near 400-hPa level. Level above the minimum moist static energy $h$ | | Pan and Wu (1995) |
| | Located within a certain range above USL | $150-200$ hPa | Kain (2004) |

higher that this threshold value, set to 25 hPa, will activate convection. Besides, the authors assumed that the convection

starting level for deep convection is at the level of maximum moist static energy *h* between the surface and the level of 700

hPa, while for shallow convection it starts at the level of maximum $h$ within the PBL. Table 5 lists a sample of the main

assumptions and empirical values used to determine the starting levels.

While the starting level for the ascending currents (updrafts) is reasonably evident, the starting level for the descending currents (downdrafts), usually called the level of free sinking (LFS), may start at any vertical level no lower than the cloud base. Several convective parameterizations, such as those proposed by Tiedtke (1989) or Bechtold et al. (2001), follow the definition suggested by Fritsch and Chappell (1980), who assumed that LFS is the level at which the temperature of a saturated mixture

of equal amounts of updraft and environmental air becomes smaller than the environmental temperature. In contrast, Grell et al. (1991) determined LFS as the minimum value of $h$, and Zhang and McFarlane (1995) matched LFS with the lowest updraft detrainment level. However, if the minimum value of $h$ is lower than the bottom level of updraft detrainment, LFS is determined as in Grell (1993).

### 3.3 Impact of trigger functions on convective models

Differences between trigger functions depend on the identification of the source layer of convective air and on how this layer of unstable air can give rise to convection. While near-surface air is selected as the source layer in some CPs (Tiedtke, 1989; Donner, 1993; Bechtold et al., 2001; Tawfik and Dirmeyer, 2014), in others, the choice is the layer of maximum moist static energy, $h$ (Arakawa and Schubert, 1974; Grell, 1993; Zhang and McFarlane, 1995; Wu, 2012). On the other hand, different convection triggers are used to determine whether unstable air turns into convection, as mentioned in the previous section.

However, the best way to construct a trigger function is still unknown and, in many cases, an *ad-hoc* formulation leads to poor performance in the activation of convection at the right location and time (Suhas and Zhang, 2014; Song and Zhang, 2017). Comparison between the performance of different trigger functions and observations from different climates leads to improvements in the formulation of the activation criteria for convection. Suhas and Zhang (2014) used three intensive observation period (IOP) datasets from the Atmospheric Radiation Measurement (ARM) program, and long-term single-

column models (SCMs) to evaluate the performance of different trigger functions (AS scheme, Bechtold scheme, Donner scheme, KF scheme, Tiedtke scheme, and four variants of the ZM scheme). The dilute dCAPE trigger function showed the best performance in both the tropics and midlatitudes, while the undilute dCAPE was as good as the dilute dCAPE only for the tropics. Furthermore, the Bechtold and the dilute CAPE trigger functions were among the best performing schemes. As a follow-up, Song and Zhang (2017) used observations from the Green Ocean Amazon (GOAmazon) field campaign to evaluate

and improve the trigger functions selected in Suhas and Zhang (2014), with the addition of the HCF. In their study, the dCAPE-type triggers also ranked first, followed by the Bechtold and HCF triggers. The undilute dCAPE trigger performed better with the inclusion of a 700-hPa upward motion, while the dCAPE trigger improved with an optimization of the entrainment rate and dCAPE threshold. Using the GOAmazon, the authors set the values for the dCAPE threshold and entrainment rate. The new values are 55 J kg$^{-1}$s$^{-1}$ for the dCAPE threshold and $2.5 \cdot 10^{-4}$m$^{-1}$ for the entrainment rate.

The convection trigger criterion plays a crucial role in the simulation of a wide number of atmospheric events. The impact of the trigger function on the correct simulation of the diurnal cycle of convection and precipitation in atmospheric models has

been widely studied, especially over land (Bechtold et al., 2004; Knievel et al., 2004; Lee et al., 2007a, b, 2008; Hara et al., 2009; Evans and Westra, 2012). The common problem in the simulation of the diurnal cycle is that it peaks too early and its amplitude is too high (Yang and Slingo, 2001; Collier and Bowman, 2004). Moreover, the diurnal cycle of precipitation peaks too early over land (in general, 2 to 4 hours before the observed maxima) (Dai, 2006), which is related to the formulation of the trigger function (Betts and Jakob, 2002; Bechtold et al., 2004). Lee et al. (2008) performed a sensitivity analysis with four different trigger functions implemented in the RAS scheme and found significant differences in the diurnal cycle of precipitation over the Great Plains in the United States. Several studies have performed sensitivity analyses and found possible ways to improve the simulation of the diurnal cycle. Models with finer resolution provided a better simulation in the amplitude, variability, and timing of the diurnal cycle (Wang et al., 2007; Sato et al., 2009). The inclusion of the effect of moisture advection in the trigger function improved the distribution and intensity of convective precipitation in the MM5 (Ma and Tan, 2009). The use of different initiation and termination conditions in the SAS scheme led to a better diurnal variation of precipitation (Han et al., 2019) although it increased the excessive precipitation and did not alleviate the bias in the phase of precipitation intensity. The modification of both the trigger and closure criteria by considering cold pools could minimize the bias in the diurnal cycle of convection (Rio et al., 2009, 2013). Another important case are the deficiencies in the simulation of the MJO (Lin et al., 2006), which are often improved by the modification of the trigger function. For example, Wang and Schlesinger (1999) found that a better representation of the MJO was possible by adding a moisture trigger to the convective parameterization used in the atmospheric general circulation model at the University of Illinois, Urban–Champaign (UIUC). Zhang and Mu (2005b) used the same approach in the National Center for Atmospheric Research (NCAR) Community Climate Model version 3 (CCM3) as well as Lin et al. (2008) in the Seoul National University (SNU) atmospheric general circulation model. Another example is a better representation of the Indian summer monsoon rainfall by the addition of HCF to the trigger function in the Climate Forecast System version 2 (CFSv2) (Bombardi et al., 2015).

The lack of "convective memory" effects in the models based on the QE assumption causes a convective parameterization to be triggered, regardless of the convection stage, as long as the convection criteria are met. Different ways to include the memory effect have been proposed, such as using prognostic cumulus kinetic energy (Pan and Randall, 1998), or an ensemble of cold pools (Grandpeix and Lafore, 2010; Del Genio et al., 2015) (see section 2.9).

## 4 Cloud model: types and choices

The cloud model represents the interaction between cumulus clouds and the large-scale environment. Thus, it determines the vertical distribution of convective heat and moisture through the parameterization of the mass flux profile, the entrainment/detrainment, and the microphysics. This section discusses the main types of mass flux and entrainment/detrainment schemes adopted in the literature, as well as the main assumptions and empirical values employed in the formulation of the cloud model.

### 4.1 Mass flux scheme types

According to the approach used to estimate the unknown quantities in Eq. (5.1), Eq. (5.2) and Eq. (5.3), mass flux schemes are classified into spectral, bulk and episodic mixing models.

#### 4.1.1 Spectral models

Spectral models represent the ensemble of clouds within a grid box with a spectrum of clouds, each of them with a cloud model. Therefore, multiple types of convection are considered in these models in contrast to the bulk ones, where the use of only one cloud model for each grid box makes necessary to decide a priori the type of convection and to characterize the cloud model by averages over the ensemble of clouds.

In spectral models, clouds within a grid box are grouped into different cloud models according to a certain parameter. The majority of spectral schemes generate an ensemble of plumes based on a distribution of entrainment rates (Arakawa and Schubert, 1974; Hack et al., 1984; Nober and Graf, 2005; Chikira and Sugiyama, 2010), although care has to be taken such that the results (convective regime) are not dominated by the least entraining parcels. Each cloud type contributes in a different amount to the ensemble mean depending on their cloud base mass flux. This type of model was original proposed by AS. Since then, the scheme has undergone several modifications, some of them make the scheme no longer a spectral model but a bulk mass flux scheme (e.g., Grell, 1993; Pan and Wu, 1995). For example, Moorthi and Suarez (1992) modified the closure in AS scheme by replacing the QE assumption for a relaxation towards the equilibrium. This scheme is also known as the Relaxed Arakawa-Schubert (RAS). Numerous studies described models based on the spectral representation (e.g., Wagner and Graf, 2010; Donner, 1993; Sušelj et al., 2012, 2013; Hong et al., 2013; Neggers, 2015; Olson et al., 2019; Brast et al., 2018; Hagos et al., 2018).

#### 4.1.2 Bulk models

The ensemble of clouds within a grid box is represented by a single cloud model, in contrast to spectral models. Yanai et al. (1973) are the main representatives of this type of scheme. In their diagnostic study, clouds are classified according to their cloud tops, and the steady plume hypothesis (Morton et al., 1956) is applied. It is assumed that all clouds have a common cloud base height, and that the values on detrainment are identical to the values inside the plume. In mesoscale models, Fritsch and Chappell (1980) and Kain and Fritsch (1992) also applied the steady hypothesis, as did Singh et al. (2019) in their study of the relationship between humidity, instability, and precipitation in the tropics. Tiedtke (1989), and Gregory and Rowntree (1990) applied the same approach as Yanai et al. (1973) in their schemes at the ECMWF, and at the U.K. Meteorological Office. The scheme used at ECMWF has undergone several modifications since then (e.g., Nordeng, 1994; Gregory et al., 2000; Li et al., 2007; Zhang et al., 2011; Kim and Kang, 2012; Stevens et al., 2013). Other studies, such as Grell (1993), changed the spectrum of cloud sizes in AS for a simple non-entraining cloud within a single grid box. Pan and Wu (1995) developed the so-called simplified Arakawa-Schubert model (SAS), which is a modified version of the model proposed by Grell (1993). The cloud

ensemble is also represented by a single non-entraining cloud and the downdraft starting level is modified to avoid excessive
cooling below cloud base. Han and Pan (2011) further modified entrainment, detrainment and cloud base mass flux in SAS to overcome unrealistic grid-scale precipitation, and develop a bulk mass flux parameterization for shallow convection. Many mass flux parameterizations use the bulk-cloud approach (e.g., Siebesma and Holtslag, 1996; Bechtold et al., 2001; Neggers et al., 2009; Yano and Baizig, 2012; Loriaux et al., 2013) with different formulations of their cloud models (i.e., formulation of the mass flux at cloud base, entrainment, detrainment, microphysics).

### 4.1.3 Episodic mixing models

Drawing on the continuous entrainment and average buoyancy used in entraining/detraining plume models in both bulk and spectral formulations, Emanuel (1991, 1994) proposed the so-called episodic mixing model, which is based on the stochastic mixing model of Raymond and Blyth (1986), and the observations of Taylor and Baker (1991), among others. Thus, Emanuel assumed that mixing is highly inhomogeneous and episodic, and applied the buoyancy sorting hypothesis (Telford, 1975; Taylor and Baker, 1991), which is the basis of a number of cumulus parameterizations (e.g., James and Markowski, 2010; Park, 2014a), especially those focused on shallow convection (e.g., Bretherton et al., 2004; De Rooy and Siebesma, 2008; Neggers et al., 2009; Pergaud et al., 2009). The Emanuel scheme and its modified versions (Emanuel and Živković-Rothman, 1999; Grandpeix et al., 2004; Peng et al., 2004) are widely used in RCMs (e.g., Zou et al., 2014; Raju et al., 2015; Bhatla et al., 2016; Gao et al., 2016; Kumar and Dimri, 2020).

The aforementioned mass flux scheme types are explained from the point of view of the ascending currents. However, convective downdrafts, i.e., descendent currents caused by evaporation of condensate and rainwater loading, should be taken into account. Simply put, they may be considered as bottom-up updrafts. Downdrafts are of great importance in atmospheric convection. As Plant and Yano (2015) highlighted, they have opposite effects on the organization and evolution of convective systems. The transport of cooler and drier air into the sub-cloud layer may stabilize it and therefore inhibit convection or may lead to the development of new convective elements if downdrafts cause an increase in low-level convergence. The majority of convective parameterizations include downdrafts with assumptions about their starting level, entrained and detrained air, or the amount of condensate available for evaporation. However, many schemes, such as Grell (1993), the ZM scheme used in CESM, or the Tiedtke scheme in the ECHAM model, have described downdrafts as simple saturated plumes, i.e., "inverse plume", with a mass flux proportional to the updraft mass flux (Thayer-Calder, 2012). Other authors have proposed a more complex parameterization including unsaturated downdrafts in their formulations and a downdraft mass flux based on Eq. (5.1), Eq. (5.2) and Eq.(5.3) (e.g., Emanuel, 1991; Xu et al., 2002).

### 4.2 Entrainment and detrainment

The mixing of air masses due to entrainment of environmental air into clouds and detrainment of cloudy air into the environment are key processes in convective parameterizations (Blyth, 1993; Luo et al., 2010; Donner et al., 2016) as they

modify the vertical profiles of heat and moisture within cloudy air. Sanderson et al. (2008) identified the entrainment rate as one of the dominant parameters affecting climate sensitivity after evaluating thousands of GCM simulations. Other authors, such as Rougier et al. (2009), Klocke et al. (2011) and Zhao (2014) have obtained similar conclusions in their analyses. In addition, the influence of convective detrainment of water vapor and hydrometeors from cumulus clouds is an important source of water that strongly impacts climate simulations (e.g., Ramanathan and Collins, 1991; Lindzen et al., 2001).

In this section, attention is drawn to the most important model types of entrainment and detrainment, the main assumptions and empirical values used in the literature, and the impact that the different formulations have in convective models. The main assumptions and empirical values used in the formulation of entrainment and detrainment are listed in Tables 6 and 7 and in Tables 8 and 9, respectively.

### 4.2.1 The choice of lateral vs cloud-top entrainment

Since Stommel (1947) provided the first description of cumulus cloud dilution by entrainment of environmental air, two conceptual models are still competing: the lateral entrainment model and the cloud-top entrainment model.

In the lateral entrainment model, Stommel (1947) considered that environmental air enters the cloud through the lateral cloud edges and continuously dilutes cloudy air during its ascent, regardless of whether it is considered a plume or a bubble. Several aircraft observations and experiments in water tanks (Turner, 1962; Morton, 1965) contributed to the formulation of the lateral entrainment theory. However, authors such as Warner (1970) pointed out the deficiencies of this theory in predicting the right profile of liquid water content (LWC).

In order to address these deficiencies, Squires (1958) proposed another entrainment model, the cloud-top entrainment. This author suggested that environmental air enters the cloud predominantly at or near the cloud top, descends through penetrative downdrafts created by evaporative cooling, and dilutes the cloud by turbulent mixing. Paluch (1979) provided more evidence for cloud-top entrainment in her study on cumulus clouds over Colorado. The author found that the cloud water-mixing ratio and the wet equivalent potential temperature follow a line at a single level, the so-called "mixing line", which connects cloud base and cloud top. Paluch interpreted it as evidence for a two-point mixing scenario. Further studies (Boatman and Auer, 1983; Lamontagne and Telford, 1983; Jensen et al., 1985; Reuter and Yau, 1987) confirmed Paluch's results. However, several authors have criticized the mixing line source levels (e.g., Blyth et al., 1988; Malinowski and Pawlowska-Mankiewicz, 1989; Raga et al., 1990; Grabowski and Pawlowska, 1993; Neggers et al., 2002; Zhao and Austin, 2005), and the interpretation of the mixing line (e.g., Betts and Albrecht, 1987; Taylor and Baker, 1991; Grabowski and Pawlowska, 1993; Siebesma, 1998; Böing et al., 2014).

Which of the two models predominates in cumulus convection remained unclear for many years. The increase in computational power in recent decades has promoted the use of LES to study entrainment and detrainment mainly in shallow cumulus clouds. Several authors, such as Heus et al. (2008) and Böing et al. (2014), have applied LES to identify the dominant process in mixing in cumulus clouds, concluding that cloud-top entrainment is insignificant compared to lateral entrainment.

### 4.2.2 Main empirical values in entrainment and detrainment formulations

Aircraft observations and experiments in water tanks (Turner, 1962; Morton, 1965) led to the formulation of the lateral entrainment theory, which anticipates that the fractional entrainment rate (hereafter entrainment rate) changes with the cloud radius (Malkus, 1959; Squires and Turner, 1962; Simpson and Wiggert, 1969; Simpson, 1971)

$$\frac{1}{M}\frac{\partial M}{\partial z} = \varepsilon \simeq \frac{C}{R}, \tag{9}$$

where $M$ is the mass flux, $z$ is the height, $\varepsilon$ denotes the entrainment rate, $C$ is a constant, and $R$ is the radius of the rising plume. These first parameterizations set $C = 0.2$ based on laboratory results. As De Rooy et al. (2013) pointed out in their review article on entrainment and detrainment in cumulus convection, many cloud models still use this formulation (e.g., Arakawa and Schubert, 1974; Kain and Fritsch, 1990; Donner, 1993), sometimes assuming a constant entrainment rate.

Houghton and Cramer (1951) improved this theory by taking into account the increase of vertical velocity due to buoyancy. Thus, the authors distinguish between dynamical entrainment due to larger-scale organized inflow, $\varepsilon_{\mathrm{dyn}}$, and turbulent entrainment caused by turbulent mixing, $\varepsilon_{\mathrm{turb}}$. The turbulent entrainment rate is related to the flux across the updraft boundary, which is often described with an eddy diffusivity approach (Kuo, 1962; Asai and Kasahara, 1967; De Rooy et al., 2013; Cohen et al., 2020). Under the eddy diffusivity approach, the eddy flux is modelled by a downgradient and an eddy diffusivity. that for the case of the turbulent entrainment is proportional to the radial scale of a plume (used as a mixing length) and the turbulent velocity scale of the environment. The change of mass flux with height, including the detrainment $\delta$ of negative buoyant mixtures, is given by

$$\frac{1}{M}\frac{\partial M}{\partial z} = \varepsilon_{\mathrm{dyn}} + \varepsilon_{\mathrm{turb}} - \delta_{\mathrm{dyn}} - \delta_{\mathrm{turb}}. \tag{10}$$

Tiedtke (1989) and Nordeng (1994) assumed that turbulent entrainment is inversely proportional to cloud radii, as in Simpson and Wiggert (1969) and Simpson (1971). They used typical cloud sizes, based on observations, for different types of convection to fix the values of entrainment rates. For penetrative and midlevel convection, the entrainment rate was fixed to $\varepsilon_{\mathrm{turb}} = 1 \cdot 10^{-4}\,\mathrm{m}^{-1}$. This is a typical value for tropical clouds as showed in the analysis of aircraft observations in Simpson (1971). For shallow convection, the entrainment rate was based on typical values for large trade cumuli, $\varepsilon_{\mathrm{turb}} = 3 \cdot 10^{-4}\,\mathrm{m}^{-1}$ (Nitta, 1975). Gregory and Rowntree (1990) also assumed a turbulent entrainment rate, but inversely proportional to the height, while in Bechtold et al. (2008), $\varepsilon_{\mathrm{turb}}$ is $\mathrm{O}(1 \cdot 10^{-3}\,\mathrm{m}^{-1})$ in better agreement with CRM results, and also relative humidity dependent, which turned out to be important to represent realistic tropical variability (Table 6). Dynamical entrainment $\varepsilon_{\mathrm{dyn}}$ is proportional to moisture convergence and occurs only in the lower part of the cloud layer up to the level of strongest vertical ascent in Tiedtke (1989). In Nordeng (1994), it is based on momentum convergence. Gregory and Rowntree (1990) did not include it in their parameterization, whereas in Bechtold et al. (2008), it depends on RH and is only applied to deep convection. For downdraft, Bechtold et al. (2014) set $\varepsilon_{\mathrm{turb}} = 3 \cdot 10^{-4}\,\mathrm{m}^{-1}$ and $\varepsilon_{\mathrm{dyn}}$ as a function of $B$. A common practice in the definition of entrainment rates for downdraft consists in assuming a similar parameterization as for updrafts (Table 7).

Kain and Fritsch (1990) introduced another type of parameterization based on the buoyancy sorting. In their parameterization, homogeneous mixing of cloudy and environmental air was assumed, leading to mixtures with different buoyancy properties that have the same probability of occurrence. Moreover, the authors modified Eq. (9) to make it pressure dependent. The fraction of environmental air that makes the mixture neutrally buoyant is the so-called critical mixing fraction $\chi_c$, which determines whether a mixture entrains or detrains after mixing. Thus, entrainment of positive buoyant mixtures occurs if $\chi < \chi_c$, while $\chi > \chi_c$ leads to immediate detrainment of negative buoyant mixtures. Therefore, detrainment can occur at any level where $\chi > \chi_c$, unlike in the AS scheme, where only the cloud top detrainment is considered. Moreover, the maximum entrainment rate is proportional to pressure and inversely proportional to updraft radius. However, the KF scheme had deficiencies, such as excessive detrainment or the production of unrealistic deep saturated layers. In newer versions of the KF scheme, a mitigation of unrealistic deep saturated layers is achieved by assuming that the entrainment of environmental air cannot be lower than 50 % of the total environmental air involved in the mixing process in the updraft, and that cloud radius depends on the convergence of the sub-cloud layer (Kain, 2004). Recently, Zheng et al. (2016) modified the minimum entrainment equation in Kain (2004) to include both organized and turbulent entrainment. The authors made the equation scale-dependent and expressed it in terms of sub-cloud layer depth instead of cloud radius. Another scheme based on the buoyancy-sorting hypothesis, but assuming episodic mixing, is the Emanuel scheme (Emanuel, 1991), where, in contrast to the KF scheme, the resulting mixtures just ascend or descend to their level of neutral buoyancy to detrain.

Other approaches use in-cloud quantities instead of only the environmental quantities to estimate the entrainment rate. For instance, Gregory (2001) proposed an entrainment rate that depends on $B$ and inversely on the square of the updraft speed $w$ calculated using Eq. (6). The value of $a_w$ also comes from the equation and is selected by comparing SCM simulations against LES/CRM studies and available observations. This parameterization deals with both shallow and deep convection. What distinguishes one type of convection from another is the value of a constant $C_e$, whose values were specified by using a SCM in ECMWF model.

Apart from buoyancy, another environmental quantity that might influence entrainment, and therefore convection, is RH. A number of studies have analyzed the effect of RH in parameterization of entrainment/detrainment rates, drawing different conclusions. For instance, Jensen and Del Genio (2006) found a positive correlation between entrainment rate and RH in their analysis of remote sensing observations and soundings at Nauru Island, while Bechtold et al. (2008) and Zhao et al. (2018) found a negative correlation using the Atmospheric Model version 4 (AM4.0). The same conclusion was achieved by Stirling and Stratton (2012) using a CRM formulation and the Met Office Unified Model (Met Office UM).

Mapes and Neale (2011) addressed the so-called "entrainment dilemma", in which the excessive entrainment values tend to excessively restrain convection, while insufficient entrainment values abundantly ease its activation. To overcome this, they proposed a new formulation of the entrainment rate dependent on a prognostic variable called *organization*, which expresses the interaction between the environment and convection. In their formulation, the rain evaporation rate controls the *organization* and produces more deep convection for lower values of the entrainment rate.

The previous discussion about entrainment and detrainment rates was focused on deep convective schemes with some references to unified schemes. However, parameterizations of these processes are also important in shallow convection. Tiedtke (1989) fixed the entrainment and detrainment rates for shallow convection to $\varepsilon = \delta = 3 \cdot 10^{-4}$ m$^{-1}$ based on typical values for large trade cumuli (Nitta, 1975). Using LES based on BOMEX, Siebesma and Cuijpers (1995) found typical values of entrainment for the core between $1.5 \cdot 10^{-3}$ m$^{-1}$ and $2 \cdot 10^{-3}$ m$^{-1}$ and around $3 \cdot 10^{-3}$ m$^{-1}$ for the updraft. Siebesma (1998) found typical values for entrainment in shallow convection in the range $1.5 - 2.5 \cdot 10^{-3}$ m$^{-1}$. In their revision and performance analysis of the ECMWF IFS, Gregory et al. (2000) found values of $\varepsilon = 1.2 \cdot 10^{-3}$ m$^{-1}$ at cloud base and $\varepsilon = 3 \cdot 10^{-3}$ m$^{-1}$ 150 hPa above it employing a control physics package that included a cloud scheme based on Tiedtke (1989, 1993).

Grant and Brown (1999) and Grant and Lock (2004) described a similarity theory for shallow convective transport. In this theory, buoyancy production and turbulent dissipation are assumed to nearly balance within QE shallow convective fields. As for the entrainment formulation, it is scaled based on observable quantities such as CAPE or mass-flux at cloud base with a constant $A_\varepsilon$ that represents the fraction of TKE available for entrainment. The value of this constant is derived from LES results. Kirshbaum and Grant (2012) used this formulation with $A_\varepsilon = 0.06$. Drueke et al. (2019) found also used this TKE similarity theory for cloud ensembles to retrieve values of entrainment rates based on sub-cloud and environmental conditions. Besides, the authors compared this method with the parcel model of Jensen and Del Genio (2006), which coupled surface remote sensing observations and soundings at Nauru Island to a parcel model, and Entrainment Rate In Cumulus Algorithm (ERICA) proposed by Wagner et al. (2013), which uses an algorithm to retrieve values of entrainment from ground-based remote sensing observations. The analysis was performed using LES simulations of a range of shallow cumulus over ocean and land showing a strong contrast in entrainment between them, as well as a lower dilution for wider clouds. The parcel method and TKE similarity theory better capture the sensitivity within continental cumuli and showed a lower mean error compared to ERICA. The diurnal variations of entrainment within continental shallow cumulus were only reproduced by the TKE method. With this method, the authors found values of $A_\varepsilon$ in the range $0.037 - 0.035$. More recently, Kirshbaum and Lamer (2021) performed a climatological sensitivity analysis of shallow cumulus entrainment in oceanic and continental locations using the parcel method and the TKE as in Drueke et al. (2019). Four years of observations at two ARM observatories were used. The analysis confirmed the results obtained by Drueke et al. (2019) and identified other sources of entrainment variability such as sub-cloud wind speed in oceanic flows and cloud base mass flux in individual cumuli. Median values of entrainment at a continental site range between 0.5 and 0.6 km$^{-1}$ and between 1.0 and 1.1 km$^{-1}$ at the oceanic site.

Neggers et al. (2002) developed a new formulation using LES. The authors proposed an entrainment rate inversely proportional to a turnover timescale that seems to be independent of cloud depth, and the vertical velocity of the parcel. Thus, each parcel will have its own entrainment rate depending on their vertical velocity. For the ensemble of parcels, the fractional entrainment rate is of the order of the values shown in Siebesma and Cuijpers (1995). Sušelj et al. (2012) followed Neggers et al. (2002) but with a different value of the turnover timescale (see Table 6). Model results using a SCM probed to be sensitive to the choices of this parameter.

In their EDMF model, Soares et al. (2004) used a constant entrainment rate within the cloud layer following the entrainment rate in Siebesma (1998), while in the sub-cloud layer the entrainment is inversely proportional to height.

Bretherton et al. (2004) proposed an entrainment formulation similar to that of KF but modified $\chi_c$ by defining a critical eddy-mixing distance $d_c$ based on observations and LES results that revealed fractions of negative buoyant air in the updrafts (Taylor and Baker, 1991; Siebesma and Cuijpers, 1995). The so-called fractional mixing rate $\varepsilon_0$ is defined as inversely proportional to the top of the cumulus layer $H$. In their unified scheme, Hohenegger and Bretherton (2011) applied the buoyancy sorting idea to compute entrainment and detrainment rates as in Bretherton et al. (2004) defining $\varepsilon_0$ in a different way. Taking into account LES simulations performed with the System for Atmospheric Modelling (SAM), this value is here link to the convective precipitation at cloud base (see Table 6).

Based on the results obtained from using tracers in LES simulations of shallow convection during BOMEX, that pointed to a description of entrainment through a stochastic Poisson process, Romps and Kuang (2010b) developed a parcel model with stochastic entrainment similar to the one proposed in Romps and Kuang (2010a). The authors used a Monte Carlo method to model entrainment rate. The parameterization uses two probability functions characterized by two parameters, i.e., the mean ratio of the entrained mass $m_{ent}$, and the distance that parcel travels between entrainment events $d_{ent}$. The mean fractional entrainment per distance is given by the ratio of these two parameters. The values that best fit to the CRM results were $d_{ent} = 226\text{ mm}$ and $m_{ent} = 0.91$, i.e., $\varepsilon = 4.0 \cdot 10^{-3}\text{ m}^{-1}$. Nie and Kuang (2012) specified $m_{ent} = 0.32$ and $d_{ent} = 125\text{ m}$ for their LES simulations of BOMEX to reduce the number of undilute updrafts to a number comparable to their 25-m resolution run. For the sub-cloud layer, the parameters were set to $d_{ent} = 30\text{ m}$ and $m_{ent} = 0.06$. Sušelj et al. (2013) replaced the entrainment parameterization in Sušelj et al. (2012) by a stochastic formulation. The authors considered a constant entrainment rate for dry updrafts below the condensation level, and an entrainment formulation similar to the one proposed by Romps and Kuang (2010b). In this case, the authors found a typical distance of 100 m between entrainment events for BOMEX phase-3 experiment. Sušelj et al. (2014) parameterized the entrainment rate as in Sušelj et al. (2013) although with different values for the constant entrainment rate and $d_{ent}$.

Recently, in their shallow cumulus study, Lu et al. (2018) identified deficiencies in the previous studies about the impact of RH on entrainment that could lead to erroneous conclusions regarding the effects of RH on entrainment, such as the use of conserved quantities related to RH to estimate entrainment rates, or that no observations had thus far been used to determine the relationship between RH and entrainment. To address these deficiencies, the authors analyzed aircraft observations from the Routine AAF (ARM Aerial Facility) CLOWD (Clouds with Low Optical Water Depths) Optical Radiative Observations (RACORO) (Vogelmann et al., 2012) and Rain In Cumulus over the Ocean (RICO) field campaigns (Rauber et al., 2007) for shallow cumulus and concluded that $\varepsilon$ and RH are positively correlated. Nonetheless, there is no general consensus on the effects of environmental RH on entrainment rates (Lu et al., 2018).

**Table 6:** A sample of empirical values and assumptions used in the parameterization of entrainment in the updraft. (Note: subscript *sh* refers to shallow convection)

| Type | Empirical value or assumption | Choices in the literature | Reference |
|---|---|---|---|
| Turbulent | Constant | $\varepsilon_{turb}^{u} = 1 \cdot 10^{-4}$ m$^{-1}$ for penetrative (only occurs in the lower part of the cloud layer) and midlevel convection, and $\varepsilon_{turb,sh}^{u} = 3 \cdot 10^{-4}$ m$^{-1}$ | Tiedtke (1989); Nordeng (1994); Zhang et al. (2011); Möbis and Stevens (2012) |
| | | $\varepsilon_{turb}^{u} = 3 \cdot 10^{-4}$ m$^{-1}$ | Wang et al. (2007) |
| | Inversely proportional to height $z$ | $\varepsilon_{turb}^{u} = C_t^u / z$, with $C_t^u = 3\, A_e\, f(p)$, where $A_e = 1.5$ for all levels above LCL, and $f(p) = p / p_s^2$, with $p_s$ the surface pressure | Gregory and Rowntree (1990) |
| | | $C_t^u = 0.55 + 8.0 \left(1.2 - \frac{z_{LCL}}{100}\right)^2$, with $0.55 \leq C_t^u \leq 3.5$ | Stratton and Stirling (2012) only for deep convection over land |
| | | $\varepsilon_{turb}^{u} = \frac{1}{z} \cdot \left[\frac{A \cdot RH}{z_{LCL}}\right]$, where $z_{LCL}$ is the height of the LCL and A=2.0 | Stirling and Stratton 2012) only for deep convection over land |
| | | $\varepsilon_{uni}^{u} = F(z)\, f_{dp}\, 3\, A_e \rho g\, f(p)$, where F(z) is a scaling factor in the range 0.5 to 2.5, and $f_{dp}$ is a tuning parameter set to 1.13 (deep) and 1.0 (shallow) | Willet and Whitall (2017) |
| | Proportional to the environmental humidity $\bar{q}$ | $\varepsilon_{turb}^{u} = c_0 F_{\varepsilon,0}$, where $F_{\varepsilon,0} = \left(\frac{\overline{q_s}}{\overline{q_{s,b}}}\right)^2$ and $\overline{q_s}$ and $\overline{q_{s,b}}$ are the saturation specific humidity at the parcel level and cloud base, respectively | Bechtold et al. (2008); Han and Pan (2011); Zhang and Song (2016) Del Genio and Wu (2010) found $c_0 = 0.5$ |
| Dynamical | Proportional to moisture convergence | | Tiedtke (1989); Möbis and Stevens (2012) |
| | Depends on momentum convergence | $\varepsilon_{dyn}^{u} = \frac{1}{2} \frac{B}{w_{d,LFS}^2 - \int_z^{LFS} B\, dz} + \frac{1}{\rho} \frac{d\rho}{dz}$, where $w_{d,LFS} = 1$ m s$^{-1}$ is the downdraft velocity at LFS | Nordeng (1994); Möbis and Stevens (2012) |
| | Proportional to the environmental humidity $\bar{q}$ | $\varepsilon_{dyn}^{u} = c_1 \frac{\overline{q_s} - \bar{q}}{\bar{q}} F_{\varepsilon,1}$, where $F_{\varepsilon,1} = \left(\frac{\overline{q_s}}{\overline{q_{s,b}}}\right)^3$, $c_1$ is a tunable parameter, and $\overline{q_s}$ and $\overline{q_{s,b}}$ are the saturation specific humidity at the parcel level and cloud base, respectively | Bechtold et al. (2008); Del Genio and Wu (2010) found $c_1 = 0.1$ |
| | | $\varepsilon_{dyn}^{u} = d_1 (1 - RH) F_{\varepsilon,1}$ where $d_1$ is a tunable parameter | Han and Pan (2011) |
| | | $\varepsilon_{dyn}^{u} = C_e (1.3 - RH) F_{\varepsilon,1}$, where $C_e = 1.8 \cdot 10^{-3} m^{-1}$, and $\varepsilon_{sh}^{u} = 2 \cdot \varepsilon_{dyn}^{u}$ | Bechtold et al. (2014) |
| | Occurs when cloud parcels accelerate upward and the buoyancy B is positive | | Zhang et al. (2011) |
| No distinction | Inversely proportional to cloud radius $R$ | $\varepsilon^{u} = C_e^u / R$, with $C_e^u = 1$ | Malkus (1959) |
| | | $C_e^u = 0.2$ (T62, ST62), 0.18 (SW69) | Turner (1962); Squires and Turner (1962); Simpson and Wiggert (1969); Arakawa and Schubert (1974); Wagner and Graf (2010) |
| | Function of a critical mixing fraction $\chi_c$ | $\chi < \chi_c$ | Kain and Fritsch (1990); Bechtold et al. (2001); Pergaud et al. (2009) |
| | Proportional to a critical mixing function $\chi_c$ | $\varepsilon^{u} \geq M_u \frac{C_e^u \delta p}{R} \chi_c$, where $M_u$ is the updraft mass flux at cloud base, $C_e^u = 0.03$ m Pa$^{-1}$, and $\chi_c = 0.5$ | Kain (2004) |
| | Does not exist around cloud edges | | Grell et al. (1994) |
| | Defined by the requirement that the temperature of the plume that detrains at a certain level $z$ equals $T_{env}$ | Reaches its maximum value at the height of minimum $h$ for a saturated state | Zhang and McFarlane (1995) |
| | Inversely proportional to height $z$ | $\varepsilon = \frac{C_{e,sh}}{z}$ with $C_{e,sh} = 1.0$ | Siebesma and Cuijpers (1995); Siebesma et al. (2003); De Rooy and Siebesma (2008) |

| Type | Empirical value or assumption | Choices in the literature | Reference |
|---|---|---|---|
| | Set to a constant value | $\varepsilon_{sh}^u = 2 \cdot 10^{-3} \, m^{-1}$ | Siebesma (1998); Soares et al. (2004) |
| | | $\varepsilon_{sh}^u = 1.2 \cdot 10^{-3} \, m^{-1}$ at cloud base and $\varepsilon_{sh}^u = 3 \cdot 10^{-3} \, m^{-1}$ 150 hPa above it | Gregory et al. (2000) |
| | | Below condensation level $\varepsilon_{uni}^u = 2.5 \cdot 10^{-3} \, m^{-1}$ (S13), $8.5 \cdot 10^{-4} \, m^{-1}$ (S14) | Sušelj et al. (2013); Sušelj et al. (2014) |
| | | $\varepsilon^u = 2.5 \cdot 10^{-4} \, m^{-1}$ | Song and Zhang (2017) |
| | | $\varepsilon_{sh}^u = 2 \cdot 10^{-3} \, m^{-1}$ | Siebesma (1998); Siebesma et al. (2003); Soares et al. (2004) |
| | Proportional to the fraction of TKE available for entrainment $A_\varepsilon$ | $\varepsilon_{sh}^u = A_\varepsilon \frac{w^*}{m_b} \frac{1}{CD}$, where $w^*$ is the convective velocity-scale, $m_b$ cloud base mass flux, CD is the cloud depth and $A_\varepsilon = 0.03$ for the core (GB99), 0.06 (KG12) | Grant and Brown (1999); Grant and Lock (20049; Kirshbaum and Grant (2012) |
| | | $\varepsilon_{sh}^u = A_\varepsilon \frac{CAPE^{1/3}}{m_b^{2/3}} \frac{1}{CD}$, where $A_\varepsilon = 0.037 - 0.035$ | Drueke et al. (2019) |
| | Function of the buoyancy of the parcel $B$ and the in-cloud updraft velocity, $w$ | $\varepsilon^u = C_e^u \frac{a_w B}{w^2}$, where $C_e^u = 0.25$ (deep G01), 0.5 (shallow G01) and $a_w = 1/6$ | Gregory (2001), Kim et al. (2013) |
| | | $C_e^u = 0.6$ | Chikira and Sugiyama (2010) |
| | | $C_e^u = 0.3$ | Del Genio et al. (2012) |
| | | $C_e^u = (\frac{1}{RH} - 1)$ | Kim and Kang (2012) |
| | | $C_e^u = 0.52$ | Hirota et al. (2014) |
| | | $\varepsilon_{sh}^u = C_{e,sh}^u \frac{B}{w^2}$, $C_{e,s}^u = 0.55$ (sub-cloud layer) | Pergaud et al. (2009) |
| | Function of the in-cloud vertical velocity $w$ and a turnover timescale $\tau_t$ | $\varepsilon_{sh}^u = \frac{\eta}{\tau_t} \frac{1}{w}$, with $\tau_t = 300$ s and $\eta = 0.9$ for BOMEX and 1.2 for SCMs (N02) $\tau_t = 400$ s and $\eta = 1$ (N09) $\tau_t = 500$ s and $\eta = 1$ (S12) $\tau_{t,sh} = 320$ s and $\eta = 1$ (S16) | Neggers et al. (2002, 2009); Sušelj et al. (2012); Sakradzija et al. (2016) |
| | | $\eta/\tau_t = 2.4 \cdot 10^{-3} \, s^{-1}$ | Chikira and Sugiyama (2010) |
| | Inversely proportional to height $z$ | $\varepsilon^u = C_e^u/z$, where $C_e^u = 0.55$ (JS03), 0.1 (HP11) | Jakob and Siebesma (2003); Han and Pan (2011) (only in sub-cloud layers) |
| | | $\varepsilon_{sh}^u = C_{e,sh}^u/z$, where $C_{e,sh}^u = 1.0$ (RS08), 0.3 (HP11) | De Rooy and Siebesma (2008); Han and Pan (2011) |
| | | (in sub-cloud layer) $\varepsilon_{sh}^u = C_{e,sh}^u \left( \frac{1}{z+\Delta z} + \frac{1}{(z_i - z)+\Delta z} \right)$, where $\Delta z$ is the vertical grid spacing and $C_{e,sh}^u = 0.5$ (S04), 0.4 (S07) | Soares et al. (2004); Siebesma et al. (2007) |
| | Depends on a critical eddy-mixing distance $d_c$ and a critical mixing fraction $\chi_c$ | $\varepsilon_{sh}^u = \varepsilon_0 \chi_c^2$, where $\varepsilon_0 = \frac{15}{d_c}$ (B04) $\varepsilon_0(z) = \varepsilon_0(z_{cb})(z/z_{cb})^{c_e}$ (HB11), where $z_{cb}$ is cloud-base height, and $c_e$ is computing by specifying $\varepsilon_0$ at cloud base and at $z_{cb} + 2000$ m | Bretherton et al. (2004); Hohenegger and Bretherton (2011) |
| | Inversely proportional to height $z$ | $\varepsilon_{sh}^u = \frac{C_{e,sh}}{z}$ with $C_{e,sh} = 1.0$ | De Rooy and Siebesma (2008) |
| | Proportional to detrainment rate $\delta_{sh}^u$ in the sub-cloud layer | $\varepsilon_{sh}^u = 0.4 \, \delta_{sh}^u$ | Rio and Hourdin (2008) |
| | Function of the buoyancy $B$ and the in-cloud vertical velocity $w$ | $\varepsilon^u = max\left[0, \frac{1}{1+\beta_1}\left(\frac{a\beta_1 B}{w^2} - b'\right)\right]$, where $a\beta_1(1+\beta_1)^{-1} = 0.315$, $a = 2/3$ and $b' = 0.002$ | Rio et al. (2010) |
| | Stochastic parameterization. Depends on mean ration of entrained mass $m_{ent}$ and distance that parcel travels between entrainment events $d_{ent}$ | $\varepsilon_{sh}^u = m_{ent}/d_{ent}$, where $d_{ent} = 226$ m (RK10), 125 m (NK12), 30 mm NK12-sub-cloud layer), 100 m (S13), 200 m (S14) $m_{ent} = 0.91$ (RK10), 0.32 (NK12), 0.06 (NK12-sub-cloud layer), 0.1 (S13), 0.2 (S14) | Romps and Kuang (2010); Nie and Kuang (2012); Sušelj et al. (2013, 2014) |

| Type | Empirical value or assumption | Choices in the literature | Reference |
|---|---|---|---|
| | Depends on a prognostic variable | | Mapes and Neale (2011) |
| | Depends on RH and the height of the LCL $z_{LCL}$ for the early stages of developing convection over land | | Stirling and Stratton (2012) |
| | Depends on the PBL depth and the height $z$. Sets a maximum value for $\varepsilon^u$ | $\varepsilon^u = \mu/\min(z, z_{PBL})$ with $\mu = 0.185$ as default value and $\varepsilon^u_{max} = 1 \cdot 10^{-4}$ m$^{-1}$. The value of $\mu$ is modified within the paper ($\mu \times 2$, $\mu \times 5$, $\mu/2$) | Oueslati and Bellon (2013) |
| | Function of the pressure $p$ | $\varepsilon^u = 4.5\, F \frac{p(z)\rho)g(z)}{p_s^2}$ with $F = 0.9$ as a default value and $p_s$ the surface pressure | Klingaman and Woolnough (2014) |
| | Uses PDFs | Lognormal, gamma and Weibull distributions | Guo et al. (2015) |
| | The entrained mass depends on the pressure depth of a model layer $\Delta p$, horizontal grid spacing $Dx$, and the height of LCL above the ground $z_{LCL}$ | $\Delta M_e = M_b \frac{\alpha\beta}{z_{LCL}}\Delta p$, where $M_b$ is the updraft mass flux at cloud base, $\alpha = 0.03$, and $\beta = [1 + \ln(25/Dx)]$ | Zheng et al. (2016) |
| | Values using retrieval methods | $\varepsilon^u_{sh} = 0.5\ km^{-1}$ over land | Drueke et al. (2019) |
| | | $\varepsilon^u_{sh} = 0.5 - 0.6$ km$^{-1}$ ($1.0 - 1.1$ km$^{-1}$) over land(ocean) | Kirshbaum and Lamer (2021) |
| | Function of buoyancy $B$ and detrainment rate $\delta^u$ | $\varepsilon^u w^2 = C_1 B - C_2 \delta^u w^2$ with $C_1 = C_2 \approx 0.2$ | Baba (2019) |

**Table 7:** A sample of empirical values and assumptions used in the parameterization of entrainment in the downdraft.

| Type | Empirical value or assumption | Choices in the literature | Reference |
|---|---|---|---|
| Turbulent | Set to a constant value | $\varepsilon^d_{turb} = 2 \cdot 10^{-4}$ m$^{-1}$ | Tiedtke (1989); Nordeng (1994); Möbis and Stevens (2012); Baba (2019) |
| | | $\varepsilon^d_{turb} = 3 \cdot 10^{-4}$ m$^{-1}$ | Bechtold et al. (2014) |
| Dynamical | Function of in-cloud buoyancy $B$ and downdraft velocity at the LFS $w_{d,LFS}$ | $\varepsilon^d_{dyn} = \frac{-B}{w^2_{d,LFS} - \int_z^{LFS} B\, dz} + \frac{1}{\rho}\frac{d\rho}{dz}$ , where $w_{d,LFS} = 1$ m s$^{-1}$ is the downdraft velocity at the LFS | Baba (2019) |
| | Function of in-cloud buoyancy $B$ | | Bechtold et al. (2014) |
| No distinction | Set to a constant value | $\varepsilon^d = 2 \cdot 10^{-4}$ m$^{-1}$ (K13) | Gerard and Geleyn (2005); Gerard (2007); Kim et al. (2013) |
| | Proportional to $\varepsilon^u$. Its maximum value $\varepsilon^d_{max}$ is constrained | $\varepsilon^d = 2\,\varepsilon^u$ and $\varepsilon^d_{max} = 2/(z_D - z_b)$ where $z_D$ is height of the detrainment level, and $z_b$ is the cloud base height | Zhang and McFarlane |

Less attention has been paid to the parameterizations of the detrainment process. Many convection schemes set it as a constant value (see Tables 8 and 9), while others consider detrainment to be negligible (Lu et al., 2012). Tiedtke (1989) and Nordeng (1994) assumed a turbulent detrainment inversely proportional to cloud radii and fixed its value to $\delta_{turb} = 1 \cdot 10^{-4}$ m$^{-1}$ for penetrative and midlevel convection (see Table 8). On the other hand, Gregory and Rowntree (1990) assumed a turbulent

detrainment rate inversely proportional to the height and smaller than $\varepsilon_{turb}$, while Bechtold et al. (2008) set $\delta_{turb}$ to a constant value. Dynamical detrainment $\delta_{dyn}$ is defined to occur in Tiedtke (1989), Bechtold et al. (2008) and Gregory and Rowntree (1990) when the updraught buoyancy becomes negative. In the former two schemes it is then set proportional to the decrease in updraught kinetic energy while in the latter it is computed implicitly. For downdraft, Bechtold et al. (2014) set $\delta_{turb} = \varepsilon_{turb}$,

and enforced $\delta_{dyn}$ over the lowest 50 hPa. As in the case of entrainment rates in downdrafts, a common practice in the
105    definition of detrainment rates for downdraft consists in assuming a similar parameterization as for updrafts (Table 9).

**Table 8:** A sample of empirical values and assumptions used in the parameterization of detrainment in the updraft. (Note: subscript *sh* refers to shallow convection)

| Type | Empirical value or assumption | Choices in the literature | Reference |
|---|---|---|---|
| Turbulent | Constant | $\delta^u_{turb} = 1 \cdot 10^{-4}\ \text{m}^{-1}$ | Tiedtke (1989); Nordeng (1994); Bechtold et al. (2008); Zhang et al. (2011) |
| | | $\delta^u_{turb,sh} = 3 \cdot 10^{-4}\ \text{m}^{-1}$ | Tiedtke (1989) |
| | Dependent on RH | $C^u_{dt} = C^u_{dt}(1.6 - RH)$ , where $\quad C^u_{dt} = 0.75 \cdot 10^{-4} m^{-1}$ | Bechtold et al. (2014) |
| | Proportional to the entrainment rate $\varepsilon^u_{turb}$ | $\delta^u_{turb} = C^u_{dt} \cdot \varepsilon^u_{turb}$ where $C^u_{dt} = 2/3$ | Gregory and Rowntree (1990) |
| | | $C^u_{dt} = (1 - RH)$ | Derbyshire et al. (2011); Walters et al. (2019) |
| | | $C^u_{dt} = 15(1 - RH)^2$ | Stirling and Stratton (2012) |
| | | $C^u_{dt} = 2.5(1 - RH)$ | Stratton and Stirling (2012) |
| | | $\delta^u_{turb,sh} = \varepsilon^u_{turb,sh}$ where $C^u_{dt,sh} = (1.6 - RH)$ | Bechtold et al. (2014) |
| Dynamical | Initiated if the buoyancy of the parcel is less than a minimum value, $B_{min}$ | $B_{min} = 2 - 3\ \text{K}$ | Yanai et al. (1973) |
| | | $B_{min} = 0.2\ \text{K}$ | Gregory and Rowntree (1990) |
| | Only at levels of neutral buoyancy | | Tiedtke (1989) |
| | Non-zero above the lowest possible organized detrainment level $z_{low}$ | $\delta^u_{dyn} = \frac{1}{\sigma}\frac{d\sigma}{dz}$ , where $\sigma = \sigma_0 \cos\left(\frac{\pi}{2}\frac{z-z_{low}}{z_{ct}-z_{low}}\right)$ with $z_{ct}$ the cloud top height, and $\sigma$ the horizontal area covered by the updraft. $z_{low}$ is the level of neutral buoyancy with entrainment rate $\varepsilon = \frac{1}{2(\zeta+z-z_{cb})}$, where the subscript cb means cloud base, and $\zeta = 25\ \text{m}$ corresponds to an excess buoyancy of 1 K at cloud base and a vertical velocity of 1 m s$^{-1}$ at that level. | Nordeng (1994), |
| | Proportional to the decrease in updraft vertical kinetic energy at the top of the cloud | | Bechtold et al. (2008); Zhang and Song (2016) |
| | Proportional to the loss of buoyancy | | Derbyshire et al. (2011) |
| | When updraft becomes negatively buoyant | | Bechtold et al. (2014) |
| No distinction | Occurs only in a thin layer at cloud top | | Arakawa and Schubert (1974) |
| | Only at levels of neutral buoyancy | | Emanuel (1991); Moorthi and Suarez (1992) |
| | Does not exist around cloud edges | | Grell et al. (1994) |
| | Constant | $\delta^u = 2 \cdot 10^{-4}\ \text{m}^{-1}$ (deep) and $\delta^u_{sh} = 2 \cdot 10^{-3}\ \text{m}^{-1}$ (shallow) | Gregory (2001) |
| | | $\delta^u_{sh} = 3 \cdot 10^{-3} m^{-1}$ | Soares et al. (2004) |
| | Depends on a critical eddy-mixing distance $d_c$ and a critical mixing fraction $\chi_c$ | $\delta^u_{sh} = \frac{C^u_d}{d_c}(1 - \chi_c)^2$, where $C^u_d = 1.5$ | Bretherton et al. (2004); Zhao et al. (2018) |
| | Function of average of $\chi_c$ from cloud base up to the middle of the cloud layer $\langle\chi_c\rangle_*$ | $\delta^u_{sh} \propto \langle\chi_c\rangle_*$ | De Rooy and Siebesma (2008) |

| Type | Empirical value or assumption | Choices in the literature | Reference |
|---|---|---|---|
| | Depends on in-cloud vertical velocity $w$, buoyancy $B$ and the difference in the water mixing ratio ($\Delta q$) between the mean plume ($q_l$) and the environment ($q$) | $\delta^u = max\left[0, -\frac{a_1\beta_1}{1+\beta_1}\frac{B}{w^2} + c\left(\frac{\Delta q}{q}\right)^d\right]$, where $a_1 = 2/3$, $\beta_1 = 0.9$, $c = 0.012$ s$^{-1}$ and $d = 0.5$ | Rio et al. (2010) |
| | Constant at all levels | $\delta^u = \varepsilon_{cb}$ , and $\delta^u_{sh} = \varepsilon_{cb,sh}$ with $\varepsilon_{cb(sh)}$ the entrainment at cloud base for deep(shallow) | Han and Pan (2011) |
| | Function of buoyancy $B$ and in-cloud vertical velocity $w$ | $\delta^u = -C^u_d\frac{aB}{w^2}$ where $C^u_d$ takes different values | Kim et al. (2013) |
| | Function of buoyancy $B$ | $\delta^u = B/2$ | Baba (2019) |

**Table 9:** A sample of empirical values and assumptions used in the parameterization of detrainment in the downdraft.

| Type | Empirical value or assumption | Choices in the literature | Reference |
|---|---|---|---|
| Turbulent | Set to a constant value | $\delta^d_{turb} = 2 \cdot 10^{-4}$ m$^{-1}$ | Tiedtke (1989): Nordeng (1994); Baba (2019) neglects it when the downdraft is thermodynamically positive buoyant or reaches below the cloud base |
| | | $\delta^d_{turb} = 3 \cdot 10^{-4}$ m$^{-1}$ | Bechtold et al. (2014) |
| Dynamical | Enforced over the lowest 50 hPa | | Bechtold et al. (2014) |
| | When the downdraft is thermodynamically positive buoyant or reaches below the cloud base | $\delta^d_{dyn}$ inversely proportional to layer thickness (if in-cloud) or to height (if below cloud base) | Baba (2019) |
| No distinction | Set to a constant value that is replaced when vertical velocity decreases with height, usually near cloud top | $\delta^d = 2 \cdot 10^{-4}$ m$^{-1}$ | Gregory (2001) |
| | Only at levels of neutral buoyancy | | Emanuel (1991) |
| | Only over a fixed layer of 60 hPa that extends from downdraft detrainment level to downdraft base layer | $\delta^d = 0$ m$^{-1}$ apart from the detrainment layer | Bechtold et al. (2001) |
| | Linear function of pressure between the top of USL and the base of the downdraft | | Kain (2004) |
| | Proportional to the updraft convergence of the updraft mass flux | | Gerard and Geleyn (2005) |
| | When downdraft becomes positively buoyant, with 75% of its mass detraining at each subsequent | | Kim et al. (2013) |
| | Only in the lowest 1000 m above the ground or starting at LFC, whichever is located higher above the ground | | Grell and Freitas (2014) |

In the parameterization of detrainment in shallow convection schemes, De Rooy and Siebesma (2008) treated the mass flux and the entrainment formulation separately based on LES results, that suggest that variations in the mass flux profile are mostly related to the fractional detrainment (Jonker et al., 2006; De Rooy and Siebesma, 2008). De Rooy and Siebesma (2008) kept $\varepsilon$ fixed as an inverse function of height, and developed a dynamical formulation for $\delta$ dependent on the average of $\chi_c$ from cloud base up to the middle of the cloud layer $\langle\chi_c\rangle_*$(the reader is referred to equation A11 in De Rooy and Siebesma (2008) for a detailed calculation of $\chi_c$), and on the cloud layer depth. For shallow convection, Siebesma and Cuijpers (1995) found vales of detrainment rates that were rather constant showing around $3 \cdot 10^{-3}$ m$^{-1}$ for the core and $4 \cdot 10^{-3}$ m$^{-1}$ for the

updraft. Using LES output from BOMEX, Siebesma (1998) found typical values of detrainment in the range

$2.5 - 3 \cdot 10^{-3}$ m$^{-1}$. Other studies, such as Soares et al. (2004) used a constant detrainment rate following Siebesma (1998), set it to the value of entrainment at cloud base (e.g., Han and Pan, 2011), or proportional to the entrainment rates (e.g., Bechtold et al., 2014), among others.

### 4.2.3 Impact of entrainment and detrainment on convective models

The discussion above illustrates the many nuances in the modeling of convection, the importance of empirical values in the

final results and the need to further research to disentangle the many details involved. It is accepted that the parameterizations of entrainment and detrainment still have great uncertainties (e.g., Romps, 2010; Becker and Hohenegger, 2018) and problems in producing a realistic representation of convection (e.g., Mapes and Neale, 2011). For example, Stratton and Stirling (2012) improved the timing and amplitude of the diurnal cycle of tropical convection in the Met Office climate model by setting the entrainment for deep convection as a function of the height of LCL.

Perhaps not surprisingly, MJO simulations are also sensitive to entrainment (e.g., Hannah and Maloney, 2011; Del Genio et al., 2012; Kim et al., 2012; Hirons et al., 2013; Klingaman and Woolnough, 2014). Hannah and Maloney (2011) applied the RAS scheme in a GCM and analyzed the influence of minimum entrainment rate and rain evaporation fraction in the simulation of MJO. Larger values of any of the two parameters led to a better representation of the MJO and interseasonal variability, although higher values of minimum entrainment produced a drier and cooler atmosphere in contrast to the effect of higher

values of rain precipitation fraction. Klingaman and Woolnough (2014) evaluated the effects of 22 model configurations and subgrid parameterizations on the simulation of MJO in the Hadley Centre Global Environmental model Global Atmosphere version 2 (HadGEM3 GA2.0) and tested the changes in 14 hindcast cases. A better representation of the MJO for both hindcast and climate simulations was achieved by increasing entrainment and detrainment rates for mid-level and deep convection. A better representation of MJO was also achieved by Kim et al. (2012) using a GCM to evaluate the tropical subseasonal

variability. However, this improvement was at the expense of an increased bias in the mean state, typical for other GCMs with stronger MJO (Kim et al., 2011).

The entrainment parameterization proposed by Gregory (2001) for both deep and shallow convection achieved satisfactory results in various analyses (e.g., Chikira and Sugiyama, 2010; Del Genio and Wu, 2010) but proved to be cloud- and altitude-dependent. Recently, Baba (2019) modified Gregory's parameterization of the entrainment rate by relating it to the detrainment

rate and *B*. This new parameterization led to improvements in the positive bias of precipitation in western Pacific region, in the positive bias of outgoing shortwave radiation over the ocean as well as in the simulation of MJO, equatorial waves, and precipitation over the western Pacific region. Using an RCM over the Maritime Continent region, Wang et al. (2007) demonstrated that changes in the values of the fractional entrainment/detrainment rates in Tiedtke scheme, including both shallow and deep convection, affect the simulation of the tropical precipitation diurnal cycle. Over land, Del Genio and Wu

(2010) used a CRM to study the transition from shallow to deep convection in diurnal cycles and inferred entrainment rates. Subsequently, the authors compared results from three different entrainment parameterizations to the results obtained with

CRM and concluded that the best results were achieved by the entrainment parameterization of Gregory (2001). Through a version of the Goddard Institute for Space Studies Global Climate Model (GISS GCM) with the entrainment rate proposed by Gregory (2001), Del Genio et al. (2012) efficiently reproduced the MJO transition from shallow to deep convection.

The advantage of the formulation of entrainment and detrainment rates in the unified scheme of Hohenegger and Bretherton (2011) is that it does not require an explicit distinction between deep and shallow convection. This formulation linking the fractional mixing rate $\varepsilon_0$ to the convective precipitation at cloud base improved the simulation of the precipitation diurnal cycle compared to CAM, as well as relative humidity, cloud cover and mass flux profiles, and could realistically simulate the transition between shallow and deep convection. Willet and Whitall (2017) also achieved a more realistic representation of the

diurnal cycle in the tropics with this fractional mixing rate in their parameterization of entrainment in the UK MetOffice model. Other studies have evaluated the impact of entrainment/detrainment formulation on large-scale features, such as the double ITCZ (e.g., Chikira, 2010; Chikira and Sugiyama, 2010; Möbis and Stevens, 2012; Oueslati and Bellon, 2013). Möbis and Stevens (2012) used both the Tiedtke and Nordeng schemes in an aquaplanet GCM to evaluate the sensitivity of ITCZ to the choice of the convective parameterization. The Tiedtke scheme produced a double ITCZ, while the Nordeng scheme, with a

higher lateral entrainment rate, led to a single ITCZ. In the works by Chikira (2010) and Chikira and Sugiyama (2010), the entrainment rate from AS was replaced by a formulation that depends on the surrounding environment following Gregory (2001) and Neggers et al. (2002). With this new formulation, variability and climatology improved, including the double ITCZ and the South Pacific Convergence Zone (SPCZ). Oueslati and Bellon (2013) obtained similar improvements in their study of the effects of entrainment on ITCZ by increasing entrainment in a hierarchy of models (coupled ocean–atmosphere GCM,

atmospheric GCM, and aquaplanet GCM), at the cost of an overestimation of precipitation in the center of convergence zones. The role of entrainment on large-scale features was also underlined by Hirota et al. (2014) in their comparison of four atmospheric models with different entrainment formulations over tropical oceans.

Based on Zhang (2002) and using sounding data from the Coupled Ocean-Atmosphere Response Experiment (COARE), the

South Pacific Convergence Zone (SGP97) and the Tropical Warm Pool – International Cloud Experiment (TWP-ICE), Zhang (2009) concluded that the entrainment of environmental air also affects CAPE and closure assumptions in CPs. The drier the entrained air, the stronger is the dilution effect that acts to reduce CAPE. Moreover, dilute CAPE shows a better correlation with the consumption of CAPE than undilute CAPE.

As for the impact of entrainment and detrainment formulations for shallow convection, Siebesma and Holtslag (1996)

evaluated a mass flux shallow cumulus based on BOMEX results and found that lateral entrainment and detrainment rates were one order of magnitude larger than those used in Tiedtke scheme. Neggers et al. (2002) evaluated their multiparcel model with LES results based on BOMEX and Small Cumulus Microphysics Study (SCMS). The model reproduced the features of the buoyant part of the clouds and the variability of temperature, moisture and velocity observed in cumulus clouds. Romps and Kuang (2010) found that their stochastic formulation of entrainment reproduces well the variability observed in the CRM

even when the cloud base variability is turned off. While the convective updrafts simulated with the approach proposed by

Sušelj et al. (2012) did not reach high enough compared to LES results and observations, the stochastic entrainment formulation described in Sušelj et al. (2013) properly simulated shallow cumulus, including the height of the updrafts and their reduction of horizontal area with height.

As mentioned in Sect. 4.2.2, less attention has been paid to the parameterizations of the detrainment process. Based on LES results for shallow convection, De Rooy and Siebesma (2008) proposed a new detrainment parameterization that led to improvements for ARM, BOMEX, and RICO shallow convection cases compared to the standard parameterizations of entrainment and detrainment (Siebesma and Cuijpers, 1995; Siebesma et al., 2003). Moreover, the authors revealed a greater variation in the detrainment rates from hour to hour and case to case than the variation in the entrainment rates. Derbyshire et al. (2011) confirmed this finding using a CRM and an adaptive detrainment proportional to the environmental relative humidity. Later, De Rooy and Siebesma (2010) showed that detrainment strongly influences the vertical structure of the mass flux.

## 4.3 Microphysics in convective clouds

The representation of microphysical processes in cumulus parameterizations is key to simulations of climate change (e.g., Ramanathan and Collins, 1991; Rennó et al., 1994; Lindzen et al., 2001). Convective microphysics greatly affects the representation of convective clouds due to its influence on detrainment of water vapor and hydrometeors, and the interaction between clouds and aerosols (e.g., Khain et al., 2005; Koren et al., 2005; Rosenfeld et al., 2008; Song and Zhang, 2011; Song et al., 2012; Tao et al., 2012). However, many convective parameterization schemes treat microphysical processes crudely, specifying an empirically determined conversion rate from cloud water to rainwater (e.g., Arakawa and Schubert, 1974; Tiedtke, 1989; Zhang and McFarlane, 1995; Han and Pan, 2011) or a certain precipitation efficiency, defined as the fraction of condensed cloud water converted to precipitation (Emanuel, 1991). The reader should keep in mind that other authors also take into account the effect of precipitation evaporation and thus, precipitation efficiency is defined as the fraction of condensate that reaches the surface (see Table 10). This is used in the calculations of the initial downdraft mass flux like in

**Table 10:** A sample of empirical values and assumptions used in precipitation efficiency accounting for evaporation.

| Empirical value or assumption | Choices in the literature | Reference |
|---|---|---|
| Function of the wind shear $\Delta V$ and cloud depth $CD$ | $PE_{ws} = 1.591 - 0.639 \frac{\Delta V}{CD} + 0.0953 \left(\frac{\Delta V}{CD}\right)^2 - 0.00496 \left(\frac{\Delta V}{CD}\right)^3$ | Fritsch and Chappell (1980) set $PE = 0.9$ if $\frac{\Delta V}{CD} < 1.35$ |
| Function of wind shear $\Delta V$ (similar as in FC80) and cloud base height $z_{LCL}$ | $PE = f(PE_{ws}, PE_{LCL})$ $PE_{LCL} = \frac{1}{1+PE_z}$ where $PE_z = 0.967 - 0.700 z_{LCL} + 0.162 z_{LCL}^2 - 1.257 \cdot 10^{-2} z_{LCL}^3$ | Zhang and Fritsch (1986); Kain and Fritsch (1990); Bechtold et al. (2001) |
| Function of wind shear $\Delta V$ and sub-cloud RH | | Grell (1993); Grell and Dévényi (2002) |
| Proportional to the total volume of condensed water accumulated over the cloud lifetime $M_V$ and droplet concentration $N_d$ | $PE \approx M_V^{0.9} N_d^{1.13}$ | Jiang et al. (2010); Grell and Freitas (2014) used CCN instead of $N_d$ |

Bechtold et al., (2001). A brief description of the main assumptions and empirical values used in the representation of microphysics in CPs is presented here for the sake of completeness. For a detailed review of microphysics parameterizations, the reader is referred to Zhang and Song (2016) for convection and Tapiador et al. (2019a) for a full account.

### 4.3.1 Conversion of cloud water to precipitation

Despite the importance of microphysical processes in the simulation of surface precipitation, radiation or cloud cover, only a
few convection schemes attempted to realistically represent these processes. A common approach is to assume that a specified fraction of the condensate is instantaneously removed as rain. In Yanai et al. (1973) and Tiedtke (1989), the conversion rate from cloud water to rainwater is assumed to be proportional to cloud water mixing ratio $l_w$ with an empirical function $K(z)$ conversion coefficient that depends on height, as shown in Table 11. Other assumptions include a constant conversion coefficient $C_c$ (Arakawa and Schubert, 1974; Grell, 1993; Zhang and McFarlane, 1995) or define a temperature-dependent
threshold water content $l_{wc}$, above which all cloud water is converted to precipitation (Emanuel and Živković-Rothman, 1999). Park and Bretherton (2009) modified the shallow cumulus parameterization described in Hack (1994) and used in the UW scheme based on the shallow convective parameterization of Bretherton et al. (2004). Among the modifications introduced, cloud condensate exceeding a certain threshold value of the cloud condensate mixing ratio is converted into precipitation, and includes the evaporation of convective precipitation above cloud base. In general, shallow convective schemes do not include
a parameterization of conversion to precipitation.

Few schemes with a more realistic treatment of the conversion of cloud water to rainwater can be found in the literature on convection. Autoconversion of cloud water in the convection scheme is considered in Sud and Walker (1999), following Sundqvist (1978), as well as in Zhang et al. (2005). The latter included the autoconversion of cloud water and other microphysical processes for both cloud water and ice in the Tiedtke scheme. However, neither the size nor the number
concentration of both hydrometeors is considered explicitly. This makes it impossible to account for aerosol-convection interaction, which is of great importance in climate simulations. To overcome this shortcoming, Song and Zhang (2011) and Song et al. (2012) added mass mixing ratio and number concentration of each hydrometeor in their parameterization. Another more realistic treatment of condensation is that proposed by Bony and Emanuel (2001). In this scheme, the condensed water produced at the subgrid scale is predicted by the convection scheme, while its spatial distribution is predicted by a statistical
cloud scheme through a probability distribution function of the total water. Indeed, the parameterization of the microphysics is more comprehensively devoted to this specific problem.

**Table 11:** A sample of empirical values and assumptions used in the conversion of cloud water to precipitation. (Note: subscript *sh* refers to shallow convection)

| Empirical value or assumption | Choices in the literature | Reference |
|---|---|---|
| Proportional to the liquid water content $l_w$ and an empirical function $K(z)$ that depends on height $z$ | $Pr = K(z)l_w$, where $$k(z) = \begin{cases} 0, & z \leq z_b + 1500 \text{ m} \\ 2 \cdot 10^{-3} \text{ m}^{-1}, & z > z_b + 1500 \text{ m} \end{cases} \text{(T89)}$$ | Yanai et al. (1973); Tiedtke (1989) |
| Constant conversion rate $C_c$ | | Arakawa and Schubert (1974) |
| | $Pr = C_c M_u l_{w'}$, where $C_c = 6 \cdot 10^{-3} \text{ m}^{-1}$ (W12), $M_u$ is the updraft mass flux, $l_w$ is the liquid water content and $\rho$ is the air density | Lord et al. (1982); Wu (2012) |
| | $C_c = 2 \cdot 10^{-3} \text{ m}^{-1}$ | Zhang and McFarlane (1995); Han and Pan (2011) |
| | $C_c = \begin{cases} a \cdot \exp\{b[T(z) - T_0]\}, & T \leq 0\,^{\circ}\text{C} \\ a, & T > 0\,^{\circ}\text{C} \end{cases}$, with $a = 2 \cdot 10^{-3} \text{ m}^{-1}$, and $b = 0.07\,^{\circ}\text{C}^{-1}$ | Han et al. (2016) |
| Function of a condensate to precipitation conversion factor $c_r$ and the in-cloud vertical velocity $w$ | $Pr \propto 1 - exp(-c_r \Delta z/w)$, with $c_r = 0.01 \text{ s}^{-1}$ (KF90) $c_r = 0.02 \text{ s}^{-1}$ (B00) | Kain and Fritsch (1990); Bechtold et al. (2001) |
| Varies linearly between 150 mb and 500 mb | $$Pr = \begin{cases} 0, & p_b - p_i < 150 \text{ hPa} \\ \frac{p_b - p_i - 150}{350} & 150 \text{ hPa} < p_b - p_i < 500 \text{ hPa} \\ 1, & p_b - p_i > 500 \text{ hPa} \end{cases}, \text{ where}$$ $p_b$ is the pressure at cloud base | Emanuel (1991) |
| Function of the detrainment pressure | $$Pr = \begin{cases} 1, & p < 500 \text{ hPa} \\ 0.8 + \dfrac{800 - p}{1500} & 500 \text{ hPa} < p < 800 \text{ hPa} \\ 0.58, & p > 800 \text{ hPa} \end{cases}$$ | Moorthi and Suarez (1992) |
| | $$Pr = \begin{cases} 0.975, & p < 500 \text{ hPa} \\ 0.500 + 0.475\dfrac{800 - p}{300} & 500 \text{ hPa} < p < 800 \text{ hPa} \\ 0.500, & p > 800 \text{ hPa} \end{cases}$$ | Anderson et al. (2004); Li et al. (2018) |
| Function of a threshold of the cloud water content $l_{wc}$ is converted to precipitation | $Pr = C_{eff}(l_w - lw_c)$ $$l_{wc} = \begin{cases} l_0, & T \geq 0\,^{\circ}\text{C} \\ l_0(1 - T/T_c), & T_c < T < 0\,^{\circ}\text{C} \\ 0, & T \leq T_c \end{cases},$$ where $l_0 = 1.1 \text{ g kg}^{-1}$ is a warm cloud autoconversion threshold, and $T_c = -55\,^{\circ}\text{C}$ | Emanuel and Živković-Rothman (1999) set $C_{eff} = 1$ ; Bony and Emanuel (2001) set $C_{eff} = 0.999$ |
| Precipitation of condensate above a threshold cloud condensate mixing ratio $q_{max,sh}$ | $q_{max,sh} = 1 \text{ g kg}^{-1}$ | Bretherton et al. (2004); Park and Bretherton (2009) |
| Function of the cloud water content $l_{wc}$, temperature and cloud droplet number concentration $CDNC$ | $Pr = l_w f(T, CDNC)$, where $f(T, CDNC)$ $$= \begin{cases} 1.0, & CDNC < 750\ cm^{-3} \text{ or } T < 263 \text{ K} \\ 0.25, & 750\ cm^{-3} < CDNC < 1000\ cm^{-3} \text{ or } T > 263 \text{ K} \\ 0.0, & CDNC > 1000\ cm^{-3} \text{ or } T > 263 \text{ K} \end{cases}$$ | Nober et al. (2003) |

### 4.3.2 Evaporation in downdrafts

Downdrafts are greatly affected by evaporation of hydrometeors and detrained cloud droplets due to latent cooling. Therefore, a realistic representation of this microphysical process is needed. However, only a limited number of convective parameterizations, such as Emanuel (1991), include an explicit calculation of this process, as shown in Table 12. Instead, crude assumptions can be found in the literature. The evaporation in downdraughts is often implicitly computed by assuming that the evaporation maintains a saturated or quasi-saturated downdraught while the equivalent potential temperature is conserved

(e.g., Fritsch and Chappell, 1980; Zhang and McFarlane, 1995). More sophisticated formulations include those of Kreitzberg and Perkey (1976) based on Kessler (1969), and Song and Zhang (2011) based on Sundqvist (1988).

**Table 12:** A sample of empirical values and assumptions used in the evaporation in the downdraft.

| Empirical value or assumption | Choices in the literature | Reference |
|---|---|---|
| Evaporation takes place at the same level where water detrains and is proportional to the liquid water mixing ratio of the detrained air $l_{dw}$ | $EVP \propto l_{dw}$ | Arakawa and Schubert (1974) |
| Detrained cloud condensates evaporate immediately | | Tiedtke (1989) |
| Function of the precipitation mixing ratio $q_{prec}$ and environmental thermodynamic properties | $EVP = \frac{(1-q_d^i/q_{sat}^i)\sqrt{q_{prec}^i}}{2\cdot10^3+10^4/(p^i q_{sat}^i)}$ where $q_d$ is the mixing ratio in the downdrafts, and $q_{sat}$ the saturation mixing ratio | Emanuel (1991) |
| Evaporation in the downdrafts cannot exceed a fraction of the precipitation | | Zhang and McFarlane (1995) |
| Constant evaporation coefficients | $C_{evap} = 1.0$ (for rain), $0.8$ (for snow) | Emanuel and Živković-Rothman (1999) |
| Estimated using a specified value of RH | $RH = 90\%$ | Bechtold et al. (2001) |
| Related to vertical profiles of grid-mean relative humidity RH and precipitation flux $R$ | $EVP = K_e(1-RH)R^{1/2}$, where $K_e = 0.2\cdot10^{-5}(\text{km m}^{-2}\,\text{s}^{-1})^{-1/2}\,\text{s}^{-1}$ | Park and Bretherton (2009) |
| Function of RH and the conversion of cloud water to rainwater $Pr$ | $EVP = C_{evap}(1-RH)Pr^{1/2}$, where $C_{evap} = 2.0\cdot10^{-4}(\text{km m}^{-2}\,\text{s}^{-2})^{-1/2}\,\text{s}^{-1}$ | Wu (2012) |

### 4.3.3 Aerosols

Aerosols play a key role in the climate system due to their influence on the Earth's energy budget through absorption and scattering of solar radiation. Focused on microphysical processes, aerosols serve as cloud condensation nuclei (CCN) and ice nuclei (IN) and thus affect cloud properties, dynamics, and precipitation. However, aerosol-convection interactions are very complex processes, seldom included in convection microphysics. Zhang et al. (2005) developed a new parameterization accounting for the effects of aerosols in stratiform and convective clouds. This was later modified by Lohmann (2008) to

include droplet activation by aerosols in terms of the updraft velocity $w$, temperature, aerosol number concentration, and size distribution, while ice nucleation is a function of $w$, aerosol properties, and air temperature. More recently, Grell and Freitas (2014) developed a new convective parameterization that includes an interaction with aerosols through an autoconversion of cloud water to rainwater dependent on CCN, parameterized in terms of the aerosol optical thickness (AOT) at 550 nm, as well

as an aerosol dependent evaporation of cloud drops. The authors also included tracer transport and wet scavenging in their parameterization. This convection scheme is currently available in WRF.

## 5 Closure: strategies to close the budget equation

Closure consists in defining the intensity or strength of convection, i.e., the amount of convection regulated by large-scale variables. Therefore, it is essential to close the budget equations (Eq. (5.1), Eq. (5.2) and Eq. (5.3)). Despite the number of hypotheses proposed in the literature, it is still considered an unresolved problem (Yano et al., 2013). The following subsections discuss the main closure types, as well as their main assumptions and empirical values. The impact of the closure formulation in convective model concludes the section.

### 5.1 Closure types

Existing convective closures for can be classified into diagnostic, prognostic, and stochastic. While diagnostic closures relate cumulus effects to the large-scale dynamics at a particular time scale, prognostic closures perform a time integration of explicitly formulated transient processes. Stochastic closures include randomness elements to closure schemes.

### 5.1.1 Diagnostic closures

Diagnostic closures include different types of closures based on a certain physical variable that expresses the intensity of convection. Table 13 shows a sample of empirical values and assumptions used in the closure in the updraft. In moisture convergence schemes, moisture convergence or vertical advection of moisture are selected as the closure variable (e.g., Kuo, 1974; Anthes, 1977; Krishnamurti et al., 1980, 1983; Kuo and Anthes, 1984; Molinari and Corsetti, 1985; Tiedtke, 1989), therefore assuming that convection consumes the moisture supplied by the large-scale processes.

The first parameterizations based on moisture convergence were too crude to produce results similar to those observed in nature, which led to the formulation of mass flux schemes. Early parameterizations lacked a theoretical framework to explain the interactions between the large-scale dynamics and convection or were incomplete, such as in Ooyama (1971). In an attempt to overcome this drawback, Arakawa and Schubert (1974) proposed a closed theory based on the QE of the CWF, which is similar to CAPE. Since then, many CPs use CAPE-like closures, generally assuming that the adjustment occurs at a relaxed time scale in contrast to the instantaneous adjustment proposed in Arakawa (1969), among others. Table 14 lists the most important choices made for the relaxation time scale.

**Table 13:** A sample of empirical values and assumptions used in the closure in the updraft.

| Main closure variable | Empirical value or assumption | Choices in the literature | Reference |
|---|---|---|---|
| Moisture convergence | Convection is controlled by the column-integrated water vapor | | Kuo (1974); Tiedtke (1989); Gerard (2007) |
| CWF | QE assumption | | Arakawa and Schubert (1974); Grell (1993) |
| | Relaxed at a certain time scale $\tau$ | | Pan and Wu (1995); Lim et al. (2014) includes a factor depending on the vertical velocity at the cloud base |
| | Relaxed at a certain time scale $\tau$ and towards a CWF reference value | $CWF_{ref} = 10\ \mathrm{J\,kg^{-1}}$ | Zhao et al. (2018) |
| CAPE | Consumed by convective activity at a certain time scale $\tau$ | | Fritsch and Chappell (1980); Betts (1986); Betts and Miller (1986) (deep convection is suppressed if the precipitation rate is negative), Nordeng (1994); Gregory et al. (2000); Bechtold et al. (2001) |
| | Consumption proportional to heat and moisture sources | | Donner (1993); Donner et al. (2001); Wilcox and Donner (2007) |
| | Consumed at an exponential rate by cumulus convection | | Zhang and McFarlane (1995) |
| | Modified by the vertical velocity | | Stratton and Stirling (2012) |
| Boundary-layer QE (CAPE) | QE between increased boundary layer moist entropy and decreased entropy due to moist downdrafts | | Emanuel (1995); Raymond (1995) |
| | Cloud-base upward mass flux is relaxed toward sub-cloud-layer QE. Includes a fixed relaxation rate α and a convection buoyancy threshold $\delta T_k$ | $\alpha = 0.02\ \mathrm{kg\,(m^2\,s\,K)^{-1}}$ and $\delta T_k = 0.65\ \mathrm{K}$ (EZ99), 0.90 K (BE01) | Emanuel and Živković-Rothman (1999); Bony and Emanuel (2001) |
| Free tropospheric QE (dCAPE) | Convective and large-scale processes in the free troposphere above the boundary layer are in balance. Contribution from the free troposphere to changes in CAPE is negligible. | | Zhang (2002); Zhang and Mu (2005a); Zhang and Wang (2006); Song and Zhang (2009); Zhang and Song (2010); Song and Zhang (2018) |
| Dilute CAPE | Consumed by convective activity at a certain time scale $\tau$ | | Kain (2004); Neale et al. (2008); Wang and Zhang (2013); Walters et al. (2019) |
| PCAPE | Relaxation of an effective PCAPE that includes the imbalance between BL heating and convective overturning | | Bechtold et al. (2014); Baba (2019) |
| CAPE and moisture convergence | | | Gerard (2015); Becker et al. (2021) |

295

300

**Table 14.** A sample of the empirical values and assumptions in the relaxation time scale. (Note: subscript *sh* refers to shallow convection)

| Empirical value or assumption | Choices in the literature | Reference |
|---|---|---|
| Varies within a specified range | $\tau = 10^3 - 10^4\ s$ | Arakawa and Schubert (1974) |
| | $0.5\ h < \tau < 1\ h$ | Bechtold et al. (2001) |
| | $1800\ s < \tau_{sh} < 3600\ s$ | Kain (2004) |
| Set to a constant value | $\tau = 2\ h$ $\tau_{sh} = 3\ h$ (B86, BM86, B01) | Betts (1986); Betts and Miller (1986); Zhang and McFarlane (1995); Lin and Neelin (2000); Bechtold et al. (2001); Zhang (2002, 2003); Zhang and Mu (2005b); Zhang and Wang (2006); Song and Zhang (2009); Zhang and Song (2010); Stratton and Stirling (2012) |
| | $\tau = 3600\ s$ | Nordeng (1994) |
| | $\tau = 1\ h$ | Pan and Wu (1995) |
| | $\tau = 8\ h$ | Zhao et al. (2018) |
| Inversely proportional to cloud efficiency | | Janjić (1994) |
| Function of the cloud depth CD, the vertical average updraft velocity $\overline{w}$ and an empirical scaling function f that decreases with horizontal resolution | $\tau = \frac{CD}{\overline{w}} f$. In B14 the minimum allowed value for $\tau$ is 12 min | Bechtold et al. (2008, 2014); Baba (2019) |
| Varies with a bulk RH over the cloud layer | | Derbyshire et al. (2011) |
| Varies according to the large-scale velocity ω within the range 1200−3600 s | $\tau = \max\left\{ \min\left[\Delta t + \max(1800 - \Delta t, 0) \times \left(\frac{\omega - \omega_4}{\omega_3 - \omega_4}\right), 3600\right], 1200 \right\}$, with $\Delta t$ the real model integration time step (s), $\omega_3 = -8 \cdot 10^{-3}(-2 \cdot 10^{-4})$, $\omega_4 = -4 \cdot 10^{-2}(-2 \cdot 10^{-3})$ over (ocean) | Han and Pan (2011) Lim et al. (2014); Han et al. (2019): $\omega_3 = -250/\Delta x$, $\omega_4 = 0.1 \cdot \omega_3$, $\Delta x$ the grid size (in m) |
| Dynamic formulation. Depends on the cloud depth *CD*, the grid resolution *Dx* and the in-cloud vertical velocity *w* | $\tau = \frac{CD}{w}\left[1 + \ln\left(\frac{25}{Dx}\right)\right]$ | Zheng et al. (2016) |

Following Lin et al. (2015), CAPE-like closures can be classified into two types according to the decomposition and constraints applied to the closure variable: the flux type and the state type. In the flux type, the change of the CAPE-like variable is decomposed into its large-scale and convective components. Of these types of closures, CAPE is the most commonly used closure variable in CPs (Fritsch and Chappell, 1980; Kain and Fritsch, 1993; Zhang and McFarlane, 1995; Gregory et al., 2000; Bechtold et al., 2001) with adjustment time scales varying from constant values to functional forms (Bechtold et al., 2008).

Other schemes with CAPE closure include the KF scheme in WRF (Kain, 2004), as well as in CAM (Neale et al., 2008; Wang and Zhang, 2013), CAM6, and the Met Office Unified Model Global Atmosphere 7.0 (GA7.0) (Walters et al., 2019) for deep convection schemes. While the preceding schemes applied convective closure to the full troposphere, Emanuel (1995) and Raymond (1995) proposed the so-called boundary-layer QE, where only the boundary layer component of the CAPE closure is considered. On the other hand, Zhang (2002) introduced a modified version of the QE assumption, in which only dCAPE is

employed as the closure variable, without considering the effect of boundary layer forcing. This type of closure, known as the free tropospheric QE or the parcel-environment QE, provides a better simulation of the diurnal cycle of precipitation than the boundary-layer QE (Zhang, 2003a), as well as a better representation of MJO and ITCZ than the QE assumption used in the Zhang-McFarlane scheme (Zhang and Mu, 2005b; Zhang and Wang, 2006; Song and Zhang, 2009; Zhang and Song, 2010).

Donner and Phillips (2003) confirmed these results in their analysis over oceanic tropical areas and midlatitude continental location of ARM. More recently, Bechtold et al. (2014) used the QE assumption to formulate a closure for the free troposphere based on boundary layer forcing. The dCAPE closure variable was replaced by PCAPE, defined as the integral over pressure of the buoyancy of an entraining ascending parcel with density scaling. The authors defined a convective adjustment time scale following Bechtold et al. (2008). This adjustment time is defined as the product of a convective turnover time scale $\tau_c$ and empirical scaling function $f(n)$ that decreases with increasing spectral truncation. At the same time, $\tau_c$ is given by the ratio of the convective cloud depth and the vertical averaged updraft velocity. The authors stressed the dependency of $\tau_c$ with PCAPE through the velocity, which agrees with the observations in Zimmer et al. (2011). The implementation of this closure in the ECMWF IFS led to a better representation of the diurnal cycle of precipitation.

In contrast to the previous flux-type closures, state-type closures decompose the change of the CAPE-like variable into its boundary layer component and free troposphere component, instead of in its large-scale and convective component. The main representatives of state-type closures are the convective adjustment schemes of Betts (1986), where mesoscale and subgrid scale cloud processes maintain QE, and Emanuel (1994), where QE is related to fluctuations of entropy in the sub-cloud layer. Differences between these adjustment schemes are in the adjustment time scale and reference profiles selected for the adjustment. For example, Emanuel (1994) included an adjustment time scale for the sub-cloud layer of the order or half day, while Betts and Miller (1986) found good results for values between 1 and 2 hours based on GATE wave data. More recently, authors such as Khouider and Majda (2006, 2008) and Kuang (2008) applied a state-type scheme only to the lower troposphere. An alternative principle to QE is the so-called activation control proposed by Mapes (1997), in which the intensity of deep convection is controlled by inhibition and initiation processes at low levels, and closure is formulated in terms of CIN and the turbulent kinetic energy (TKE) (Mapes, 2000; Fletcher and Bretherton, 2010). However, as highlighted in Yano and Plant (2012b) this formulation is not self-consistent, which is a must, as models are intended to test physical hypotheses (the reader is referred to Yano et al. (2013) for a detailed explanation). In Rio et al. (2009) the intensity of convection is controlled by sub-cloud processes, such as boundary layer thermals. The authors defined the closure in terms of the so-called available lifting power (ALP), which is the flux of kinetic energy associated with thermals. Grandpeix and Lafore (2010) also used an ALP closure in their wake parameterization for GCMS couple with Emanuel's scheme (Emanuel, 1991), as well as Hourdin et al. (2013) in the development of the LMDZ5B. While in Grandpeix and Lafore (2010) the source of ALP comes from the collapse of the wakes, in Hourdin et al. (2013) the thermal plumes and the spread of cold pools are the ones providing the power.

This section presented the assumptions and empirical values used in the formulation of the closure for updrafts. However, the magnitude of the downdrafts should also be addressed. In the schemes where it is included, it is commonly expressed as a fraction $\gamma_d$ of the closure of the corresponding updraft, setting $\gamma_d$ as a certain value (Johnson, 1976; Tiedtke, 1989; Baba, 2019). Alternatively, other authors have related $\gamma_d$ to precipitation efficiency (Emanuel, 1995; Bechtold et al., 2001), the RH

in the LFS (Kain, 2004) or proposed a formula for $\gamma_d$ in terms of the total precipitation rate within the updraft (Zhang and McFarlane, 1995). Table 15 lists some of the empirical values and assumptions used in closure in the downdraft.

**Table 15:** A sample of empirical values and assumptions used in the closure in the downdraft.

| Empirical value or assumption | Choices in the literature | Reference |
|---|---|---|
| Proportional to the updraft mass flux Mu | $M_d = \gamma_d\, M_u$, where $\gamma_d = 0.2$ | Johnson (1976, 1980); Tiedtke (1989); Nordeng (1994) |
| | $\gamma_d = 0.1 - PE$ | Emanuel (1989, 1995); Bechtold et al. (2001) |
| | $\gamma_d = 0.1 - RH$ | Kain (2004) |
| | $\gamma_d = 0.3$ | Baba (2019) |
| Function of updraft mass flux Mu and re-evaporation of convective condensate | | Grell (1993); Grell et al. (1994); Pan and Wu (1995) |
| Function of updraft mass flux Mu, height z, and maximum downdraft entrainment rate $\varepsilon^d_{max}$ | $M_d(z) = -\alpha\, M_b\, \dfrac{\exp[\varepsilon^d_{max}\cdot(z_{LFS}-z)]-1}{\varepsilon^d_{max}\cdot(z_{LFS}-z)}$ , where $\alpha$ is a proportionality factor that depends on the total precipitation and evaporation rates | Zhang and McFarlane (1995) (downdraft ensemble is constrained both by the availability of precipitation and by the requirement that the net mass flux at cloud base be positive) |
| | $M_d(z) = -\alpha\, M_{d(LFS)}\, \dfrac{\exp[\varepsilon^d_{max}\cdot(z_{LFS}-z)]-1}{\varepsilon^d_{max}\cdot(z_{LFS}-z)}$ , with $M_{d(LFS)} = 2\,(1 - \overline{RH_{LFS}})\, M_{u(LFS)}$, where $RH_{LFS}$ is the mean (fractional) RH at LFS, $M_{u(LFS)}$ is $M_u$ at LFS, and $\varepsilon^d_{max} = 5\cdot 10^{-4}\ \mathrm{m}^{-1}$ | Wu (2012) |

The discussion above focused on closure in deep convective and unified schemes. As for shallow convection closures, different approaches have been proposed since the publication of the first convection schemes. In this paper, we present a framework for the main empirical values and assumptions for shallow convection following the classification in Neggers et al. (2004). The authors classified the main shallow convection closures into moist static energy convergence, CAPE adjustment and sub-cloud convective velocity scaling.

In the moist static energy closures, the QE budget for moist static energy controls shallow convection activity. Based on the results obtained by LeMone and Pennell (1976) from trade wind cumuli, and the moisture convergence hypothesis from Kuo (1965, 1974) and (Lindzen, 1988), Tiedtke (1989) proposed a shallow convection closure based on the moist static energy closure. Later, Raymond (1995) and Emanuel (1995) used it in the boundary layer quasi-equilibrium for shallow convection, and Gregory et al. (2000) included it in a revised version of the ECMWF scheme. More recently, Bechtold et al. (2014) parameterized the mass flux for shallow convection in terms of the vertically integrated moist static energy tendency.

Other authors proposed shallow convection closures based on the relaxation of the system towards a certain reference state within a relaxation time scale, i.e., adjustment scheme. For example, Albrecht et al. (1979) used this closure in their study of the trade wind boundary layer specifying a constant adjustment time set to 1/3 day according to the observation results obtained by Betts (1975) for BOMEX. Later, based on observations from BOMEX and ATEX, Betts (1986) used an adjustment scheme

for shallow in which the thermodynamic structure tends towards a mixing line with an adjustment time set to 3 hours. Bechtold et al. (2001) used the same value for the relaxation time in their CAPE closure formulation for shallow convection.

One of the main representatives of TKE budget closures is Grant (2001), who assumed that mass flux at cloud base is proportional to the convective velocity scale proposed by Deardorff et al. (1969), $w_*$. The proportionality constant is the area fraction of cumulus updrafts and was determined by plotting the cloud-base mass flux versus the sub-cloud layer velocity scale in LES (see Tabe 16). This shallow closure was further used by other authors such as Soares et al. (2004), Siebesma et al. (2007) or Pergaud et al. (2009) in an EDMF, or Han and Pan (2011) and Han et al. (2017) in their revision of the NCEP GFS, among others. While Soares et al. (2004) defined the mass flux as the product of the updraft vertical velocity and a constant updraft fraction, Siebesma et al. (2007) scaled the mass flux with the standard vertical velocity deviation and set the proportionality constant to 0.3. In Pergaud et al. (2009) the closure is based on the mass flux near the surface instead of at the LCL. The authors set the proportionality constant to 0.065 based on LES results. Han et al. (2017) modified the closure by making the cloud base mass flux a function of the mean updraft velocity. This way, shallow convection can be triggered in the stable boundary layer. Another closure based on the relationship between mass flux and TKE is that described in Kain (2004), where the mass flux is scaled with the maximum TKE in the sub-cloud layer. The convective time period in this parameterization ranges from 1800 to 3600 s.

Similar to these parameterizations, Hourdin et al. (2002) developed a new mass parameterization of vertical transport in the convective boundary layer, known as the thermal plume model, where the closure depends on the maximum vertical velocity and an area fraction. As stated in Rio and Hourdin (2008) the area fraction is predicted according to the entrainment and detrainment in contrast to the constant values used in Soares et al. (2004) or Siebesma et al. (2007), among others.

Using LES simulations and observations, Grant and Lock (2004) proposed a shallow convective closure proportional to CAPE and the convective velocity scale $w_*$ More recently, Zheng et al. (2016) extended the shallow convection study of Grant and Lock (2004) and expressed the closure in terms of CAPE and cloud depth-averaged vertical velocity.

In the DualM framework, Neggers et al. (2009) defined the vertical structure of the updraft mass flux as the product of the updraft vertical velocity and updraft fraction. Based on results from De Rooy and Siebesma (2008) and the statistical distribution type in Sommeria and Deardorff (1977), the authors used a moist-zero buoyancy deficit to estimate the updraft area fraction and through it, the vertical velocity and mass flux.

A different shallow convection closure was suggested by Mapes (2000). Thea author expressed the mass flux in terms of CIN and TKE. Later, Bretherton et al. (2004) developed a new parameterization consisting in coupling a PBL turbulence model based on Grenier and Bretherton (2001) with a shallow convective mass flux scheme based on an entraining–detraining single-plume model. The closure assumes that a buoyant cumulus cloud can form if the vertical velocity of source air is high enough to penetrate the inversion layer in the sub-cloud layer and reach its LFC. The critical velocity is a function of CIN and the distribution of velocities is assumed to be Gaussian. The mass flux closure has a form similar to that proposed by Mapes (2000). In this case, it is an exponential function of the ratio between CIN and the average TKE in the sub-cloud layer calculated by the PBL scheme. In their simulations of the transition from shallow to deep convection, Kuang and Bretherton (2006)

405  applied the CIN-based closure proposed by Mapes (2000) with the updraft velocity at cloud base set to the sub-cloud layer TKE as in Bretherton et al. (2004). In the unified scheme of Hohenegger and Bretherton (2011), the shallow closure is a function of the ratio between CIN and mean planetary boundary layer TKE. Despite its use in several convection schemes, this parameterization is not self-consistent as already mentioned in section 5.1.1.

410  **Table 16:** A sample of empirical values and assumptions used in the cloud fraction. (Note: subscript *sh* refers to shallow convection)

| Empirical value or assumption | Choices in the literature | Reference |
|---|---|---|
| Function pf the relative humidity RH, liquid water mixing ratio $q_l$ and the saturation specific humidity $q_s$ | $a_{sh} = RH^{k_1}\left(1 - exp\left\{-\frac{k_2 q_l}{[(1-RH)q_s]^{k_3}}\right\}\right)$, where $k_1 = 0.25$, $k_2 = 100$ and $k_3 = 0.49$ | Xu and Randall (1996); Han and Pan (2011) |
| Constant | $a_{sh} = 0.03$ (G01, JS03), 0.01(S04), 0.065 (P09) | Grant (2001); Jakob and Siebesma (2003); Soares et al. (2004); Pergaud et al. (2009) |
| For deep convection, it is allowed to vary on the coarse mesh $j\Delta x$ | $a(j\Delta x) = [1 - \bar{a}_l(j\Delta x)]a^+$, where $0 \le \bar{a}_l \le 1$, and $a^+ = 0.002$ (K03) | Majda and Khouider (2002); Khouider et al. (2003) |
| For stratiform clouds, it is a function of RH and the difference in potential temperature between the surface $\theta_{surf}$ and 700 hPa $\theta_{700\,hPa}$ | $\theta_{700\,hPa} - \theta_{surf} = 20$ K (T04, N09) | Klein and Hartmann (1993); Tompkins et al. (2004); Neggers et al. (2009) |
| Prognostic | | Gerard and Geleyn (2005); Gerard (2007); Gerard et al. (2009); Tan et al. (2018) |
| Depends on the transition layer depth $d_{tr}$ and the sub-cloud mixed layer depth $h_{ml}$ | (For moist updraft) $a_{m,sh} = \left(\frac{d_{tr}}{h_{ml}}\right)\frac{1}{2p+1}$, with $p = 2.2$, (for dry updraft) $a_{d,sh} = A - a_m$, where $A = 0.1$(N07, N09*) is the total updraft fractional area | Neggers et al. (2007, 2009); Neggers (2009)=N09* |
| Depends on the wake radius $R_w$ and density $D_w$ | $a_w = D_w \pi R^2$ | Grandpeix and Lafore (2010) |
| Depends on the turbulent kinetic energy TKE | $a = (2TKE/3)^{1/2}$ | Mapes and Neale (2011) only for the first generation |
| Depends on the previous generation value and *organization* | $a_{g+1} = a_g^2 + org(a_g - a_g^2)$, where g indicates the generation | Mapes and Neale (2011) for generations different than the first one. |
| Stochastic formulation | Conditioned on CAPE | Bengtsson et al. (2013) for deep convection using cellular automat (CA); Dorrestijn et al. (2015); Gottwald et al. (2016) Sakradzija et al. (2015, 2016) for shallow convection |
| Function of the convective updraft radius $R$ and the grid-box area $A_{grid}$ | $a = \frac{\pi R}{A_{grid}}$ | Grell and Freitas (2014); Han et al. (2017) |

In the MM5, Deng et al. (2003) proposed three different shallow convection closures depending on the values of the cloud depth CD, cloud top height $z_t$, and LFC height $z_{LFC}$, and assumed a uniform updraft geometry. The closures include a TKE-based closure, a CAPE closure and a hydrid closure. TKE-based closure is used when $z_t \le z_{LFC}$. In this closure, the cloud

base mass flux in the sub-cloud layer scales with the maximum diagnosed TKE in the sub-cloud mass-source layer over a relaxation time scale. If $CD \geq 4$ km, the CAPE closure applies, while for $CD < 4$ km and $z_t > z_{LFC}$ a hybrid closure between TKE and CAPE closures is used. The transition is done through a simple linear averaging. More recently, Freitas et al. (2020) proposed a trimodal formulation instead of the unimodal deep plume used in Grell and Freitas (2014) to represent shallow, congestus and deep convection. Closures for shallow convection include the boundary layer quasi-equilibrium from Raymond (1995), the closure proposed in Grant (2001), and a closure based on the heat engine treatment of convection applied in Rennó et al. (1994). This closure relates the updraft cloud base mass flux to the buoyancy surface flux, a certain thermodynamic efficiency, and the total CAPE that is equivalent to the standard CAPE.

### 5.1.2 Prognostic closures

Compared to the QE assumption used in the majority of the diagnostic closures mentioned above, prognostic closures do not distinguish between large-scale and convective processes and substitute the QE assumption with time integration of prognostic equations. These equations explicitly account for the time changes of different physical variables, i.e., convective kinetic energy or $h$, which are related to the cloud-base mass flux through a dimensional parameter. Energy dissipation rate is also included in this type of closure through a dissipation term, either determined by a second dimensional parameter called dissipation time (e.g., Randall and Pan, 1993; Pan and Randall, 1998; Yano and Plant, 2012a) or expressed in terms of the entrainment rate and an aerodynamic friction coefficient (e.g., Gerard and Geleyn, 2005). Gerard and Geleyn (2005) defined cloud base mass flux as $M_u = -a_u w_u$ where $a_u$ is a prognostic updraft fraction area, obtained by a moist static energy closure, and $w_u$ is a prognostic vertical updraft velocity. Gerard (2007) and Gerard et al. (2009) also used this approach and even applied it for downdrafts (Gerard et al. 2009). Other schemes using prognostic updraft fractional areas include those of Grandpeix and Lafore (2010), Mapes and Neale (2011) and Tan et al. (2018), among others (see Table 16).

### 5.1.3 Stochastic closures

Usually subgrid-scale processes are considered in an ensemble mean sense in CPs (Lin and Neelin, 2000, 2002). Stochastic closures include randomness elements to convective schemes closures to represent these subgrid-scale processes in a more realistic way. Numerous stochastic convective parameterizations have been proposed (e.g., Lin and Neelin, 2000, 2002; Majda and Khouider, 2002; Lin and Neelin, 2003; Khouider et al., 2003; Khouider, 2014). However, as Stechmann and Neelin (2011) stated, sometimes the distinction between stochastic triggers and stochastic closures is not clear. Differences between the proposed closures are in the type of stochastic process employed. For instance, Stechmann and Neelin (2011) proposed a stochastic closure for precipitation using a Gaussian white noise, while Majda and Khouider (2002) and Khouider et al. (2003) used a Markov jump process.

For deep convection, Lin and Neelin (2000) include a first-order autoregressive random noise component in the convective parameterization of Betts and Miller (1986) keeping the convective relaxation timescale. This random noise is expressed as

$\xi_t = c_\xi \xi_{t-1} + z_t$, where $c_\xi$ is an autoregressive coefficient that yields an autocorrelation time $\tau_\xi$ for the process and $z_t$ is white noise with zero mean and standard deviation $\sigma_z$. The authors evaluated three values for $\tau_\xi$, i.e., 20 min, 2 hours and 1 day, with three different $\sigma_z$, i.e., 4.5 K, 0.8 K and 0.1 K, respectively. Longer $\tau_\xi$ produced better results compared to observations. Lin and Neelin (2003) introduced this stochastic component in the ZM closure with $\tau_\xi = 1$ day and $\sigma_z = 1000$ J kg$^{-1}$. This

scheme increased precipitation variance toward observations. Based on the variability around the equilibrium state, Plant and Craig (2008) and Groenemeijer and Craig (2012) used a PDF to obtained random values for the cloud-base mass flux. This PDF expresses the chance of launching a cloud with a certain radius between two calls of the convective scheme. The radius is assumed to be related to the mass flux. It is defined as $p(m)dm = \frac{1}{\langle m \rangle} \exp\left(\frac{-m}{\langle m \rangle}\right) dm$, where $m$ is the mass flux per cloud and $\langle m \rangle$ is its ensemble average, both related through the definition of updraft radius $m = \frac{\langle m \rangle}{\langle R^2 \rangle} R^2$. Moreover, the closure time

scale in Plant and Craig (2008) is defined as $\tau_c = kL = k\sqrt{\frac{\langle m \rangle}{\langle \bar{M} \rangle}}$, where $\langle \bar{M} \rangle$ is the ensemble-mean total coud-base mass fux calculated as in Kain and Fritsch (1990), and $k$ is a constant that depends on the definition of adjustment. The default parameter choices in Plant and Craig (2008) are $\langle m \rangle = 2 \cdot 10^{-7}$ kg s$^{-1}$, a root mean squared cloud radius of $\langle R^2 \rangle^{1/2} = 450$ m and $k = 0.3$ s m$^{-1}$. In Groenemeijer and Craig (2012) these values did not produce enough convective, so they were changed to $\langle m \rangle = 1 \cdot 10^{-7}$ kg s$^{-1}$ and $\langle R^2 \rangle^{1/2} = 1200$ m, and fixed $\tau_c = 600$ s. Bengtsson et al. (2013) introduced a CA in the

parameterization of the updraft mesh fraction $a_u$ used in the Gerard et al. (2009) cumulus convective scheme closure. Using observational data, Dorrestijn et al. (2015) determined the $a_u$ for various cloud types using Markov chains. The one for deep convection was later implemented in the Tiedtke cumulus scheme in the Simplified Parameterizations, Primitive Equation Dynamics (SPEEDY).

For shallow convection, Sakradzija et al. (2015) developed a stochastic shallow parameterization following the studies of Craig

and Cohen (2006) and Plant and Craig (2008) for deep convection. In this scheme, the number of new clouds is sample form a Poisson distribution while the lifetime average mass flux for each new cloud is randomly sampled from a Weibull distribution with two modes, namely forced and passive clouds on one hand, and active clouds on the other. This Weibull distribution is defined through a scale $\lambda$ and a shape $k$ parameter. The cloud lifetime is defined as $\tau_{clt} = \alpha_i m^{\beta_i}$, where the coefficients are obtained from the non-linear least square fitting of the joint distribution of cloud mass flux and cloud lifetime. The total cloud-

base mass flux is then calculated by integrating the instantaneous mass flux distribution, i.e., $\langle M \rangle = \int_0^\infty m \langle \tau_{clt}(m) \rangle \langle Gp(m) \rangle dm$ or $\langle M \rangle = G\alpha\lambda^{k+1}\Gamma\left(2 + \frac{1}{k}\right)$, where $G$ is the cloud generating rate. The following values were used for this parameterization: $k = 0.7$, $\lambda_1 = 7269.08$ kg s$^{-1}$, $\lambda_1 = 29868.48$ kg s$^{-1}$, $\alpha_1 = 0.02$ kg$^{-1}$, $\alpha_2 = 0.33$ kg$^{-1}$, and $G = 4.55$ s$^{-1}$ (subscript 1 refers to forced and passive clouds, and subscript 2 for active clouds. The reader is referred to Sakradzija et al. (2015) for values of other parameters). This scheme was later implemented in EDMF

(Sakradzija et al., 2016) and ICON (Sakradzija and Klocke, 2018) with variations in the values of the aforementioned parameters.

## 5.2 Impact of closure on convective models

The closure problem is one of the major challenges in CPs. As well as being essential to close the budget equations (Eq. (5.1), Eq. (5.2) and Eq. (5.3)), it plays an important role in the performance of CPs. For instance, Bechtold et al. (2008) obtained a better representation of the rainfall pattern and tropical wave activity with their modifications of the entrainment and convective adjustment time in the deep convection scheme in IFS. In Rio et al. (2009), the representation of the diurnal cycle of precipitation is greatly improved using the ALP deep closure in a 1D model. In their formulation, the convective mass flux scheme is coupled with cold pools and the thermal plume model through the ALP. Using a dilute CAPE closure together with convective momentum transport, Neale et al. (2008) improved the representation of ENSO in CAM3. Adding a stochastic component to the deep convection closure in BMJ, Lin and Neelin (2000) obtained a better representation of the intraseasonal variability. Later, Lin and Neelin (2003) include a stochastic component in the deep closure of the ZM scheme. The daily variance was much closer to observations than without the stochastic component. Moreover, the SPCZ was better placed.

Replacing the CAPE closure used in the ZM scheme by a dCAPE closure, Zhang (2002) improved the simulation of precipitation, moisture and temperature for midlatitude continental convection. This closure also improved the diurnal cycle of precipitation over the southern great planes in the U.S. (Zhang, 2003b). The replacement of the ZM closure by dCAPE provided a better representation of the tropical precipitation in NCAR CCM in Zhang and Mu (2005a). With this closure, the precipitation was enhanced over the western Pacific monsoon region during June, July and August, as well as the SPCZ during December, January and February. In the representation of the MJO, Zhang and Mu (2005b) used the closure and convection trigger proposed in Zhang and Mu (2005a) and removed the restriction in the convection originating level. The simulated MJO was more consistent with the observations in terms of variability in precipitation, outgoing longwave radiation and zonal wind, and exhibited a clear eastward propagation. However, the precipitation signal and the time period of the MJO differ from the observations. This revision of the ZM scheme used in the NCAR Community Climate System Model (CCSM3) also alleviates the biases related to the double ITCZ in precipitation and cold tongue in Sea Surface Temperature (SST) over the equator, among other benefits (Zhang and Wang, 2006; Song and Zhang, 2009; Zhang and Song, 2010). Wang and Zhang (2013) evaluated three different trigger and closures assumptions in CAM4 and CAM5 and highlighted the need of using multiple independent observations simultaneously to constrain models to reduce the degrees of freedom as well as the need to avoid the individual treatment of model physical parameterizations. Wang et al. (2016) obtained a better representation of the precipitation intensity, especially over the tropical belt as well as improved simulations of the eastward propagating intraseasonal signals of precipitation and zonal wind by coupling the Plant and Craig (2008) stochastic parameterization with the ZM scheme in CAM5. More recently, Becker et al. (2021) showed a better representation of the propagation and organization of mesoscale convective systems, such as African squall lines, when adding a term for the integrated and scaled total advective moisture tendency to the CAPE closure.

Using CRM simulations, Kuang and Bretherton (2006) tested the viability of representing the transition from shallow to deep convection using a CIN-based closure similar to the shallow closure in Bretherton et al. (2004). Results from an idealized

numerical experiment of shallow-to-deep convection transition are in agreement with the CIN-based closure and do not support a closure based solely on CAPE. Later, Fletcher and Bretherton (2010) extended the Bretherton et al. (2004) shallow closure to deep convection with the goal of finding a closure that works well for both shallow and deep convection without changing any parameter. Three CRM simulations forced with observations from ARM Great Plains, Kwajalein Experiment (KWAJEX) and BOMEX were used to test this closure as well as a CAPE and a Grant closure (Grant, 2001). The CIN-based closure was more skillful in the prediction of the cloud-base mass flux and performed well for both deep and shallow convection. Hohenegger and Bretherton (2011) modified the UW shallow convection scheme to develop a unify scheme for shallow and deep convection. The closure introduced also relates the cloud base mass flux to TKE and CIN taking into account the contribution of cold pools to the increase of TKE. LES simulations and BOMEX, KWAJEX and ARM were used to formulate and improve this parameterization. Tested in the Single-column Community Atmosphere Model (SCAM) single-column modeling framework, this parameterization was able to represent both shallow and deep convection and mid-latitude continental convection. Han and Pan (2011) modified the deep scheme in SAS (Pan and Wu, 1995) by increasing the allowable cloud-base mass flux, originally set to $0.1 \ \mathrm{kg} \ (\mathrm{m}^2\mathrm{s}^{-1})^{-1}$, with a Courant-Friedrichs-Lewy (CFL) criterion to make cumulus deeper and stronger. This scheme effectively eliminated the remaining instability in the atmospheric column that was producing excessive grid-scale precipitation in the original formulation. Using a PCAPE closure with boundary layer forcing, the scheme for shallow and deep convection described in Bechtold et al. (2014) represented fairly well the observed daytime evolution of convection over land when compared with observations such as satellite data. Moreover, the evolution of shallow and deep convection agreed with CRM results. Over Europe, better represented the mainly surface-driven convection over the Balkans and the Atlas Mountains, as well as forced convection over Central Europe, and reduced unrealistic rates of snowfall along the coast of the British Isles and near European continent for a particular winter case. Han et al. (2020) obtained similar results using this closure in KIM (The Netherlands Institute for Transport Policy Analysis). The afternoon peak was delayed and the biases of the overestimated precipitation over land in the morning and late afternoon was reduced.

Focused on closures for shallow convection, different authors have analyzed the impact that shallow convection closures have on the simulation of the diurnal cycle. For instance, Neggers et al. (2004) evaluated moist static energy closure, CAPE adjustment and sub-cloud convective velocity scaling closure against LES simulations and analyzed the impact of each closure on the simulation of the diurnal cycle. Among those, the sub-cloud convective velocity scaling closure showed the best results. The onset, dissipation time and cloud cover of cumulus clouds was well captured by the EDMF scheme in Soares et al. (2004). Scaling the mass flux with the standard vertical velocity deviation in the EDMF, Siebesma et al. (2007) obtained realistic representation of the main properties of dry convective boundary layers. Using a similar closure, Pergaud et al. (2009) showed the ability of the EDMF scheme to represent mixing in the countergradient zone and to handle the diurnal cycle of boundary layer cumulus clouds. Similar results were obtained by Rio and Hourdin (2008) in terms of the diurnal cycle of the boundary layer. The shallow cumulus parameterization developed by Bretherton et al. (2004) reproduced well LES results obtained by Siebesma and Cuijpers (1995) and Siebesma et al. (2003) for a subperiod of BOMEX, and by Wyant et al. (1997) for the transition from stratocumulus to trade. However, this transition was slightly abruptly in the simulations with the shallow

parameterization. McCaa and Bretherton (2004) further analyzed the performance of this scheme in a regional climate simulation of the subtropical northeast Pacific Ocean in MM5. The regional mean shortwave cloud radiative forcing and vertical structure was better represented by this scheme compared to other parameterizations of cloud-topped boundary layer processes. In the DualM framework, Neggers et al. (2009) defined the cloud-base mass flux as the product of updraft fraction and updraft vertical velocity. Examined for ATEX, this closure, produced steeper gradients closer to LES results than the ones obtained with a fixed structure of the mass flux, and concluded that this result is an indicator of the interaction between the mass flux and environmental humidity introduced by the closure. Han and Pan (2011) replaced the shallow convection in SAS with a new formulation using the shallow closure describe in Grant (2001). Compared to the original formulation, this new scheme did not destroy stratocumulus clouds off the west coasts of South America and Africa.

## 6 Conclusions

Numerical models need simplifications in order to cope with the complexity of the physical processes actually ocurring in the atmosphere. The degree of simplification in the physics is evolving at a pace inverse to the availability of computational power. Thus, early convective parameterizations (as well as parameterizations of radiation, turbulence, microphysics, etc.) were based on very simple assumptions, such as the conditional instability of the second kind (CISK) first presented by Charney and Eliassen (1964) and Ooyama (1964) in tropical cyclone modeling. Manabe et al. (1965) proposed a different parameterization, the so-called adjustment scheme, where atmospheric instability is removed through an adjustment towards a reference state. The instability was removed instantaneously, and a condensed water precipitated immediately. However, the scheme produced very large precipitation rates, and a saturated final state after convection, which is rarely observed in nature (Emanuel and Raymond, 1993). To alleviate this issues, relaxed adjustment schemes and penetrative adjustment schemes (Betts, 1986; Betts and Miller, 1986) were proposed. Such improvements were only possible when more powerful computers became available. However, novel theoretical approaches ahead of the technological capabilities of the time have also greatly impacted the field. Thus, the first parameterizations based on moisture convergence were too crude to produce results similar to those observed in nature, which led to the formulation of mass flux schemes. Simulations improved with further refinements of the interaction of cumulus clouds with the large-scale environment by, for instance, Ooyama (1971) (a statistical ensemble of bubbles represent cumulus convection) or Yanai et al. (1973) (detrainment and cumulus-induced subsidence). Early parameterizations lacked a theoretical framework to explain the interactions between the large-scale dynamics and convection or were incomplete, such as in Ooyama (1971). In an attempt to overcome this drawback, Arakawa and Schubert (1974) proposed a closed theory based on the cloud work function and adjustment towards QE. A few years after, thanks to the increase in computational power, more complex parameterizations and new variables based on observations were implemented to achieve better spatial and temporal resolutions. Krueger (1988) put forward the Cloud Systems Resolving Model (CSRM) idea to explicitly simulate convective processes over a kilometer scale, instead of using parameterizations. However, this approach entails an extremely high computational cost. As an alternative with a lower computational cost, Multiscale Model Framework

(MMF) or superparameterizations (SP) emerged. In this case, convective parameterizations are replaced by 2D cloud resolving models (CRMs), or even a 3D LES model, at each grid cell of a GCM (Grabowski and Smolarkiewicz, 1999).

To alleviate problems associated to traditional convective parameterizations, e.g. the representation of the diurnal cycle of convection (e.g.,Yang and Slingo, 2001; Guichard et al., 2004), several studies introduce modifications in existing models. Challenges remain for convective parameterizations. As highlighted in Rio et al. (2019), three of these major challenges include (a) improve the representation of convective cloud ensembles, (b) improve the representation of convective memory and organization, and (c) improve the representation of convection to large-scale interactions. The reader is referred to Rio et al. (2019) for a comprehensive review. Here, only the main representatives of each challenge are mentioned.

Regarding the first challenge, current approaches to improve the representation of convective cloud ensemble include unified and multi-object frameworks parameterizations that account for the coexistence of more numerous cloud types within a model grid cell, and different methods to compute the vertical profile of cloud properties. Traditionally, models have used separate parameterizations for shallow and deep convection. Guichard et al. (2004) stressed the necessity of using and ensemble of parameterizations that represents a succession of convective regimes. Some modelers proposed to keep shallow and deep convection parameterizations separate due to their different nature and then use a parameterization to couple them (e.g., Rio et al., 2013), while others proposed unified schemes that attempt to merge shallow and deep convection into one parameterizations (e.g., Guérémy, 2011; Arakawa and Wu, 2013; Wu and Arakawa, 2014; Park, 2014a, b; D'Andrea et al., 2014; Kwon and Hong, 2017; Zhao et al., 2018). Besides, models traditionally split the turbulence parameterization among the PBL and moist convection simplifying the treatment of turbulence but requiring the addition of an artificial closure to match both schemes (Sušelj et al., 2014). Unified models have been also used to merge these parameterizations, such as the so-called Cloud Layers Unified By Binomials (CLUBB) (Golaz et al., 2002a, b; Larson et al., 2002). Two different approaches have been proposed that unify the PBL, shallow and deep convection. Those approaches are the so-called EDMF framework (e.g., Hourdin et al., 2002; Köhler et al., 2011; Hourdin et al., 2013; Bhattacharya et al., 2018) and third-order turbulent schemes (e.g., Guo et al., 2014, 2015). Parameterizations account for the coexistence of more numerous cloud types within a model grid cell include the use of Markov chains considering a certain number of cloud types (Khouider et al., 2010; Dorrestijn et al., 2013b; Peters et al., 2013) or the use of a probability density function (PDF) (e.g., Plant and Craig, 2008; Sakradzija et al., 2016) , among others. As for the methods to compute the vertical profile of cloud properties, numerous studies apply a deterministic entrainment to different cloud types; others use stochastic entrainment parameterizations (e.g., Raymond and Blyth, 1986; Emanuel and Živković-Rothman, 1999; Grandpeix et al., 2004; Romps and Kuang, 2010; Sušelj et al., 2013; Romps, 2016). The vertical profile of vertical velocity also needs further attention as many schemes do not solve an equation for the vertical velocity, and the ones that do it are mostly based on the equation proposed by Simpson and Wiggert (1969) as highlighted in Roode et al. (2012).

For the second challenge, improving the representation of convective memory and organization, there are at least two outstanding issues. On the one hand, as pointed out in Davies et al. (2009), the QE hypothesis does not account for convective memory. Different strategies have been proposed to include it in convective parameterizations, such as the use of prognostic

variables (e.g., Pan and Randall, 1998; Gerard and Geleyn, 2005; Piriou et al., 2007; Mapes and Neale, 2011; Hohenegger and Bretherton, 2011; Willet and Whitall, 2017; Tan et al., 2018), Markov chains (e.g., Khouider et al., 2010; Hagos et al., 2018), cellular automaton (CA) assigning a prescribed lifetime to each active cell (e.g., Bengtsson et al., 2011, 2013) or cold pools (e.g., Grandpeix and Lafore, 2010; Park, 2014; Del Genio et al., 2015; Colin et al., 2019). On the other hand, as for the representation of convective organization, Donner (1993), Alexander and Cotton (1998) and Donner et al. (2001) represented the effects of mesoscale circulations and Mapes and Neale (2011) introduced a prognostic variable called *organization* that represents the degree of subgrid organization. Other studies accounting for convective organization use surface cold pools (e.g., Rio et al., 2009; Grandpeix and Lafore, 2010; Rochetin et al., 2014a, b; Park, 2014a, b; Böing, 2016), slantwise overturning model (e.g., Moncrieff et al., 2017), CA (e.g., Shutts, 2005; Bengtsson et al., 2011, 2013, 2019, 2021), or PDF-based or spectral schemes based on a discretized distribution (e.g., Neggers et al., 2003; Wagner and Graf, 2010; Neggers, 2012; Park, 2014; Neggers, 2015). Accurate representations of precipitation and cloud cover are important for the spatial organization and the time evolution of convective systems. Parameterizations accounting for the microphysics of precipitation include those of Feingold (2003), Genio et al. (2005), McFiggans et al. (2006) and Heymsfield et al. (2013), among others. Besides, several studies attempted to improve convective cloud radiative effects using PDFs (e.g., Bogenschutz et al., 2010; Perraud et al., 2011; Hourdin et al., 2013; Storer et al., 2015; Qin et al., 2018).

The third main challenge is to achieve better representations of convection to large-scale interactions, i.e., shallow convection, transitions from shallow to deep and from deep to organized convection. For transitions from shallow to deep, various approaches have been proposed, especially focused on the representation of the diurnal cycle of precipitation (e.g., Rio et al., 2009; Stratton and Stirling, 2012; Rio et al., 2013; Bechtold et al., 2014; Rochetin et al., 2014; Peters et al., 2017). Other aspects that deserve more attention, among others, are the representation of the impact of sea breeze in deep convection initiation over islands, and the tendency to show strong positive tropical rain biases for model with strong intraseasonal variability due to the sensitivity of convection to free tropospheric humidity through entrainment (Rio et al., 2019). Transitions from deep to organized convection also deserve more attention due to the role that mesoscale convective system play on weather and climate.

The field of modeling convection is full of details and intricacies. As already mentioned, mass flux convective parameterization schemes are still the most common convective parameterizations used in ESMs, RCMs, and NWP models. Besides, models have traditionally used separate parameterizations for shallow and deep convection Therefore, we mainly focused our attention to the assumptions and empirical values used in shallow and deep mass flux schemes for their three main elements, i.e., trigger, cloud model and closure. In the activation of convection, the main differences between shallow and deep convection are in the cloud-depth criterion, the updraft radius and in the buoyancy threshold. Both cloud depth and radius are always set to smaller values compared to deep convection. As for the temperature perturbation that some deep convective parameterizations include in the buoyancy threshold, it is absent in shallow convection trigger. Commonly, the procedure followed to find cloud base and trigger convection is the same for both schemes, though some studies set different conditions for the USL (Han and Pan,

2011) or use a vertical velocity criterion to trigger shallow convection (Bretherton et al., 2004; Park and Bretherton, 2009). The cloud-depth criterion is what decides which type of convection activates.

Numerous parameterizations of entrainment and detrainment have been proposed for shallow and deep convection including turbulent and dynamical components (e.g., Tiedtke (1989) and Nordeng (1994) for deep and shallow convection), constant values (e.g. Song and Zhang (2017) for deep and (Siebesma, 1998) for shallow convection), inverse proportionality to height (e.g., Siebesma and Cuijpers (1995) for deep and Jakob and Siebesma (2003) for shallow convection) or to the vertical velocity of the parcel (e.g., Gregory (2001) for both deep and shallow convection), or dependence on a critical mixing fraction (e.g., Kain and Fritsch (1990) for deep and Bretherton et al. (2004) for shallow convection), among others. For those schemes using the same parameterization for shallow and deep convection, the main difference between the two types is in the values, higher for shallow than for deep convection. Entrainment and detrainment formulations for downdrafts usually use similar parameterization as for updrafts. In terms of the microphysics, shallow convective schemes usually do not include a parameterization of conversion to precipitation.

As for the closure formulation, numerous deep convective schemes use CAPE-based closures, although formulations based on convective adjustment in terms of CIN and TKE or using stochastic closure have been also proposed. For shallow convection, the most used are TKE-based closures. Other closures such as moist static energy convergence (Tiedtke, 1989) and CAPE adjustment closures (Betts, 1986) are also used in shallow convection. For the latter, the adjustment time is usually higher for shallow than for deep convection. In the parameterizations where it is included, downdraft closure is commonly expressed as a fraction of the closure of the corresponding updraft.

Convective parameters require fine tuning, but there is no explicit methodology to do so. In some cases, the authors use the variables that are easiest to measure. In others, mean values describe processes that cannot be modeled in sufficient detail, or the values represent particular conditions for certain locations and atmospheric events (Mauritsen et al., 2012). For instance, Bony and Emanuel (2001) adjusted their water vapor and temperature prediction using the TOGA-COARE data measured in Western Pacific Ocean in 1993, while Betts and Miller (1986) used GATE datasets measured over the tropical Atlantic Ocean in 1974 to develop their deep convection scheme. Hence, empirical values and assumptions selected this way might yield good results when compared to observations from certain locations and less good results for others. Commonly, manual tuning of convective parameters is used, although various automatic methods have recently been used to estimate parameters, including the variational method (Emanuel and Živković-Rothman, 1999), Bayesian calibration (e.g., Hararuk et al., 2014; Wu et al., 2018), simulated annealing method (e.g., Jackson et al., 2004, 2008; Liang et al., 2014), genetic algorithm (e.g., Lee et al., 2006), ensemble data assimilation (e.g., Ruiz et al., 2013; Li et al., 2018), or machine learning (e.g., Schneider et al., 2017) among others. Recently, Couvreux et al. (2021) proposed a new method that performs a multi-case comparison between SCM and LES results to calibrate parameterizations. The method uses machine learning without replacing parameterizations.

Comparisons with observations were, and still are, crucial to the development of convective parameterizations. For instance, the underprediction of large-scale precipitation by dry adiabatic models compared to observations led to the inclusion of moist

adiabatic processes in NWP models (Smagorinsky, 1956), and the lake-effect snow observations (Niziol et al., 1995) forced to reduce the minimum cloud-depth threshold in Kain and Fritsch (1993) to 2 km. However, observations suffer from data gaps and the instruments used are not able to sampling key variables in parametric equations. The use of observations by the convective modeling community has not been sufficient so far. The reasons being twofold. Basic convective quantities like mass flux and important parameters like adjustment time scale, entrainment and microphysical parameters can often be only indirectly inferred from observations like infrared and microwave satellite data, radar data, rainfall rates, radiosonde networks and reanalysis data. When we say that they are indirectly inferred, we mean that these quantities are adjusted to optimize the model fit to the observed radiative and surface fluxes as well as the observed temperature and wind field. On the other hand, long-term instrumentation deployment at meteorological supersites (e.g. Neggers et al., 2012; Song et al., 2013; Gustafson et al., 2020; Zheng et al., 2021) or dedicated convection field campaigns like GATE, TOGA-COARE, DYNAMO, PECAN (Geerts et al., 2017), EUREC4A (Bony et al., 2017), to mention a few, have been conducted to quantify convection and its effect on the large-scale flow, and powerful LES data are available with statistical samples of the convective updraft and downdraft properties. However, the dilemma is that these data are only available locally or for specific setups, LES data also need to be constrained by observations and an accurate convection parameterization in a global model needs to be constrained globally.

Modern extensive big datasets such as those derived from COPERNICUS data are very relevant to constrain assumptions and calibrate parameterizations. Recently, Neggers et al. (2012) and Gustafson et al. (2020), among others, have provided a successful attempt to reconcile observations and LES data. This new approach consists in combining LES outputs with observations. Indeed, high-resolution models provide additional information in 4D that is not possible to be obtained from point-based measurements (Gustafson et al., 2020). The complementary approach consists of new dedicated satellite missions such as INCUS or the follow-on to CloudSat and CALIPSO, which can provide global, homogeneous and time-extended observations. Satellite estimates of the convective mass flux are becoming available (Jeyaratnam et al., 2021) and new missions are in the planning to fill the gap in global, multiple-regime observations of convection. Although observations have long been used to tune parameters in convective schemes to reduce errors, it is still unclear whether these tuned parameters based on particular datasets can improve model skills across different locations, model resolutions or atmospheric events. As described above, it is known that model results are sensitive to the empirical values in convection. To summarize here the numerous sensitivity studies, some have reported that the location and intensity of precipitation are extremely sensitive to cumulus parameterization (e.g., Bechtold et al., 2008; Ma and Tan, 2009; Chikira and Sugiyama, 2010). For instance, Wang et al. (2007) improved the simulated diurnal cycle over land and ocean by increasing the entrainment/detrainment rates for deep and shallow convection used in the Tiedtke scheme, which tends to simulate convective precipitation too early in the day and with an unrealistic amplitude over land. Thus, the choice of a convective scheme impacts the diurnal cycle (e.g., Bechtold et al., 2004; Wang et al., 2007), as well as the simulation of monsoon precipitation in climate models (e.g., Mukhopadhyay et al., 2010), the MJO (e.g., Lin et al., 2006), the ENSO (e.g., Wu et al., 2007; Neale et al., 2008), the ITCZ configuration (e.g., Liu et al., 2019) or cloud cover and precipitation over urban areas (e.g., Karlický et al., 2020), among others. This topic has profound

practical effects: it has been shown that choices in the convective parameterization affect the prediction of track, intensity and associated rainfall of tropical cyclones (e.g., Mohandas and Ashrit, 2014). However, the impacts of the empirical values in convection are extremely code-specific and often errors in calibration of one parameter are hidden by errors in another. Examples of these include masking errors in vertical structure due to errors in cloud overlap (Neggers and Siebesma, 2013) or the too-few, too-bright problem (e.g., Nam et al., 2014). Therefore, results obtained in one GCM with a particular set of empirical values might differ from results obtained in a different GCM with the same set of empirical values.

Timely providing the correct amount of precipitation at the right location is still a challenge for models. In the weather realm, Fig. 2 is an example of how different the precipitation field may look depending on the cumulus parameterization used. All a priori sensible methods locate the maximum and minima in different parts of typhoon Megi and predict different areas and total accumulations. Fig. 3 shows differences in the location and pressure of typhoon Megi and Chaba with initial perturbations, and when 7 different convection parameters are perturbed using SPP. Compared to the initial perturbations, changes in convection parameters show a bigger dispersion and yield to a wider range of pressure values for each of the cyclones. In the climate model realm, validation exercises focusing on precipitation (Tapiador et al., 2012, 2017, 2018) have shown the importance and challenges of comparing model outputs with precipitation measurements in order to improve model performance. Indeed, the difficulties of quantitative precipitation estimation suggest precipitation as a privileged metric to gauge model performance (Tapiador et al., 2019b). The "ultimate test", as has been described, makes precipitation science an active field of research. As discussed in such paper, there is no complete agreement even in the reference data, with datasets differing even in such aggregated value as the global mean value of the precipitation on Earth. Advances in satellite precipitation estimation (Kummerow et al., 1998; Joyce et al., 2004; Okamoto et al., 2005; Ushio and Kachi, 2010; Watanabe et al., 2010, 2011; Kucera et al., 2013; Hou et al., 2014; Huffman et al., 2015; Xie et al., 2017; Levizzani and Cattani, 2019; Skofronick-Jackson et al., 2019) are indispensable to advance further, since direct estimates of precipitation (pluviometers, disdrometers) and ground radars are limited to land areas. In the near future, it is likely that satellites will continue to play a vital role in validating models and therefore in opening new directions in the way key physical processes are modeled. These advances need to be parallel with an explicit account of what is empirical in models in order to benefit both fields, observations and models. Algorithm developers in the satellite realm are perhaps more used to specifying their assumptions through the Algorithm Theoretical Basis Documents (ATBD) but a full comparison between the physics and empirical values behind both algorithms and parameterizations is much needed to advance the field. On that note, it is clear that better access to climate models code would contribute to address scientific gaps in climate models and to improve their reliability (Añel et al., 2021). It would be also highly desirable that scientists not only specify the parameterizations they have used, but also the assumptions and empirical values they have actually selected within these. Tables 2-16 can be used to easily identify and pinpoint their choices. The benefit will be immense as some discrepancies could be readily attributed to known issues (i.e. heavy spurious rainfall over warm water in adjustment schemes) or identified as cofounding variables. As in the case of the microphysics,

making transparent the codes, the assumptions and the empiricisms can only benefit the community and dispel any potential concerns.

As a final comment, it is important to note that the focus of this paper is not comparing the publicly available convection schemes or to lean users towards one or another but to explore the Physics behind the modules, and to do that from an objective

and independent point of view. Neither is the paper about criticizing the simplifications that are inherent to modeling the atmosphere, or the limitations of current methods. On the contrary, the research arises from the conviction that models are the way forward to advance climate research. Being aware of the potential misuse of the results shown here to attempt discrediting models, it is important to vaccinate uninformed critics and discourage futile attempts: neither this paper nor Tapiador et al. (2019a) cast any shadow on model outputs. On the contrary, they display and celebrate the delicate intricacies, nuances, precise

measurements and careful choices made by the community to craft complex tools to forecast, simulate and predict precipitation.

**Code and data availability**

There is no code or data relevant to this paper.

**Author contributions**

Conceptualization, F.J.T. and A.V.P.; Funding acquisition, F.J.T.; Investigation, F.J.T. and A.V.P; Methodology, F.J.T. and A.V.P.; Supervision, F.J.T.; Writing – original draft, A.V.P.; Writing – review & editing, F.J.T. and A.V.P.

**Competing interests**

The authors declare that they have no conflict of interest. They have not participated in the development any existing convection module or engaged in any collaboration or discussion with their developers in order to prepare this paper. Their review is an independent, purely objective analysis based on literature and stays neutral on the suitability or performances of any of the parameterizations for any alleged purpose.

**Acknowledgements**

Funding from projects PID2019-108470RB-C21 (AEI/FEDER, UE), and CGL2016-80609-R is gratefully acknowledged. A.V.P. acknowledges support from Grant FPI BES-2017-079685 for conducting her PhD. We are grateful to two anonymous referees for their valuable comments. Special thanks are due to Peter Bechtold for kindly performing the sensitivity experiments depicted in figure 3 with the IFS model during A.V.P. research stay at ECMWF in February 2022, and for making

some observations and suggestions that certainly improved the revised version of the manuscript.

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
