# Peer review of "Empirical values and assumptions in the convection schemes of numerical models"

_Geoscientific Model Development, 2021_

## Author Comment (AC1)

**Empirical values and assumptions in the convection schemes of numerical models**

Anahí Villalba-Pradas and Francisco J. Tapiador

**Response to referees**

Referee comments shown in black
Authors' responses shown in blue

**Referee #1**

*Paper Summary:* This is a review on the convection scheme with a focus on three main elements of the parameterization: the triggering of convection, the cloud model and the closure type. It also presents an emphasis on the choice of the free parameters in those three elements of the convection parameterization. This paper is interesting and presents a complete overview of the different assumptions made for entrainment/detrainement, microphysics which are rarely discussed. However, I think that three key points should be taken into account for the paper to be accepted. First, differences in assumptions and constitution between parameterization of shallow convection and parameterization of deep convection should be highlighted and how the presence of both is taken or not into account in the triggering or closure. Second, a schematic that summarizes the main elements of a convection parameterization should be used and may help understanding the different tables. Third, the conclusion needs a re-writing to address the major challenges that convection parameterization is facing.

Thank you very much for your comments. We have modified the paper following your valuable suggestions and we believe that it has significantly improved its merit.

*Major Comments:*

I found that the common elements and differences between parameterization used for shallow convection and deep convection should be more emphasized and discussed. Right now, most examples refer to parameterization of deep convection while some of them refer to parameterization of shallow convection with no specific discussion. I see two possible options: 1/ to get rid of the examples referring to shallow convection parameterization but a discussion on the main differences could be added at the end, 2/ to end each section, by one dedicated to the shallow convection.

We have unified the style in which we refer to deep and shallow convection following your suggested option 1. Within each section, we have first referred to deep convection, followed by parameterizations that include deep and shallow convection, and finally to shallow convection. New references have been added.

There are very few illustrations which is common in a review paper. However, one schematic summarizing the main elements of a convection parameterization could be helpful.

We agree that adding such schematic would be illustrative for the reader and therefore, we have added it in section 2.

The conclusion section should be revisited. It could be organized with 1/ an historic view of the development of the convection scheme organized around the main challenges faced and 2/ a list of the remaining challenge for convection parameterization (for that you can refer to Rio et al 2019 which

listed 3 main challenges). You may also refer to Couvreux et al 2021 which propose a new methodology for combining tuning and parameterization development. 3/ a summary of the main differences between shallow and deep convection regarding trigger, cloud model and closure, the three main elements addressed in this review.

Thank you for suggesting relevant references. We have added them to the paper and we have rewritten the conclusions to include the development overview, open challenges, main differences between deep and shallow parameterizations, parameter tuning methods, and the importance of observations.

*Minor Comments:*

*Title*: I propose to change 'convection' to 'convection scheme'

We have changed the title, so now it reads "Empirical values and assumptions in the convection schemes of numerical models."

Abstract: 'Convection has to be parameterized in NWP models, GCM models and ESM models': For NWP models it depends of the resolution and the convection. For regional NWP models, most centers now use models that resolve the deep convection. I propose to moderate this sentence.

We have modified the abstract taking into account your comments as well as the comments from the other referee.

Table 1: Very long list of acronyms. Is it really useful?

We believe that the list can be helpful for the general readership of the journal.

L 105-110: on the discussion of the tuning and the error compensation, you may want to refer to Couvreux et al 2021

We have included the reference in the text as follows: "Different approaches have been proposed to avoid these issues in tuning, including the use of convection permitting models, or machine learning approaches that replace some parameterizations by neural networks (Couvreux et al., 2021). In the former approach, the high spatial and temporal resolutions of the model allow to simulate convection directly without resorting to parameterization. Couvreux et al. (2021) proposed a new method that performs a multi-case comparison between Single Cloud Models (SCM) and Large Eddy Simulation (LES) to calibrate parameterizations. The method uses machine learning without replacing parameterizations due to their important role in the production of reliable climate projections."

l133 on the convection being a major source of uncertainty you may also want to refer to Jakob 2010

We have included the reference in the text: "Convective processes have been identified as a major source of uncertainty (Jakob, 2010; NAS, 2018, hereafter decadal survey), and dedicated efforts are needed to fill the gaps in our present knowledge of the processes involved."

For section 2, it will be useful to refer to the review of Rio et al 2019 on the parameterization of convection

We have extended the opening of section 2 by adding a paragraph citing this reference.

l 169: can you explain with one sentence the CISK for the reader.

We have added the explanation: "CISK states that cyclones provide moisture that maintains cumulus clouds, and cumulus clouds provide the heat that cyclones need."

L 177: can you add a sentence explaining what 'b~0' means ? No storage in the atmosphere? Is this realistic?

It means that there is no storage of moisture in the atmosphere, which is not a realistic situation. The following sentence has been added for clarification: "This value of $b$ is not realistic as it implies that no moisture is stored in the atmosphere."

L 294-300: this discusses criteria on positive buoyancy or unstable parcels. This is not any more really a moisture convergence trigger and should be discussed.

Thank you for pointing this out. We have updated section 3.1.1 accordingly.

Section 3.1.2 you may also refer to the ALP and ALE concept detailed in Rio et al (2009), Grandpeix and Lafore (2010) or Hourdin et al (2013)

We have added a description of the ALE concept in section 3.1.2 and of the ALP concept in section 5.1.1.

Table 3: c(z)=> please check readability. Are you sure that the condition is w²<0 this should be never reached?

We have modified the typesetting to improve readability.
As for the second part of your comment, it was a mistake. The condition is that the cloud top level is the level where the vertical velocity becomes negative (then, we have downdraft). Fixed now, thanks.

Table 4: It is not really understandable like that. Try to shorten the text.

We have shortened the text in the table.

Section 4.1.2 should be improved in order to better highlight what distinguish the different elements of a spectral models. Right now this is not very clear. Also, when you mention revision of scheme, be more specific in how this revision has modified certain characteristics of the spectral models.

We have reformulated the whole section (now numbered 4.1.1) to highlight the main features of spectral models.

For ex: l 477-478 is not clear enough. (a simpler closure formulation: what has been changed? How this affect the characteristics of the spectral model => This is should be more indicated in the closure section). Similarly for l 475-477, l 479-481.

As for lines 477-478, 475-477, and 479-481 in the original paper, we have moved them to the bulk models section (now numbered 4.1.2) and explained the modifications introduced by each study.

l 549: can you detail a bit more how the Eps_turb is described with an eddy-diffusivity approach and give references.

We have added the following text: "The turbulent entrainment rate is related to the flux across the updraft boundary, which is often described with an eddy diffusivity approach (Kuo, 1962; Asai and Kasahara, 1967; De Rooy et al., 2013; Cohen et al., 2020). Under the eddy diffusivity approach, the eddy flux is modelled by a downgradient and an eddy diffusivity that for the case of the turbulent entrainment is proportional to the radial scale of a plume (used as a mixing length) and the turbulent velocity scale of the environment."

l555: suppress the 'in' before (Simpson, 1971).

Fixed. Thanks.

l 694-695: should mention that Derbyshire proposed to make the detrainment proportional to the environmental relative humidity.

We have added: "Derbyshire et al. (2011) confirmed this finding using a CRM and an adaptive detrainment proportional to the environmental relative humidity".

l 704: please recall what a precipitation efficiency is for.

We have added a brief reminder of the precipitation efficiency definition and pointed out that some authors use it as a conversion coefficient from cloud water to precipitation, while other also include the effect of re-evaporation. In this case, precipitation efficiency is the amount of precipitation reaching the surface.

l 706: change 'for the same of' to 'for the sake of'

Corrected. Thanks.

l 802-805: please rephrase, this is difficult to understand.

We have rephrased the text to improve readability: "The dCAPE closure variable was replaced by PCAPE, defined as the integral over pressure of the buoyancy of an entraining ascending parcel with density scaling. The authors defined a convective adjustment time scale following Bechtold et al. (2008). This adjustment time is defined as the product of a convective turnover time scale $\tau_c$ and empirical scaling function $f(n)$ that decreases with increasing spectral truncation. At the same time, $\tau_c$ is given by the ratio of the convective cloud depth and the vertical averaged updraft velocity. The authors stressed the dependency of $\tau_c$ with PCAPE through the velocity, which agrees with the observations in Zimmer et al. (2011). The implementation of this closure in the ECMWF IFS led to a better representation of the diurnal cycle of precipitation."

l 807: not clear what are the differences between flux-type and state-type closures stochastic closures are not mentioned

We have clarified the differences as follows: "In contrast to the previous flux-type closures, state-type closures decompose the change of CAPE-like variable into its boundary layer component and free troposphere component, instead of in its large-scale and convective component".
Stochastic closures have been moved to a dedicated section 5.1.3.

l 825 5.2.2 is before 5.2 => check the label of the different subsection

Fixed. Thanks.

I found the 'impact of closure' section not very strong; Should be improved.

We have substantially extended this section by adding more references and elaborating on the impacts of the different closure choices.

l 850, 852 is not necessary in the conclusion

We have reformulated the conclusions as suggested by the reviewer, including removal of those lines.

**Referee #2**

This study presents a review of convection schemes that have been developed for weather and climate models. An overview is given of the classic types of convection schemes as well as their individual components, such as the triggering function, the transport or cloud model, the microphysics scheme, and closure methods. Figures of results with various convection schemes as implemented in WRF for a single hurricane case are included as illustration. A strong point is this review is pretty comprehensive, also in its referencing. Quite some work has gone into collecting and describing the many variations in formulations that have been proposed in the past few decades. The tables that summarize these formulations can well function as an overview for someone new to the field.

Thanks for the comments. Please note that as stated in the title and in the abstract, the paper is a review of the empirical values and assumptions. It is not a review of convection schemes. To make explicit the choices we had to organize the paper in some way, so we followed the conventional 'convection schemes' classification. Some information is required on those to frame the discussion, but the focus is on the empirical parameters and assumptions. There are already excellent papers specifically devoted to the taxonomy of the convection schemes. The paper is exactly what referee #1 says, "(the paper) presents a complete overview of the different assumptions made for entrainment/detrainement, microphysics which are rarely discussed."

That said, after reading the manuscript I could identify a few significant shortcomings. Firstly, the overview of convection schemes overly focuses on rather classic approaches, and fails to cover some important new developments in the field. These include recently proposed unified approaches, new concepts that address the ever more important grey zone problem, and new schemes capable of representing convective organization and memory. Not fully covering these recent developments, which result from active and intense ongoing research, is a serious omission. In my view this is detrimental for the relevance of this paper for the science community. The manuscript feels rather outdated in that sense.

We have now added several sections covering the most relevant empirical values and assumptions in recent developments in convection schemes.

A second concern, and related to this point, is that the science objectives of this study are not clearly defined. What exactly is the purpose of this review? This remains unclear.

We cannot agree. It is a review on the empirical values and assumptions. It was already stated in the original manuscript, but to stress it even more clearly, we have rewritten the objectives as follows: "The goal of the present paper is to provide a comprehensive account of the empirical choices and assumptions behind the representation of convective precipitation in models. To the best of our knowledge, there is no such extensive review of the empirical values and assumptions in the convection schemes available in the literature."

Third, the main conclusions of this study (as stated in the abstract) are not objectively supported by the content that is presented. This mainly concerns the impacts of convection schemes on weather and climate, and the need for observational datasets.

We address this comment below in the General comments section.

Fourth, the introduction contains statements that are not true, while the introduction is also overly skewed towards motivation and does not explain what is unique about this particular study.

We have modified the introduction to better explain the contributions of this study. We have revised the text and made sure that it does not contain any statements that are not true.

Finally, the figures included are not fully described and explained.

*The figures are merely illustrative of the differences between various choices. More below.*

To summarize, although I do see the use of another review of convection schemes, the current version is incomplete, somewhat outdated and should be improved at various points.

*As explained before, this is not another review on convection schemes but a review on the empirical values and assumptions made in those schemes. We ensured that this is clear to the reader by modifying the relevant part in the introduction. The manuscript is titled "Empirical values and assumptions in the convection schemes of numerical models", which also indicates the purpose of this work.*

For these reasons I recommend major revision of this manuscript to address these issues, before it can be accepted for publication.

*Thank you for suggesting a revision of the manuscript. We have carefully addressed all comments of both referees and we believe that the manuscript has been sufficiently improved for its acceptance.*

*General comments:*

1. Convective parameterization is a science field that sees active development at themoment. This is driven among others by the realization that these schemes continue to cause significant uncertainty in climate predictions, and also by the shift towards higher resolutions that become feasible in state-of-the-art weather and climate modeling. While these developments are briefly mentioned at various points in the paper, this is not reflected in its organization and structure. Recently proposed new types of convection schemes which show promise in addressing these issues and are "breaking the parameterization deadlock" (Randall et al., 2003) are not discussed in a structural way. These include i) unified parameterizations based on the EDMF approach (Siebesma et al, 2013, and many follow-up papers about EDMF), ii) PDF-based schemes (e.g. Golaz et al., 2003; Larson et al, 2012), iii) schemes that are scale-aware, scale-adaptive and also introduce stochasticity due to subsampling of a convective population inside a GCM gridbox at higher resolutions (Honnert et al, 2020), and iv) approaches that successfully capture convective memory and spatial organization. Not explicitly covering and describing these new approaches, which are now in the stage of becoming widely adopted in operational models, makes the current version of the manuscript not reflective enough of the state of the art in this research field. In that sense the manuscript is somewhat outdated, and repetative of previous reviews of convective parameterization that have already covered the classic schemese and approaches in great detail. To make the review better reflect these new approaches I strongly recommend giving them their own category / subsection in the organization of the manuscript.

*The topic of the review has been already discussed. Section 2 serves as a framework for the rest of the paper, but the focus is not on each convection scheme but on the empirical values and assumptions used. We agree that including more recent studies is beneficial for the completeness of the review. Therefore, we have added a description of the current developments in section 2.6 on PDF-based schemes, section 2.7 on unified models, section 2.8 on scale-aware and scale-adaptive models, and section 2.9 on models accounting for convective memory and spatial organization. Empirical values and assumptions used in these schemes have been added to various tables throughout the paper.*

On a related note, while the bibliography is indeed extensive, I noticed that at some points the referencing is not accurate. Sometimes the first "breakthrough" studies that proposed a new concept are not referred to, but instead only a somewhat arbitrary selection of follow-up studies are mentioned. One example are new mass flux schemes as mentioned in section 1.2, page 7, of which only one very recent example is mentioned. Another example is the description of spectral mass flux models (section 4.1.2), of which many more have recently been proposed and explored (e.g. Wagner and Graf, 2010; Neggers, 2015; Suselj et al, 2013; Olson et al, 2016; Brast et al., 2018; Hagos et al, 2018). In my opinion the

review is by far not complete at these points, which is a serious omission. I recommend going through all sections again and to make sure that key groundbreaking studies (both classic and recent) are properly mentioned.

Thank you for this comment. We carefully went through all the references in our manuscript and we have replaced or added references to ensure that the most relevant studies are cited.

2. The introduction misses a clear statement of the main science objectives of this study. Is this paper meant to be just a review, or more? Both the introduction and the abstract are misleading in this respect. While the introduction elaborately emphasizes the importance of convection schemes and the need for their careful evaluation, it fails to clearly state that this paper is meant to be a review of all existing schemes out there. It is also not clearly mentioned what is new compared to previous reviews of this kind, how this particular review can help the community. I recommend adding clear statements about both the objectives and the novel aspects in the introduction.

We have revised the abstract and the introduction to clearly stress the contributions. In the abstract, we state: "These empirical values and assumptions are rarely discussed in the literature. The present paper examines these choices and their impacts on model outputs and emphasizes the importance of observations to improve our current understanding of the physics of convection. The focus is mainly on the empirical values and assumptions used in the activation of convection (trigger), the transport and microphysics (commonly referred to as the cloud model) and the intensity of convection (closure)." We have modified the second paragraph in section 1: "The goal of the present paper is to provide a comprehensive account of the empirical choices and assumptions behind the representation of convective precipitation in models. To the best of our knowledge, there is no such extensive review of the empirical values and assumptions in the convection schemes available in the literature." We believe that the objectives are fully explained in the abstract and in the introduction of our manuscript.

3. The abstract announces conclusions that are not objectively or adequately supported bythe contents of this study. This concerns i) the examination of impacts of choices of parameters in convection schemes on weather and climate, and ii) insights concerning the need for observational datasets for constraining these choices.

The core sections in the paper, i.e., sections 3 to 5, include a dedicated subsection where the impact of the empirical values and assumptions of each parameter is presented based on the results from the literature. As for the need of observations, we refer to numerous convective parameterizations that were developed based on observations, e.g., the Bett-Miller-Janjic scheme, or the Kain and Fritsch scheme. In addition, we present examples of observations that led to modifications in the parameterizations to better represent convective processes, e.g., the observations of Niziol et al. (1995) that yielded a modification of the minimum cloud-depth threshold in Kain and Fritsch (1993). We have moved the latter from the conclusions to section 3.1.4. To further support the claims stated in the abstract, we have added more references highlighting the importance of observations in improving convective schemes. For example, in section 3.3: "Using the GreenOcean Amazon (GOAmazon, Martin et al., 2016), the authors set the values for the dCAPE threshold and entrainment rate from 2014. The new values are $55\,\mathrm{J\,kg^{-1}s^{-1}}$ for the dCAPE threshold and $2.5 \cdot 10^{-4}\mathrm{m^{-1}}$ for the entrainment rate." Numerous statements like this have been added throughout sections 3 to 5.

Concerning the impacts of parameter choices, no new analyses are included that demonstrate this impact in a statistically significant way. As is usually the case in a review paper, this review refers to previous studies for describing such impacts. However, in my experience these impacts of parameter choices are extremely code-specific, and are not universal. It is often the case that errors in calibration of one parametric component are hidden by errors in another; a good example is the too-few, too-bright problem (e.g. Nam et al., 2014) or errors in cloud overlap masking errors in vertical structure (Neggers and Siebesma, 2013). This means that results from parameter studies with one GCM code do not necessarily translate to another. This danger is not mentioned, but is very relevant for the conclusions drawn in this paper. This should be discussed.

We agree that such issue should be discussed. It is now included in section 6: "However, the impacts of the empirical values in convection are extremely code-specific and often errors in calibration of one parameter are hidden by errors in another. Examples of these include masking errors in vertical structure due to errors in cloud overlap (Neggers and Siebesma, 2013) or the too-few, too-bright problem (e.g., Nam et al., 2014). Therefore, results obtained in one GCM with a particular set of empirical values might differ from results obtained in a different GCM with the same set of empirical values."

It is also not evident from the content how exactly observations can help in constraining parameter choices, yet is is presented as a major conclusion in the abstract. That convection schemes have many constants that need constraining is not a new insight. Exclusively using observations for this purpose is problematic, due to data gaps and instruments not being capable of sampling key variables in parametric equations. Various recent international efforts have been conducted to make progress in this respect, in the form of field campaigns (e.g. EUREC4A) or long-term deployment of instrumentation at meteorological supersites (e.g. Neggers et al, 2012; Song et al., 2013; Gustafson et al., 2016; Zheng et al, 2020). A thorough discussion of both the problems and opportunities for constraining parameter choices in convection schemes with modern observations is missing, but is needed to support this conclusion. I recommend adding a section on this topic if this conclusion is to be maintained; if not, then I would remove this conclusion from the paper.

We do not agree. We state that observations are needed to improve the current understanding of the physics of convection. Besides, many convective parameterizations were developed based on observations and many of the parameters in convective schemes use values based on observations such as BOMEX or ATEX field campaigns, among others. Modifications of these schemes usually come from comparisons of the scheme simulations with observational data. While we are convinced that this discussion is not of sufficient importance as to create a dedicated section, we have included the following text in section 6: "However, observations suffer from data gaps and the instruments used are not able to sampling key variables in parametric equations. Long-term instrumentation deployment at meteorological supersites (Neggers et al., 2012; Song et al., 2013; Gustafson et al., 2020; Zheng et al., 2021) or field campaigns (e.g. EUREC4A) have been conducted to alleviate these issues".

4. The introduction contains statements that are factually wrong. Precipitation in general circulation models is not just generated by convection schemes and/or microphysics schemes. Boundary layer schemes can also contribute to both convective and stratiform precipitation, and significantly so. In fact, precipitation in subtropical marine low level stratocumulus and trade wind shallow cumulus is often completely carried by the boundary layer scheme. This error should be corrected.

To our knowledge, most boundary layer schemes contribute to precipitation indirectly. Indeed, boundary layer schemes widely used in community models alter the fluxes and then the water content ($q$) and the cloud cover. This subsequently affects precipitation, but precipitation is prognosed in the microphysics and in the convection scheme. There is no precipitation output from the boundary layer schemes, but moisture and moisture tendencies. Thus, for instance in the ECMWF model: "Eddy-Diffusivity Mass-Flux (EDMF) framework is applied, to represent the non-local boundary layer eddy fluxes (Koehler et al. 2011). The scheme is written in moist conserved variables (liquid static energy and total water) and predicts total water variance. A total water distribution function is used to convert from the moist conserved variables to the prognostic cloud variables (liquid/ice water content and cloud fraction), but only for the treatment of stratocumulus. Convective clouds are treated separately by the shallow convection scheme." However, if the referee is aware of any boundary layer scheme producing precipitation directly, we will be happy to include it in the paper. As for now, we have modified the introduction as follows: "Numerical Weather Prediction models, Global Climate Models, and Earth System Models (NWP, GCMs, and ESMs) generate precipitation **mainly** through two parameterizations: microphysics of precipitation (MP hereafter) and cumulus parameterization (CP) schemes"

More generally speaking, this point also relates to my first comment, in that new unified schemes have recently been proposed that are less strict in the separate representation between shallow and deep convection, and between convective and stratiform rain. These unified schemes are not properly described. I recommend improving the text in the introduction, and the manuscript as a whole, to more precisely and accurately describe the reality in state-of-the-art general circulation models.

This comment has been already addressed. We have added a section on unified schemes. In addition, we have included the relevant empirical values and assumptions used in these schemes.

5. Two figures are included in the manuscript that are meant to illustrate the impact of convective parameterization on the representation of convective phenomena in a circulation model. To this purpose WRF simulations of a tropical cyclone are used. However, at the points in the manuscript that these figures are referred to they are not properly explained. What is the model setup, what are the boundary conditions and forcings, and which data is used to derive these? Because these details are absent, these results can not be independently reproduced. I recommend adding an appendix in which these simulations are adequately described.

As the referee states, the figures are illustrative. Therefore, a detailed list of parameters is not included. In contrast to, e.g., validation studies, we do not expect that the readers would be interested in reproducing these simulations. Nevertheless, we have extended the simulation details in the figure captions.

*Detailed comments:*

p1, title: "... in the convection of numerical models". This does not make sense. I guess you mean "in the representation of subgrid-scale convection in circulation models", or something similar?

We have changed the title to "Empirical values and assumptions in the convection schemes of numerical models" accounting for the suggestion by the other referee.

p1, abstract: The first half (three sentences) is a general description of what convective parameterization is, and why it is important. As this is already generally known and does not reflect the particular contents of this study, I recommend to remove this part and instead add a few lines about what is new about this review compared to previous ones, its science goals, and what aspects are unique and worth remembering.

We consider it common to begin the abstract by a general introduction of the topic, similarly as in many papers in the field. It is known to people specifically working in convective parameterizations, but not to the general readership of the journal. However, we have modified the abstract by emphasizing the contribution of this review and taking into account also the comments from the other referee.

p5, line 64: "... generate precipitation through two parameterizations: microphysics... and cumulus parameterizations". This is not true: boundary layer schemes formulated in moist conserved variables also produce significant precipitation. See my 4th main comment above.

This comment has been addressed above.

p5, line 67: "... are intended ...". Intended by who? Please be specific.

We have rephrased the sentence.

p5, line 77: "biota". What does this mean?

It is the animal and plant life of a particular region, habitat, or geological period. We believe that this is commonly known and does not need to be explained to the readers. Anyway, the meaning follows from the next sentence.

p7, line 123: please add the key paper by Wyngaard (2004) as well as the recent review paper by Honnert et al (2020) to the list of references about the convective grey zone. See also my 1st main comment above. Also, "grey scales" is a term I have not come across before; I suggest to stick to "grey zone"

We have added the relevant references. Thank you for your comment about gray scales. We have corrected this mistake.

p7, line 129: Why only refer to this paper for scale-aware parameterizations? Many more have been proposed by now, also much earlier than the study cited here. Please perform more thorough literature research on scale-aware parameterization, and add the most relevant papers on this topic here.

We have added more references to the scale-aware parameterization.

p8, line 136: "latest decadal survey". Do you mean the NAS one mentioned in the previous paragraph? This is not clear; also, not everyone is familiar with this survey.

We have modified the paragraph to make it clearer: "Convective processes have been identified as a major source of uncertainty (Jakob, 2010; NAS, 2018, hereafter decadal survey), and dedicated efforts are needed to fill the gaps in our present knowledge of the processes involved.
Owing to the influence of convection on climate and weather events over a large range of spatial and temporal scales, one of the most important objectives of the latest decadal survey is to improve the predictions of the timing and location of convective storms, and their evolution into severe weather. Besides the drawbacks associated with the spatial resolution, the multiscale interactions leading to the organization and evolution of convective systems are difficult to observe and represent."

p8, line 143: "crudely". Simple parameterizations are not necessarily crude. This is not scientific language, and is also somewhat insulting towards scientists who have spend significant effort in developing and implementing such parameterizations in GCMs.

This word has been used before when referring to convective parameterizations (e.g., IFS documentation, Cy47r1 Operational implementation, ECMWF, 30 June 2020). However, we have modified the sentence: "…described with simple parameterizations…".

p8, line 146: "computing-intensive": do you mean that convective parameterization is always computationally demanding? Then please explain why this is the case.

What we meant here is that for global climate simulations at high resolution and for long time periods, e.g., a century, the required computing time would be very high if no convective parameterization is used. The sentence has been changed: "While models will likely increase their nominal resolution in the next decade, it is also likely that global, century-long simulations from multi-ensembles under different assumptions will need to resort to parameterizing convection to reduce the computational burden."

p8, line 152: ".. Fig. 2". This is the first time the figures are referred to, and would be a good point to explain what exactly we see in them, and why this is relevant in this context. See also my 5th main comment above: these simulations are not adequately described, and can not be independently reproduced. And why is testing convection schemes for a cyclone the most relevant? Usually convective parameterizations are tested for simpler cases (such as locally forced continental convection) in a

simpler setting (e.g. Single Column Models combined with a weak temperature gradient approach). Please explain these choices.

We have added the following sentence before the figure appears in the text: "Indeed, as shown in Fig. 2 for the 6-hours total accumulated precipitation for Typhoon Chaba, even today model outputs look different depending on the cumulus parameterization used." Both Fig. 2 and 3 are just illustrative to show that the simulation results are different depending on the convection scheme used. We have also added more information on how the simulations in both figures were performed, as already mentioned in an answer to a previous comment. As for simulating a cyclone, we selected this case due to the societal relevance and deeper physical complexity of this type of atmospheric events compared to idealized cases.

p10, line 160: "between cumulus clouds". Parts of convective updrafts are non-cloudy, yet in this state still contribute significantly to the total vertical transport. Please change your labels such as "cloud model" to account for this.

We have changed this sentence: "The main assumptions in convective parameterizations concern the trigger model, the transport and microphysics, commonly referred to as the cloud model in classical convection schemes, and the closure of the scheme (Fig. 3)."

p10, line 169: Please explain what CISK means.

We have added the explanation: "CISK states that cyclones provide moisture that maintains cumulus clouds, and cumulus clouds provide the heat that cyclones need."

p10, line 190: "no cloud models are needed". I would phrase this differently, because the cloud model is still needed; but it is now hidden in the adjustment procedure.

We have changed this sentence for: "the physical properties of clouds are implicit and no cloud model has to be explicitly specified."

p11, lines 194-195: "very large precipitation... rarely observed in nature". No reference is provided to back up this statement; please cite a paper that shows this.

We have added a reference to the paper of Emanuel and Raymond (1993), who mentioned this.

p11, line 200 and onwards: For clarity, I suggest to include a simple equation for the adjustment, including a time scale. Different forms of adjustment are possible (e.g. Newtonian); can all be described by the same equation?

We have added the adjustment equations described in Betts (1986) for the adjustment of temperature and moisture towards reference profiles.
As for the different forms of adjustment, we are not sure what the referee means.

p12, equation (1): These equations are not universally applicable to all mass flux schemes. Some are formulated in terms of conserved variables for moist adiabatic motion. So either it should be mentioned that (1) is only applicable to a subset of mass flux schemes, or the formulation should be changed to make it more generally applicable.

We have changed the whole section 2.3 including the formulation of the mass flux equations, which are now more general.

p12, line 241: "single entraining plume". Is this a steady state plume? Please explain. Also, I would add some references to the first classic papers about the rising plume model, such as Simpson and Wiggert (JAS, 1969).

Eq. (2) was not a steady-state plume as $\partial/\partial t$ was included in the formulation. Please note that this equation no longer appears in the manuscript as we have replaced it by more general expressions (Eq. (5.1), Eq. (5.2) and Eq. (5.3)). These equations are not steady-state either but we have added a sentence explaining that "numerous schemes have been proposed since then mostly using the steady state assumption, i.e., $\partial/\partial t = 0$".

A relatively recent review study about the rising plume model and its parameter choices that to my great surprise is not mentioned at all in this paper is the one by De Roode et al. (2012). How do the values mentioned in that paper overlap with those summarized in the Tables in this paper? See also my 1st comment about adequate referencing.

We have rewritten section 2.3 as mentioned before and we have added the reference suggested by the referee. As for the values in De Roode et al. (2012) and those in our tables, they overlap in the definition of entrainment rates (e.g., Gregory, 2001) as well as in the velocity condition to trigger convection (e.g., Jakob and Siebesma, 2003), to find cloud top (e.g., Bechtold et al., 2004), and convective closures defined in terms of the updraft vertical velocity (e.g., Neggers et al., 2009).

p12, line 244: ".. the i-th cumulus cloud". Is this a single cloud, or a sub-ensemble of clouds, such as clouds of a certain size or strength? Please define.

As Eq. (2) is expressed using the subscript $i$, it is valid for a sub-ensemble of clouds. For one single cloud the subscript $i$ would not appear in the formulation. However, this equation has been replaced by Eq. (5.1), Eq. (5.2) and Eq. (5.3), as already mentioned before. These expressions do not include the subscript $i$ and thus they refer to a single cloud.

p12, last line. The model described by equations (1) and (2) does not cover various new mass flux approaches, such as EDMF or pdf schemes. These have successfully been applied to precipitating convection, and can actually deal with scale adaptivity. See also my 1st main comment above.

Eq. (1) and Eq. (2) have been replaced by more general ones, as explained before.

p13, line 252: "produce too little heavy rain and too much light rain". Do all convection schemes suffer from this, or just one implementation in one particular GCM? See also my 3d main comment above.

We have modified the reference on this line to include studies that reflect this issue. For example, Sun et al. (2006) analyzed precipitation characteristics in eighteen coupled models, each of them with their own convective parameterizations. In terms of pattern and frequency, most models overestimated light precipitation frequency (1-10 mm/day) and underestimated heavy precipitation intensity (> 10 mm/day). It is worth noting that two models (PCM and CGM3.1) using the same parameterizations for stratiform and convective precipitation produced different values of mean JJA light and heavy precipitation frequency, especially over Africa, and of light and heavy precipitation intensity, especially over Africa (light) and South America (heavy). Therefore, results are code-specific as the referee pointed out in his/her third main comment.

p13, line 253: "Pritchard et al, 2011": I would also refer to Guichard et al. (2004), who first properly documented this behavior and which paper was also published 7 years earlier (which is a long time in science). See also my 1st comment about adequate referencing.

We have changed the references in this part and added Guichard et al. (2004) as suggested by the referee.

p16, line 350: "widely used at ECMWF". What do you mean with "widely". Please add a reference to a study that shows this.

By "widely" we meant that the temperature perturbation trigger in a form similar to the one described in Bechtold et al. (2001) has been used in many versions of the ECMWF IFS model (Bechtold et al., 2004, 2008, and IFS documentation: https://www.ecmwf.int/en/publications/ifs-documentation). We have updated the manuscript accordingly.

p22, line 445: "convective memory". This topic is intensely researched at the moment, yet is only briefly mentioned here. I think it deserves much more attention, even its own section. Doing so would make this review paper a lot more relevant and up to date.

Thank you for this comment. We have now included section 2.9 about convective memory and spatial organization as suggested.

p27, concerning the discussion on entrainment: I think it makes sense in this discussion to also refer to these recent papers, also in the table:
https://doi.org/10.1029/2019JD030889, https://doi.org/10.1175/JAS-D-20-0377.1. Apart from providing new insights, they also use observational data to constrain entrainment rates, which is relevant for this review

We have added the references suggested by the referee to our discussion on entrainment and detrainment rates in section 4.2.2 as well as in Table 5.

---

## Referee Report (RR1)

**Review of revised GMD manuscript https://doi.org/10.5194/gmd-2021-61**

The authors have done well in improving the manuscript at various points, in response to the feedback provided by both reviewers. This includes a more complete coverage of all types of convection schemes that have been proposed, including more modern and unified approaches. The authors have also put convincing effort into getting the references right, where in the first submission a few key publications were ignored. All of this has made the paper more complete and concise, which is recommendable. As I already stated in my first review, I really do appreciate the significant amount of work that has gone into scanning all convection schemes and summarizing their essential assumptions and settings.

That said, some of my main concerns have not been adequately addressed. A few specific points I raised and some questions I asked remain unanswered, or were side stepped in the response. These still open issues, which are also important, are summarized below. I remain of the opinion that these concerns need to be adequately addressed before publication is possible.

In some scientific journals a failure to address major concerns first time round automatically leads to rejection. I would still recommend a major revision, mainly because I do see merit in this work. So I leave that decision to the editor.

**Main concerns**

1) In response to my first major comment, and at various other points, the authors state that "Please note that as stated in the title and in the abstract, the paper is a review of the empirical values and assumptions. It is not a review of convection schemes". I fully disagree, for the following simple reason: these (parametric) assumptions are the defining parts of convection schemes, and what makes them differ from each other. This implies that one cannot separate the two. When the objective is to provide a review of empirical assumptions, then this in effect comes down to reviewing (differences between) convection schemes. This might be a disagreement on semantics. Still, it is important to clarify this in the manuscript, to avoid any confusion with the reader (including myself).

I also disagree that this review is the first of its kind ever, as for example stated in the introduction (line 75, "To the best of our knowledge, there is no such extensive review..."). I know of at least one previous study. De Roode et al (2012, doi.org/10.1175/MWR-D-11-00277.1) discusses empirical assumptions and values as feature in the updraft kinetic energy equation, and includes a thorough literature review. In structure and content, their Table 1 is very similar to, say, Table 6 on entrainment rates in this manuscript (among others). For this reason I think this statement should be softened, to properly acknowledge previous work.

2) In my second main comment I asked to provide a clear statement of what is the overarching science objective / higher goal of this review, or in other words, what is the added value of this review. The response is as follows: "The goal of the present paper is to provide a comprehensive account of the empirical choices and assumptions behind the representation of convective precipitation in models." But this is not an answer to my question. I ask what we learn from reviews like this. Is it just a collection of long tables with many values and references, acting as a library index? Or does it yield new insights? This remains unclear, also in the revised version. Most scientific review studies provide a vision like this, so I was expecting this as a reader.

3) The response provided does not adequately address my concern. The response is: "... we refer to numerous convective parameterizations that were developed based on observations, ...", and "We state that observations are needed to improve the current understanding of the physics of convection". All convection schemes are based on at least a few observations; that is common knowledge, and not my point. Instead, my question is what your tables can tell us about what more we need in terms of observations to make progress, for example to break the ongoing "parameterization deadlock" (Randall et al, BAMS, 2003). Has the use of observations by the convective modeling community so far sufficient? Or do we need to find new ways to adequately constrain assumptions and calibrate parameterizations, in a statistically significant way? And if so, how can we most efficiently use modern extensive big datasets to this purpose? Having put so much work into delving through all these schemes in detail, and listing all the key components (which I find really impressive), you are now in a unique position to make a statement about that. The reader expects that vision, and accordingly, I thoroughly recommend adding it. Not doing so is an omission. Hence my advice to add a section dedicated to this topic. This advice still stands.

4) Judging from the response, I think there is some confusion about what is meant by "boundary layer scheme". This is not always the same in each model. Some interpret the boundary layer as only representing dry (non-saturated) turbulence and convection; others consider cloud layers as intrinsic part of the boundary layer, thus including shallow cumulus and stratocumulus. So to avoid unnecessary confusion with the reader, I recommend to clearly define early on in the manuscript what exactly is meant by "boundary layer scheme", and then to consistently use this definition throughout the manuscript. This template may sometimes not be applicable to more unified schemes, in which microphysics, shallow transport and deep transport are interwoven and can not be strictly separated anymore into unique and single modules, as was classicaly done.

That said, I know of quite a few boundary layer schemes that do generate precipitation. For example, in contrast what you say, the IFS EDMF scheme makes use of plume equations that do include a source/sink term representing precipitation. See IFS documentation C47R3 chapters 3.2 and 6.3.1. So the EDMF scheme does produce rain in case the EDMF plume condensates. Second, when the IFS Tiedtke scheme is in shallow cumulus mode, it is in effect generating boundary layer precipitation, and can thus be classified as a "boundary layer scheme". This rain can be significant, as we have learned from field campaigns on Trade wind cumulus such as RICO and EUREC4A.

The IFS scheme is just one example; there are more boundary layer schemes that directly generate precipitation. The EDMF scheme of Neggers (2009, doi.org/10.1175/2008JAS2636.1) also produces rain. The CLUBB  scheme as implemented in CAM (Larson et al., GMD, doi.org/10.5194/gmd-8-3801-2015) also generates precipitation when in boundary layer mode; see their Section 2.4 and Fig. 1.

5) "A detailed list of parameters is not included". I do not understand; which parameters do you mean? In the figures? In my opinion, all aspects of figures should be fully explained in a scientific publication, even if they are just meant to be illustrative. This is just good scientific practice: all science should be reproducable, otherwise it is meaningless.

I also find new Figure 3 somewhat simplistic. For example, it depicts shallow convection as exclusively non-precipitating, which by now we now is totally untrue (see the many studies based on RICO, NARVAL and EUREC4A data and simulations). Second, it conforms to the old idea of how convection should be modeled, using a single bulk plume and a modular approach. The schematic certainly does not accommodate unified or spectral approaches in modeling convection. See for example Fig. 1 in Arakawa and Schubert (1974), which is a much more realistic example of

how a convective population works. If this review is to be comprehensive, as is claimed in the introduction, the figure should accommodate all approaches, not just the classic bulk one.

---

## Author Response (AR4)

**Empirical values and assumptions in the convection schemes of numerical models**

Anahí Villalba-Pradas and Francisco J. Tapiador

**Response to referees**

Referee comments shown in black
Authors' responses shown in blue

**Referee #1**

*Paper Summary:* This is a review on the convection scheme with a focus on three main elements of the parameterization: the triggering of convection, the cloud model and the closure type. It also presents an emphasis on the choice of the free parameters in those three elements of the convection parameterization. This paper is interesting and presents a complete overview of the different assumptions made for entrainment/detrainement, microphysics which are rarely discussed. However, I think that three key points should be taken into account for the paper to be accepted. First, differences in assumptions and constitution between parameterization of shallow convection and parameterization of deep convection should be highlighted and how the presence of both is taken or not into account in the triggering or closure. Second, a schematic that summarizes the main elements of a convection parameterization should be used and may help understanding the different tables. Third, the conclusion needs a re-writing to address the major challenges that convection parameterization is facing.

Thank you very much for your comments. We have modified the paper following your valuable suggestions and we believe that it has significantly improved its merit.

*Major Comments:*

I found that the common elements and differences between parameterization used for shallow convection and deep convection should be more emphasized and discussed. Right now, most examples refer to parameterization of deep convection while some of them refer to parameterization of shallow convection with no specific discussion. I see two possible options: 1/ to get rid of the examples referring to shallow convection parameterization but a discussion on the main differences could be added at the end, 2/ to end each section, by one dedicated to the shallow convection.

We have unified the style in which we refer to deep and shallow convection following your suggested option 1. Within each section, we have first referred to deep convection, followed by parameterizations that include deep and shallow convection, and finally to shallow convection. New references have been added.

There are very few illustrations which is common in a review paper. However, one schematic summarizing the main elements of a convection parameterization could be helpful.

We agree that adding such schematic would be illustrative for the reader and therefore, we have added it in section 2.

The conclusion section should be revisited. It could be organized with 1/ an historic view of the development of the convection scheme organized around the main challenges faced and 2/ a list of the remaining challenge for convection parameterization (for that you can refer to Rio et al 2019 which

listed 3 main challenges). You may also refer to Couvreux et al 2021 which propose a new methodology for combining tuning and parameterization development. 3/ a summary of the main differences between shallow and deep convection regarding trigger, cloud model and closure, the three main elements addressed in this review.

Thank you for suggesting relevant references. We have added them to the paper and we have rewritten the conclusions to include the development overview, open challenges, main differences between deep and shallow parameterizations, parameter tuning methods, and the importance of observations.

*Minor Comments:*

*Title*: I propose to change 'convection' to 'convection scheme'

We have changed the title, so now it reads "Empirical values and assumptions in the convection schemes of numerical models."

Abstract: 'Convection has to be parameterized in NWP models, GCM models and ESM models': For NWP models it depends of the resolution and the convection. For regional NWP models, most centers now use models that resolve the deep convection. I propose to moderate this sentence.

We have modified the abstract taking into account your comments as well as the comments from the other referee.

Table 1: Very long list of acronyms. Is it really useful?

We believe that the list can be helpful for the general readership of the journal.

L 105-110: on the discussion of the tuning and the error compensation, you may want to refer to Couvreux et al 2021

We have included the reference in the text as follows: "Different approaches have been proposed to avoid these issues in tuning, including the use of convection permitting models, or machine learning approaches that replace some parameterizations by neural networks (Couvreux et al., 2021). In the former approach, the high spatial and temporal resolutions of the model allow to simulate convection directly without resorting to parameterization. Couvreux et al. (2021) proposed a new method that performs a multi-case comparison between Single Cloud Models (SCM) and Large Eddy Simulation (LES) to calibrate parameterizations. The method uses machine learning without replacing parameterizations due to their important role in the production of reliable climate projections."

l133 on the convection being a major source of uncertainty you may also want to refer to Jakob 2010

We have included the reference in the text: "Convective processes have been identified as a major source of uncertainty (Jakob, 2010; NAS, 2018, hereafter decadal survey), and dedicated efforts are needed to fill the gaps in our present knowledge of the processes involved."

For section 2, it will be useful to refer to the review of Rio et al 2019 on the parameterization of convection

We have extended the opening of section 2 by adding a paragraph citing this reference.

l 169: can you explain with one sentence the CISK for the reader.

We have added the explanation: "CISK states that cyclones provide moisture that maintains cumulus clouds, and cumulus clouds provide the heat that cyclones need."

L 177: can you add a sentence explaining what 'b~0' means ? No storage in the atmosphere? Is this realistic?

It means that there is no storage of moisture in the atmosphere, which is not a realistic situation. The following sentence has been added for clarification: "This value of $b$ is not realistic as it implies that no moisture is stored in the atmosphere."

L 294-300: this discusses criteria on positive buoyancy or unstable parcels. This is not any more really a moisture convergence trigger and should be discussed.

Thank you for pointing this out. We have updated section 3.1.1 accordingly.

Section 3.1.2 you may also refer to the ALP and ALE concept detailed in Rio et al (2009), Grandpeix and Lafore (2010) or Hourdin et al (2013)

We have added a description of the ALE concept in section 3.1.2 and of the ALP concept in section 5.1.1.

Table 3: c(z)=> please check readability. Are you sure that the condition is w²<0 this should be never reached?

We have modified the typesetting to improve readability.
As for the second part of your comment, it was a mistake. The condition is that the cloud top level is the level where the vertical velocity becomes negative (then, we have downdraft). Fixed now, thanks.

Table 4: It is not really understandable like that. Try to shorten the text.

We have shortened the text in the table.

Section 4.1.2 should be improved in order to better highlight what distinguish the different elements of a spectral models. Right now this is not very clear. Also, when you mention revision of scheme, be more specific in how this revision has modified certain characteristics of the spectral models.

We have reformulated the whole section (now numbered 4.1.1) to highlight the main features of spectral models.

For ex: l 477-478 is not clear enough. (a simpler closure formulation: what has been changed? How this affect the characteristics of the spectral model => This is should be more indicated in the closure section). Similarly for l 475-477, l 479-481.

As for lines 477-478, 475-477, and 479-481 in the original paper, we have moved them to the bulk models section (now numbered 4.1.2) and explained the modifications introduced by each study.

l 549: can you detail a bit more how the Eps_turb is described with an eddy-diffusivity approach and give references.

We have added the following text: "The turbulent entrainment rate is related to the flux across the updraft boundary, which is often described with an eddy diffusivity approach (Kuo, 1962; Asai and Kasahara, 1967; De Rooy et al., 2013; Cohen et al., 2020). Under the eddy diffusivity approach, the eddy flux is modelled by a downgradient and an eddy diffusivity that for the case of the turbulent entrainment is proportional to the radial scale of a plume (used as a mixing length) and the turbulent velocity scale of the environment."

l555: suppress the 'in' before (Simpson, 1971).

Fixed. Thanks.

l 694-695: should mention that Derbyshire proposed to make the detrainment proportional to the environmental relative humidity.

We have added: "Derbyshire et al. (2011) confirmed this finding using a CRM and an adaptive detrainment proportional to the environmental relative humidity".

l 704: please recall what a precipitation efficiency is for.

We have added a brief reminder of the precipitation efficiency definition and pointed out that some authors use it as a conversion coefficient from cloud water to precipitation, while other also include the effect of re-evaporation. In this case, precipitation efficiency is the amount of precipitation reaching the surface.

l 706: change 'for the same of' to 'for the sake of'

Corrected. Thanks.

l 802-805: please rephrase, this is difficult to understand.

We have rephrased the text to improve readability: "The dCAPE closure variable was replaced by PCAPE, defined as the integral over pressure of the buoyancy of an entraining ascending parcel with density scaling. The authors defined a convective adjustment time scale following Bechtold et al. (2008). This adjustment time is defined as the product of a convective turnover time scale $\tau_c$ and empirical scaling function $f(n)$ that decreases with increasing spectral truncation. At the same time, $\tau_c$ is given by the ratio of the convective cloud depth and the vertical averaged updraft velocity. The authors stressed the dependency of $\tau_c$ with PCAPE through the velocity, which agrees with the observations in Zimmer et al. (2011). The implementation of this closure in the ECMWF IFS led to a better representation of the diurnal cycle of precipitation."

l 807: not clear what are the differences between flux-type and state-type closures stochastic closures are not mentioned

We have clarified the differences as follows: "In contrast to the previous flux-type closures, state-type closures decompose the change of CAPE-like variable into its boundary layer component and free troposphere component, instead of in its large-scale and convective component".
Stochastic closures have been moved to a dedicated section 5.1.3.

l 825 5.2.2 is before 5.2 => check the label of the different subsection

Fixed. Thanks.

I found the 'impact of closure' section not very strong; Should be improved.

We have substantially extended this section by adding more references and elaborating on the impacts of the different closure choices.

l 850, 852 is not necessary in the conclusion

We have reformulated the conclusions as suggested by the reviewer, including removal of those lines.

**Referee #2**

This study presents a review of convection schemes that have been developed for weather and climate models. An overview is given of the classic types of convection schemes as well as their individual components, such as the triggering function, the transport or cloud model, the microphysics scheme, and closure methods. Figures of results with various convection schemes as implemented in WRF for a single hurricane case are included as illustration. A strong point is this review is pretty comprehensive, also in its referencing. Quite some work has gone into collecting and describing the many variations in formulations that have been proposed in the past few decades. The tables that summarize these formulations can well function as an overview for someone new to the field.

Thanks for the comments. Please note that as stated in the title and in the abstract, the paper is a review of the empirical values and assumptions. It is not a review of convection schemes. To make explicit the choices we had to organize the paper in some way, so we followed the conventional 'convection schemes' classification. Some information is required on those to frame the discussion, but the focus is on the empirical parameters and assumptions. There are already excellent papers specifically devoted to the taxonomy of the convection schemes. The paper is exactly what referee #1 says, "(the paper) presents a complete overview of the different assumptions made for entrainment/detrainement, microphysics which are rarely discussed."

That said, after reading the manuscript I could identify a few significant shortcomings. Firstly, the overview of convection schemes overly focuses on rather classic approaches, and fails to cover some important new developments in the field. These include recently proposed unified approaches, new concepts that address the ever more important grey zone problem, and new schemes capable of representing convective organization and memory. Not fully covering these recent developments, which result from active and intense ongoing research, is a serious omission. In my view this is detrimental for the relevance of this paper for the science community. The manuscript feels rather outdated in that sense.

We have now added several sections covering the most relevant empirical values and assumptions in recent developments in convection schemes.

A second concern, and related to this point, is that the science objectives of this study are not clearly defined. What exactly is the purpose of this review? This remains unclear.

We cannot agree. It is a review on the empirical values and assumptions. It was already stated in the original manuscript, but to stress it even more clearly, we have rewritten the objectives as follows: "The goal of the present paper is to provide a comprehensive account of the empirical choices and assumptions behind the representation of convective precipitation in models. To the best of our knowledge, there is no such extensive review of the empirical values and assumptions in the convection schemes available in the literature."

Third, the main conclusions of this study (as stated in the abstract) are not objectively supported by the content that is presented. This mainly concerns the impacts of convection schemes on weather and climate, and the need for observational datasets.

We address this comment below in the General comments section.

Fourth, the introduction contains statements that are not true, while the introduction is also overly skewed towards motivation and does not explain what is unique about this particular study.

We have modified the introduction to better explain the contributions of this study. We have revised the text and made sure that it does not contain any statements that are not true.

Finally, the figures included are not fully described and explained.

The figures are merely illustrative of the differences between various choices. More below.

To summarize, although I do see the use of another review of convection schemes, the current version is incomplete, somewhat outdated and should be improved at various points.

As explained before, this is not another review on convection schemes but a review on the empirical values and assumptions made in those schemes. We ensured that this is clear to the reader by modifying the relevant part in the introduction. The manuscript is titled "Empirical values and assumptions in the convection schemes of numerical models", which also indicates the purpose of this work.

For these reasons I recommend major revision of this manuscript to address these issues, before it can be accepted for publication.

Thank you for suggesting a revision of the manuscript. We have carefully addressed all comments of both referees and we believe that the manuscript has been sufficiently improved for its acceptance.

*General comments:*

1. Convective parameterization is a science field that sees active development at themoment. This is driven among others by the realization that these schemes continue to cause significant uncertainty in climate predictions, and also by the shift towards higher resolutions that become feasible in state-of-the-art weather and climate modeling. While these developments are briefly mentioned at various points in the paper, this is not reflected in its organization and structure. Recently proposed new types of convection schemes which show promise in addressing these issues and are "breaking the parameterization deadlock" (Randall et al., 2003) are not discussed in a structural way. These include i) unified parameterizations based on the EDMF approach (Siebesma et al, 2013, and many follow-up papers about EDMF), ii) PDF-based schemes (e.g. Golaz et al., 2003; Larson et al, 2012), iii) schemes that are scale-aware, scale-adaptive and also introduce stochasticity due to subsampling of a convective population inside a GCM gridbox at higher resolutions (Honnert et al, 2020), and iv) approaches that successfully capture convective memory and spatial organization. Not explicitly covering and describing these new approaches, which are now in the stage of becoming widely adopted in operational models, makes the current version of the manuscript not reflective enough of the state of the art in this research field. In that sense the manuscript is somewhat outdated, and repetative of previous reviews of convective parameterization that have already covered the classic schemese and approaches in great detail. To make the review better reflect these new approaches I strongly recommend giving them their own category / subsection in the organization of the manuscript.

The topic of the review has been already discussed. Section 2 serves as a framework for the rest of the paper, but the focus is not on each convection scheme but on the empirical values and assumptions used. We agree that including more recent studies is beneficial for the completeness of the review. Therefore, we have added a description of the current developments in section 2.6 on PDF-based schemes, section 2.7 on unified models, section 2.8 on scale-aware and scale-adaptive models, and section 2.9 on models accounting for convective memory and spatial organization. Empirical values and assumptions used in these schemes have been added to various tables throughout the paper.

On a related note, while the bibliography is indeed extensive, I noticed that at some points the referencing is not accurate. Sometimes the first "breakthrough" studies that proposed a new concept are not referred to, but instead only a somewhat arbitrary selection of follow-up studies are mentioned. One example are new mass flux schemes as mentioned in section 1.2, page 7, of which only one very recent example is mentioned. Another example is the description of spectral mass flux models (section 4.1.2), of which many more have recently been proposed and explored (e.g. Wagner and Graf, 2010; Neggers, 2015; Suselj et al, 2013; Olson et al, 2016; Brast et al., 2018; Hagos et al, 2018). In my opinion the

review is by far not complete at these points, which is a serious omission. I recommend going through all sections again and to make sure that key groundbreaking studies (both classic and recent) are properly mentioned.

Thank you for this comment. We carefully went through all the references in our manuscript and we have replaced or added references to ensure that the most relevant studies are cited.

2. The introduction misses a clear statement of the main science objectives of this study. Is this paper meant to be just a review, or more? Both the introduction and the abstract are misleading in this respect. While the introduction elaborately emphasizes the importance of convection schemes and the need for their careful evaluation, it fails to clearly state that this paper is meant to be a review of all existing schemes out there. It is also not clearly mentioned what is new compared to previous reviews of this kind, how this particular review can help the community. I recommend adding clear statements about both the objectives and the novel aspects in the introduction.

We have revised the abstract and the introduction to clearly stress the contributions. In the abstract, we state: "These empirical values and assumptions are rarely discussed in the literature. The present paper examines these choices and their impacts on model outputs and emphasizes the importance of observations to improve our current understanding of the physics of convection. The focus is mainly on the empirical values and assumptions used in the activation of convection (trigger), the transport and microphysics (commonly referred to as the cloud model) and the intensity of convection (closure)." We have modified the second paragraph in section 1: "The goal of the present paper is to provide a comprehensive account of the empirical choices and assumptions behind the representation of convective precipitation in models. To the best of our knowledge, there is no such extensive review of the empirical values and assumptions in the convection schemes available in the literature." We believe that the objectives are fully explained in the abstract and in the introduction of our manuscript.

3. The abstract announces conclusions that are not objectively or adequately supported bythe contents of this study. This concerns i) the examination of impacts of choices of parameters in convection schemes on weather and climate, and ii) insights concerning the need for observational datasets for constraining these choices.

The core sections in the paper, i.e., sections 3 to 5, include a dedicated subsection where the impact of the empirical values and assumptions of each parameter is presented based on the results from the literature. As for the need of observations, we refer to numerous convective parameterizations that were developed based on observations, e.g., the Bett-Miller-Janjic scheme, or the Kain and Fritsch scheme. In addition, we present examples of observations that led to modifications in the parameterizations to better represent convective processes, e.g., the observations of Niziol et al. (1995) that yielded a modification of the minimum cloud-depth threshold in Kain and Fritsch (1993). We have moved the latter from the conclusions to section 3.1.4. To further support the claims stated in the abstract, we have added more references highlighting the importance of observations in improving convective schemes. For example, in section 3.3: "Using the GreenOcean Amazon (GOAmazon, Martin et al., 2016), the authors set the values for the dCAPE threshold and entrainment rate from 2014. The new values are $55\,\mathrm{J\,kg^{-1}s^{-1}}$ for the dCAPE threshold and $2.5 \cdot 10^{-4}\mathrm{m^{-1}}$ for the entrainment rate." Numerous statements like this have been added throughout sections 3 to 5.

Concerning the impacts of parameter choices, no new analyses are included that demonstrate this impact in a statistically significant way. As is usually the case in a review paper, this review refers to previous studies for describing such impacts. However, in my experience these impacts of parameter choices are extremely code-specific, and are not universal. It is often the case that errors in calibration of one parametric component are hidden by errors in another; a good example is the too-few, too-bright problem (e.g. Nam et al., 2014) or errors in cloud overlap masking errors in vertical structure (Neggers and Siebesma, 2013). This means that results from parameter studies with one GCM code do not necessarily translate to another. This danger is not mentioned, but is very relevant for the conclusions drawn in this paper. This should be discussed.

We agree that such issue should be discussed. It is now included in section 6: "However, the impacts of the empirical values in convection are extremely code-specific and often errors in calibration of one parameter are hidden by errors in another. Examples of these include masking errors in vertical structure due to errors in cloud overlap (Neggers and Siebesma, 2013) or the too-few, too-bright problem (e.g., Nam et al., 2014). Therefore, results obtained in one GCM with a particular set of empirical values might differ from results obtained in a different GCM with the same set of empirical values."

It is also not evident from the content how exactly observations can help in constraining parameter choices, yet is is presented as a major conclusion in the abstract. That convection schemes have many constants that need constraining is not a new insight. Exclusively using observations for this purpose is problematic, due to data gaps and instruments not being capable of sampling key variables in parametric equations. Various recent international efforts have been conducted to make progress in this respect, in the form of field campaigns (e.g. EUREC4A) or long-term deployment of instrumentation at meteorological supersites (e.g. Neggers et al, 2012; Song et al., 2013; Gustafson et al., 2016; Zheng et al, 2020). A thorough discussion of both the problems and opportunities for constraining parameter choices in convection schemes with modern observations is missing, but is needed to support this conclusion. I recommend adding a section on this topic if this conclusion is to be maintained; if not, then I would remove this conclusion from the paper.

We do not agree. We state that observations are needed to improve the current understanding of the physics of convection. Besides, many convective parameterizations were developed based on observations and many of the parameters in convective schemes use values based on observations such as BOMEX or ATEX field campaigns, among others. Modifications of these schemes usually come from comparisons of the scheme simulations with observational data. While we are convinced that this discussion is not of sufficient importance as to create a dedicated section, we have included the following text in section 6: "However, observations suffer from data gaps and the instruments used are not able to sampling key variables in parametric equations. Long-term instrumentation deployment at meteorological supersites (Neggers et al., 2012; Song et al., 2013; Gustafson et al., 2020; Zheng et al., 2021) or field campaigns (e.g. EUREC4A) have been conducted to alleviate these issues".

4. The introduction contains statements that are factually wrong. Precipitation in general circulation models is not just generated by convection schemes and/or microphysics schemes. Boundary layer schemes can also contribute to both convective and stratiform precipitation, and significantly so. In fact, precipitation in subtropical marine low level stratocumulus and trade wind shallow cumulus is often completely carried by the boundary layer scheme. This error should be corrected.

To our knowledge, most boundary layer schemes contribute to precipitation indirectly. Indeed, boundary layer schemes widely used in community models alter the fluxes and then the water content ($q$) and the cloud cover. This subsequently affects precipitation, but precipitation is prognosed in the microphysics and in the convection scheme. There is no precipitation output from the boundary layer schemes, but moisture and moisture tendencies. Thus, for instance in the ECMWF model: "Eddy-Diffusivity Mass-Flux (EDMF) framework is applied, to represent the non-local boundary layer eddy fluxes (Koehler et al. 2011). The scheme is written in moist conserved variables (liquid static energy and total water) and predicts total water variance. A total water distribution function is used to convert from the moist conserved variables to the prognostic cloud variables (liquid/ice water content and cloud fraction), but only for the treatment of stratocumulus. Convective clouds are treated separately by the shallow convection scheme." However, if the referee is aware of any boundary layer scheme producing precipitation directly, we will be happy to include it in the paper. As for now, we have modified the introduction as follows: "Numerical Weather Prediction models, Global Climate Models, and Earth System Models (NWP, GCMs, and ESMs) generate precipitation **mainly** through two parameterizations: microphysics of precipitation (MP hereafter) and cumulus parameterization (CP) schemes"

More generally speaking, this point also relates to my first comment, in that new unified schemes have recently been proposed that are less strict in the separate representation between shallow and deep convection, and between convective and stratiform rain. These unified schemes are not properly described. I recommend improving the text in the introduction, and the manuscript as a whole, to more precisely and accurately describe the reality in state-of-the-art general circulation models.

*This comment has been already addressed. We have added a section on unified schemes. In addition, we have included the relevant empirical values and assumptions used in these schemes.*

5. Two figures are included in the manuscript that are meant to illustrate the impact of convective parameterization on the representation of convective phenomena in a circulation model. To this purpose WRF simulations of a tropical cyclone are used. However, at the points in the manuscript that these figures are referred to they are not properly explained. What is the model setup, what are the boundary conditions and forcings, and which data is used to derive these? Because these details are absent, these results can not be independently reproduced. I recommend adding an appendix in which these simulations are adequately described.

*As the referee states, the figures are illustrative. Therefore, a detailed list of parameters is not included. In contrast to, e.g., validation studies, we do not expect that the readers would be interested in reproducing these simulations. Nevertheless, we have extended the simulation details in the figure captions.*

*Detailed comments:*

p1, title: ".. in the convection of numerical models". This does not make sense. I guess you mean "in the representation of subgrid-scale convection in circulation models", or something similar?

*We have changed the title to "Empirical values and assumptions in the convection schemes of numerical models" accounting for the suggestion by the other referee.*

p1, abstract: The first half (three sentences) is a general description of what convective parameterization is, and why it is important. As this is already generally known and does not reflect the particular contents of this study, I recommend to remove this part and instead add a few lines about what is new about this review compared to previous ones, its science goals, and what aspects are unique and worth remembering.

*We consider it common to begin the abstract by a general introduction of the topic, similarly as in many papers in the field. It is known to people specifically working in convective parameterizations, but not to the general readership of the journal. However, we have modified the abstract by emphasizing the contribution of this review and taking into account also the comments from the other referee.*

p5, line 64: ".. generate precipitation through two parameterizations: microphysics... and cumulus parameterizations". This is not true: boundary layer schemes formulated in moist conserved variables also produce significant precipitation. See my 4th main comment above.

*This comment has been addressed above.*

p5, line 67: ".. are intended ...". Intended by who? Please be specific.

*We have rephrased the sentence.*

p5, line 77: "biota". What does this mean?

It is the animal and plant life of a particular region, habitat, or geological period. We believe that this is commonly known and does not need to be explained to the readers. Anyway, the meaning follows from the next sentence.

p7, line 123: please add the key paper by Wyngaard (2004) as well as the recent review paper by Honnert et al (2020) to the list of references about the convective grey zone. See also my 1st main comment above. Also, "grey scales" is a term I have not come across before; I suggest to stick to "grey zone"

We have added the relevant references. Thank you for your comment about gray scales. We have corrected this mistake.

p7, line 129: Why only refer to this paper for scale-aware parameterizations? Many more have been proposed by now, also much earlier than the study cited here. Please perform more thorough literature research on scale-aware parameterization, and add the most relevant papers on this topic here.

We have added more references to the scale-aware parameterization.

p8, line 136: "latest decadal survey". Do you mean the NAS one mentioned in the previous paragraph? This is not clear; also, not everyone is familiar with this survey.

We have modified the paragraph to make it clearer: "Convective processes have been identified as a major source of uncertainty (Jakob, 2010; NAS, 2018, hereafter decadal survey), and dedicated efforts are needed to fill the gaps in our present knowledge of the processes involved.
Owing to the influence of convection on climate and weather events over a large range of spatial and temporal scales, one of the most important objectives of the latest decadal survey is to improve the predictions of the timing and location of convective storms, and their evolution into severe weather. Besides the drawbacks associated with the spatial resolution, the multiscale interactions leading to the organization and evolution of convective systems are difficult to observe and represent."

p8, line 143: "crudely". Simple parameterizations are not necessarily crude. This is not scientific language, and is also somewhat insulting towards scientists who have spend significant effort in developing and implementing such parameterizations in GCMs.

This word has been used before when referring to convective parameterizations (e.g., IFS documentation, Cy47r1 Operational implementation, ECMWF, 30 June 2020). However, we have modified the sentence: "…described with simple parameterizations…".

p8, line 146: "computing-intensive": do you mean that convective parameterization is always computationally demanding? Then please explain why this is the case.

What we meant here is that for global climate simulations at high resolution and for long time periods, e.g., a century, the required computing time would be very high if no convective parameterization is used. The sentence has been changed: "While models will likely increase their nominal resolution in the next decade, it is also likely that global, century-long simulations from multi-ensembles under different assumptions will need to resort to parameterizing convection to reduce the computational burden."

p8, line 152: ".. Fig. 2". This is the first time the figures are referred to, and would be a good point to explain what exactly we see in them, and why this is relevant in this context. See also my 5th main comment above: these simulations are not adequately described, and can not be independently reproduced. And why is testing convection schemes for a cyclone the most relevant? Usually convective parameterizations are tested for simpler cases (such as locally forced continental convection) in a

simpler setting (e.g. Single Column Models combined with a weak temperature gradient approach). Please explain these choices.

We have added the following sentence before the figure appears in the text: "Indeed, as shown in Fig. 2 for the 6-hours total accumulated precipitation for Typhoon Chaba, even today model outputs look different depending on the cumulus parameterization used." Both Fig. 2 and 3 are just illustrative to show that the simulation results are different depending on the convection scheme used. We have also added more information on how the simulations in both figures were performed, as already mentioned in an answer to a previous comment. As for simulating a cyclone, we selected this case due to the societal relevance and deeper physical complexity of this type of atmospheric events compared to idealized cases.

p10, line 160: "between cumulus clouds". Parts of convective updrafts are non-cloudy, yet in this state still contribute significantly to the total vertical transport. Please change your labels such as "cloud model" to account for this.

We have changed this sentence: "The main assumptions in convective parameterizations concern the trigger model, the transport and microphysics, commonly referred to as the cloud model in classical convection schemes, and the closure of the scheme (Fig. 3)."

p10, line 169: Please explain what CISK means.

We have added the explanation: "CISK states that cyclones provide moisture that maintains cumulus clouds, and cumulus clouds provide the heat that cyclones need."

p10, line 190: "no cloud models are needed". I would phrase this differently, because the cloud model is still needed; but it is now hidden in the adjustment procedure.

We have changed this sentence for: "the physical properties of clouds are implicit and no cloud model has to be explicitly specified."

p11, lines 194-195: "very large precipitation... rarely observed in nature". No reference is provided to back up this statement; please cite a paper that shows this.

We have added a reference to the paper of Emanuel and Raymond (1993), who mentioned this.

p11, line 200 and onwards: For clarity, I suggest to include a simple equation for the adjustment, including a time scale. Different forms of adjustment are possible (e.g.
Newtonian); can all be described by the same equation?

We have added the adjustment equations described in Betts (1986) for the adjustment of temperature and moisture towards reference profiles.
As for the different forms of adjustment, we are not sure what the referee means.

p12, equation (1): These equations are not universally applicable to all mass flux schemes. Some are formulated in terms of conserved variables for moist adiabatic motion. So either it should be mentioned that (1) is only applicable to a subset of mass flux schemes, or the formulation should be changed to make it more generally applicable.

We have changed the whole section 2.3 including the formulation of the mass flux equations, which are now more general.

p12, line 241: "single entraining plume". Is this a steady state plume? Please explain. Also, I would add some references to the first classic papers about the rising plume model, such as Simpson and Wiggert (JAS, 1969).

Eq. (2) was not a steady-state plume as $\partial/\partial t$ was included in the formulation. Please note that this equation no longer appears in the manuscript as we have replaced it by more general expressions (Eq. (5.1), Eq. (5.2) and Eq. (5.3)). These equations are not steady-state either but we have added a sentence explaining that "numerous schemes have been proposed since then mostly using the steady state assumption, i.e., $\partial/\partial t = 0$".

A relatively recent review study about the rising plume model and its parameter choices that to my great surprise is not mentioned at all in this paper is the one by De Roode et al. (2012). How do the values mentioned in that paper overlap with those summarized in the Tables in this paper? See also my 1st comment about adequate referencing.

We have rewritten section 2.3 as mentioned before and we have added the reference suggested by the referee. As for the values in De Roode et al. (2012) and those in our tables, they overlap in the definition of entrainment rates (e.g., Gregory, 2001) as well as in the velocity condition to trigger convection (e.g., Jakob and Siebesma, 2003), to find cloud top (e.g., Bechtold et al., 2004), and convective closures defined in terms of the updraft vertical velocity (e.g., Neggers et al., 2009).

p12, line 244: ".. the i-th cumulus cloud". Is this a single cloud, or a sub-ensemble of clouds, such as clouds of a certain size or strength? Please define.

As Eq. (2) is expressed using the subscript $i$, it is valid for a sub-ensemble of clouds. For one single cloud the subscript $i$ would not appear in the formulation. However, this equation has been replaced by Eq. (5.1), Eq. (5.2) and Eq. (5.3), as already mentioned before. These expressions do not include the subscript $i$ and thus they refer to a single cloud.

p12, last line. The model described by equations (1) and (2) does not cover various new mass flux approaches, such as EDMF or pdf schemes. These have successfully been applied to precipitating convection, and can actually deal with scale adaptivity. See also my 1st main comment above.

Eq. (1) and Eq. (2) have been replaced by more general ones, as explained before.

p13, line 252: "produce too little heavy rain and too much light rain". Do all convection schemes suffer from this, or just one implementation in one particular GCM? See also my 3d main comment above.

We have modified the reference on this line to include studies that reflect this issue. For example, Sun et al. (2006) analyzed precipitation characteristics in eighteen coupled models, each of them with their own convective parameterizations. In terms of pattern and frequency, most models overestimated light precipitation frequency (1-10 mm/day) and underestimated heavy precipitation intensity (> 10 mm/day). It is worth noting that two models (PCM and CGM3.1) using the same parameterizations for stratiform and convective precipitation produced different values of mean JJA light and heavy precipitation frequency, especially over Africa, and of light and heavy precipitation intensity, especially over Africa (light) and South America (heavy). Therefore, results are code-specific as the referee pointed out in his/her third main comment.

p13, line 253: "Pritchard et al, 2011": I would also refer to Guichard et al. (2004), who first properly documented this behavior and which paper was also published 7 years earlier (which is a long time in science). See also my 1st comment about adequate referencing.

We have changed the references in this part and added Guichard et al. (2004) as suggested by the referee.

p16, line 350: "widely used at ECMWF". What do you mean with "widely". Please add a reference to a study that shows this.

By "widely" we meant that the temperature perturbation trigger in a form similar to the one described in Bechtold et al. (2001) has been used in many versions of the ECMWF IFS model (Bechtold et al., 2004, 2008, and IFS documentation: https://www.ecmwf.int/en/publications/ifs-documentation). We have updated the manuscript accordingly.

p22, line 445: "convective memory". This topic is intensely researched at the moment, yet is only briefly mentioned here. I think it deserves much more attention, even its own section. Doing so would make this review paper a lot more relevant and up to date.

Thank you for this comment. We have now included section 2.9 about convective memory and spatial organization as suggested.

p27, concerning the discussion on entrainment: I think it makes sense in this discussion to also refer to these recent papers, also in the table:
https://doi.org/10.1029/2019JD030889, https://doi.org/10.1175/JAS-D-20-0377.1. Apart from providing new insights, they also use observational data to constrain entrainment rates, which is relevant for this review

We have added the references suggested by the referee to our discussion on entrainment and detrainment rates in section 4.2.2 as well as in Table 5.

**Response to referee – second revision**

Referee comments shown in black
Authors' responses shown in blue

The authors have done well in improving the manuscript at various points, in response to the feedback provided by both reviewers. This includes a more complete coverage of all types of convection schemes that have been proposed, including more modern and unified approaches. The authors have also put convincing effort into getting the references right, where in the first submission a few key publications were ignored. All of this has made the paper more complete and concise, which is recommendable. As I already stated in my first review, I really do appreciate the significant amount of work that has gone into scanning all convection schemes and summarizing their essential assumptions and settings.

Thank you very much for the comments.

That said, some of my main concerns have not been adequately addressed. A few specific points I raised and some questions I asked remain unanswered or were side stepped in the response. These still open issues, which are also important, are summarized below. I remain of the opinion that these concerns need to be adequately addressed before publication is possible.

We address this comment in the Main concerns section below.

In some scientific journals a failure to address major concerns first time round automatically leads to rejection. I would still recommend a major revision, mainly because I do see merit in this work. So, I leave that decision to the editor.

Thank you for suggesting a revision of the manuscript. We have carefully addressed all comments and we believe that the manuscript has been sufficiently improved for its acceptance.

**Main concerns**

1) In response to my first major comment, and at various other points, the authors state that "Please note that as stated in the title and in the abstract, the paper is a review of the empirical values and assumptions. It is not a review of convection schemes". I fully disagree, for the following simple reason: these (parametric) assumptions are the defining parts of convection schemes, and what makes them differ from each other. This implies that one cannot separate the two. When the objective is to provide a review of empirical assumptions, then this in effect comes down to reviewing (differences between) convection schemes. This might be a disagreement on semantics. Still, it is important to clarify this in the manuscript, to avoid any confusion with the reader (including myself).

Indeed, reviewing the empirical values and assumptions implies reviewing convection schemes, but what we mean is that we focus on the values and assumptions and not on reviewing each particular convection scheme separately.

We have deleted line 95 ("This is briefly and schematically done, as the focus of this paper is not reviewing the convection schemes in themselves but to identify the assumptions and empirical values embedded in them").
I also disagree that this review is the first of its kind ever, as for example stated in the introduction (line 75, "To the best of our knowledge, there is no such extensive review..."). I know of at least one previous study. De Roode et al (2012, doi.org/10.1175/MWR-D-11-00277.1) discusses empirical assumptions and values as feature in the updraft kinetic energy equation and includes a thorough literature review. In structure and content, their Table 1 is very similar to, say, Table 6 on entrainment rates in this manuscript (among others). For this reason, I think this statement should be softened, to properly acknowledge previous work.

We agree that other reviews, such as the work of De Roode et al. (2012), thoroughly discuss the empirical assumptions and values used in convective models. However, these reviews usually focus on one particular parameter. For example, De Roode et al (2012) mostly focus on the vertical velocity equation and does not include a revision of the detrainment or the closure, among others. To make it clearer, we have added the following sentence in line 80: "There are indeed several reviews thoroughly discussing the empirical values and assumptions in convective models (e.g. De Roode et al. 2012), but they are generally focused on a particular parameter."

2) In my second main comment I asked to provide a clear statement of what is the overarching science objective / higher goal of this review, or in other words, what is the added value of this review. The response is as follows: "The goal of the present paper is to provide a comprehensive account of the empirical choices and assumptions behind the representation of convective precipitation in models." But this is not an answer to my question. I ask what we learn from reviews like this. Is it just a collection of long tables with many values and references, acting as a library index? Or does it yield new insights? This remains unclear, also in the revised version. Most scientific review studies provide a vision like this, so I was expecting this as a reader.

We have added the following sentence in the abstract:

Such information can assist satellite missions focused on elucidating convective processes (e.g. the INCUS mission) and the evaluation of those model output uncertainties due to spatial and temporal variability of the empirical values embedded into the parameterizations.

And also added this paragraph in the introduction:

The scientific interest of our endeavor is twofold. First, it can assist dedicated satellite missions such as the Investigation of Convective Updrafts (INCUS) mission, a new Earth Venture Mission-3 (EVM-3) of three SmallSats expected to be launch in 2027 that aims to increase our knowledge of precipitation processes, and specifically on the many nuances behind convection (Stephens et al. 2020). Indeed, INCUS aims to advance our present understanding and modeling of convection on the directions identified in the 'decadal survey' (cf. Jakob, 2010; National Academies of Sciences, Engineering and Medicine, 2018, hereafter 'decadal survey'). The precise description and rationale behind the empirical parameters in the parameterization of convection can help INCUS and similar missions to focus on the key parameters, and to analyze their impacts on weather and climate models.
Another science goal of our review is to pinpoint the more relevant empirical values so systematic sensitivity studies can be readily carried out. We exemplify the latest goal showing that the spread of a perturbed ensemble of just a few parameters can be substantial. Thus, we have used the European Centre for Medium-Range Forecasts (ECMWF) Integrated Forecasting System (IFS) to perform a sensitivity experiment with seven parameters (organized entrainment, entrainment for shallow convection, turbulent detrainment, adjustment time, rain conversion, momentum transport, and shallow vs deep cloud thickness). While this is a small subset of the many parameters we have identified in this review, and the experiment is intended as an illustration of the spread in the simulations for two tropical storms, the case invites to more systematic runs in both space (global coverage) and time (decadal simulations) over the whole empirical set of parameters of any given model. The spread of the results will help to gauge the uncertainties due to the empiricisms embedded in the convection modules, and to constraint those through dedicated campaigns and targeted observations.

3) The response provided does not adequately address my concern. The response is: "... we refer to numerous convective parameterizations that were developed based on observations, ...", and "We state that observations are needed to improve the current understanding of the physics of convection". All convection schemes are based on at least a few observations; that is common knowledge, and not my point. Instead, my question is what your tables can tell us about what more we need in terms of observations to make progress, for example to break the ongoing "parameterization deadlock" (Randall et al, BAMS, 2003). Has the use of observations by the convective modelling community so far sufficient? Or do we need to find new ways to adequately constrain assumptions and calibrate

parameterizations, in a statistically significant way? And if so, how can we most efficiently use modern extensive big datasets to this purpose? Having put so much work into delving through all these schemes in detail and listing all the key components (which I find really impressive), you are now in a unique position to make a statement about that. The reader expects that vision, and accordingly, I thoroughly recommend adding it. Not doing so is an omission. Hence my advice to add a section dedicated to this topic. This advice still stands.

In section 6 we have already mentioned that "…observations suffer from data gaps and the instruments used are not able to sampling key variables in parametric equations." Therefore, other techniques to improve parameterizations were proposed, such as the use of CRMs, LES or SCMs. However, these techniques suffer from drawbacks, such as the ability of idealized simulations to represent the actual climate. More recently, a combination of observations and LES simulations is used, such as the one proposed in Neggers et al. (2012). We have added the following explanation in line 1683: "Despite the increase of observational supersites worldwide, data gaps still remain. A statistically process-level evaluation has been proposed by authors such as Neggers et al. (2012) or Gustafson et al. (2020), among others. This new approach consists in combining LES outputs with observations. Indeed, high resolution models provide additional information in 4D that is not possible to be obtained from point-based measurements (Gustafson et al., 2020). Another complementary approach to fill observational gaps and provide scientists with more information about the physics of convection is dedicated satellite missions such as INCUS. Although observations have long been used to tune parameters in convective schemes to reduce errors, it is still unclear whether these tuned parameters based on particular datasets can improve model skills across different locations, model resolutions or atmospheric events. Spaceborne sensors can help to palliate the situation through global, homogeneous and time-extended observations. INCUS and forthcoming missions can shed new light on the empiricisms and help characterizing the adequate values for the many empirical parameters in models. As described above, it is known that model results are sensitive to the empirical values in convection."

4) Judging from the response, I think there is some confusion about what is meant by "boundary layer scheme". This is not always the same in each model. Some interpret the boundary layer as only representing dry (non-saturated) turbulence and convection; others consider cloud layers as intrinsic part of the boundary layer, thus including shallow cumulus and stratocumulus. So, to avoid unnecessary confusion with the reader, I recommend to clearly define early on in the manuscript what exactly is meant by "boundary layer scheme", and then to consistently use this definition throughout the manuscript. This template may sometimes not be applicable to more unified schemes, in which microphysics, shallow transport and deep transport are interwoven and cannot be strictly separated anymore into unique and single modules, as was classically done.

We have added an explanation on boundary layer schemes in section 1: "While other schemes, such as planetary boundary layer (PBL) parameterization used to parameterize turbulence within the PBL without accounting for moist convection …"

That said, I know of quite a few boundary layer schemes that do generate precipitation. For example, in contrast what you say, the IFS EDMF scheme makes use of plume equations that do include a source/sink term representing precipitation. See IFS documentation C47R3 chapters 3.2 and 6.3.1. So the EDMF scheme does produce rain in case the EDMF plume condensates. Second, when the IFS Tiedtke scheme is in shallow cumulus mode, it is in effect generating boundary layer precipitation, and can thus be classified as a "boundary layer scheme". This rain can be significant, as we have learned from field campaigns on Trade wind cumulus such as RICO and EUREC4A.

The IFS scheme is just one example; there are more boundary layer schemes that directly generate precipitation. The EDMF scheme of Neggers (2009, doi.org/10.1175/2008JAS2636.1) also produces rain. The CLUBB scheme as implemented in CAM (Larson et al., GMD, doi.org/10.5194/gmd-8-3801-2015) also generates precipitation when in boundary layer mode; see their Section 2.4 and Fig. 1.

Thank you very much for the insight into boundary layer schemes that directly produce precipitation.

We have added these examples in section 2.7 of the manuscript.

5) "A detailed list of parameters is not included". I do not understand; which parameters do you mean? In the figures? In my opinion, all aspects of figures should be fully explained in a scientific publication, even if they are just meant to be illustrative. This is just good scientific practice: all science should be reproducible, otherwise it is meaningless.

We meant that in the first version of the manuscript we did not include detailed information about the setup used to perform the simulations, e.g., microphysics scheme or radiation scheme, among others. However, we added these details in the revised version of the manuscript after the referee recommended to do so.

I also find new Figure 3 somewhat simplistic. For example, it depicts shallow convection as exclusively non-precipitating, which by now we now is totally untrue (see the many studies based on RICO, NARVAL and EUREC4A data and simulations). Second, it conforms to the old idea of how convection should be modelled, using a single bulk plume and a modular approach. The schematic certainly does not accommodate unified or spectral approaches in modelling convection. See for example Fig. 1 in Arakawa and Schubert (1974), which is a much more realistic example of how a convective population works. If this review is to be comprehensive, as is claimed in the introduction, the figure should accommodate all approaches, not just the classic bulk one.

We have changed the figure accordingly.

**Response to topical editor – third revision**

Editor comments shown in black
Authors' responses shown in blue

I think that in the current revision, you have addressed the reviewer's comments much better, I find only the answer to point 3 still not adequate. The reviewer states a few important questions:
* Has the use of observations by the convective modelling community so far sufficient?
* Or do we need to find new ways to adequately constrain assumptions and calibrate parameterizations, in a statistically significant way? And if so, how can we most efficiently use modern extensive big datasets to this purpose? I would like to ask you to make sure that these questions are _explicitly_ answered in your text. The reviewer is correct that you are in a unique position, and indeed it would be an omission if this is not done properly.

Thank you very much for the comments. To include and explicit answer to these questions, we have modified lines 1681 to 1688 in the previous version of the manuscript as follows: "The use of observations by the convective modeling community has not been sufficient so far. The reasons being twofold. Basic convective quantities like mass flux and important parameters like adjustment time scale, entrainment and microphysical parameters can often be only indirectly inferred from observations like infrared and microwave satellite data, radar data, rainfall rates, radiosonde networks and reanalysis data. When we say that they are indirectly inferred, we mean that these quantities are adjusted to optimize the model fit to the observed radiative and surface fluxes as well as the observed temperature and wind field. On the other hand, long-term instrumentation deployment at meteorological supersites (e.g. Neggers et al., 2012; Song et al., 2013; Gustafson et al., 2020; Zheng et al., 2021) or dedicated convection field campaigns like GATE, TOGA-COARE, DYNAMO, PECAN (Geerts et al., 2017), EUREC4A (Bony et al., 2017), to mention a few, have been conducted to quantify convection and its effect on the large-scale flow, and powerful LES data are available with statistical samples of the convective updraft and downdraft properties. However, the dilemma is that these data are only available locally or for specific setups, LES data also need to be constrained by observations and an accurate convection parameterization in a global model needs to be constrained globally.
Modern extensive big datasets such as those derived from COPERNICUS data are very relevant to constrain assumptions and calibrate parameterizations. Recently, (Neggers et al., 2012) and (Gustafson et al., 2020), among others, have provided a successful attempt to reconcile observations and LES data. This new approach consists in combining LES outputs with observations. Indeed, high-resolution models provide additional information in 4D that is not possible to be obtained from point-based measurements (Gustafson et al., 2020). The complementary approach consists of new dedicated satellite missions such as INCUS or the follow-on to CloudSat and CALIPSO, which can provide global, homogeneous and time-extended observations. Satellite estimates of the convective mass flux are becoming available (Jeyaratnam et al., 2021) and new missions are in the planning to fill the gap in global, multiple-regime observations of convection".

**Authors' changes in the manuscript**

1. Title
2. Abstract
3. List of acronyms
4. Section 1 Introduction: added why we do this review and what is its scientific interest.
5. Section 1.1 Model parameterization: added alternative approaches to tuning.
6. Figure 1 and Figure 2 captions: corrected typhoon name and addition of simulation details.
7. Addition of Figure 3.
8. Addition of Figure 4 to include a schematic representation of the convection parameterization.
9. Section 2.1 Convergence schemes: definition of CISK and explanation of 'b~0' added.
10. Added equations in section 2.2 Adjustment schemes.
11. Change equations in section 2.3 Mass flux schemes to make them more general. Addition of the vertical velocity equation and its empirical values in table 2.
12. Addition of sections PDF-based schemes, unified models, scale-aware and scale-adaptive models, and models accounting for convective memory and spatial organization.
13. Change name of the most used trigger functions in convective parameterizations.
14. Section 3.1.1: clarify that subsequent modifications of the Tiedtke scheme cannot longer be classified as moisture convergence schemes.
15. Table 4 modified (now Table 5).
16. Distinction between empirical values and assumptions used for deep and shallow convection. More values added for shallow convection within the text and tables throughout the paper.
17. Better explanation of spectral and bulk models and its differences in sections 4.1.1 and 4.1.2.
18. Distinction between conversion of cloud water to rainwater and precipitation efficiency in section 4.3.
19. Section 5 Closure. Better explanation of the different closures, especially, the stochastic one.
20. Extension of section 5.2 Impact of closure on convective models.
21. Reformulation of section 6 Conclusions to include an historic view of the development of the convection scheme organized, a list of the remaining challenge for convection parameterization, the use and limitations of observations to calibrate parameterizations, and a summary of the main differences between shallow and deep convection regarding trigger, cloud model and closure.
22. Acknowledgements.
23. More references have been added throughout the paper to include the most relevant papers in the topic.